# Incentivizing Reasoning for Advanced Instruction-Following of Large Language Models

**Yulei Qin**[1]    **Gang Li**[1]    **Zongyi Li**[1]    **Zihan Xu**[1]    **Yuchen Shi**[1]

**Zhekai Lin**[1,2]    **Xiao Cui**[3]    **Ke Li**[1]    **Xing Sun**[1]

[1]Tencent Youtu Lab    [2]Xiamen University    [3]The Chinese University of Hong Kong
`yuleiqin@tencent.com`
 Codes & Datasets          Model Checkpoints

## Abstract

Existing large language models (LLMs) face challenges of following complex instructions, especially when multiple constraints are present and organized in paralleling, chaining, and branching structures. One intuitive solution, namely chain-of-thought (CoT), is expected to universally improve capabilities of LLMs. However, we find that the vanilla CoT exerts a negative impact on performance due to its superficial reasoning pattern of simply paraphrasing the instructions. It fails to peel back the compositions of constraints for identifying their relationship across hierarchies of types and dimensions. To this end, we propose RAIF, a systematic method to boost LLMs in dealing with complex instructions via incentivizing reasoning for test-time compute scaling. First, we stem from the decomposition of complex instructions under existing taxonomies and propose a reproducible data acquisition method. Second, we exploit reinforcement learning (RL) with verifiable rule-centric reward signals to cultivate reasoning specifically for instruction following. We address the shallow, non-essential nature of reasoning under complex instructions via sample-wise contrast for superior CoT enforcement. We also exploit behavior cloning of experts to facilitate steady distribution shift from fast-thinking LLMs to skillful reasoners. Extensive evaluations on seven comprehensive benchmarks confirm the validity of the proposed method, where a 1.5B LLM achieves 11.74% gains with performance comparable to a 8B LLM. Evaluation on OOD constraints also confirms the generalizability of our RAIF.

## 1 Introduction

Large language models (LLMs) exhibited remarkable performance on real-world tasks [1, 2, 3, 4]. Such generalizability is built upon the instruction-following capabilities of LLMs [5, 6, 7]. To benchmark whether LLMs can produce the desired outputs under complex instructions, existing studies predominantly focus on modeling various types of constraints and rules where all requirements are expected to be satisfied simultaneously [8, 9, 10, 7]. Recently, the compositions of constraints (*And*, *Chain*, *Selection*, and *Nested*) have been systemized [11] to enhance the complexity of instructions and demonstrate that LLMs still fail to meet expectations under intricate structures. These complex instructions, which are often composed of multiple sub-instructions, enforce various constraints on the expected responses. Existing LLMs either ignore certain constraints in *And* structures or misinterpret the instruction to respond to the wrong sub-instructions in *Selection* structures.

To improve LLMs in solving complex instructions, most prior methods leverage two kinds of techniques: 1) supervised fine-tuning (SFT) [12, 13], and 2) template-guided inference [14, 15]. Despite

39th Conference on Neural Information Processing Systems (NeurIPS 2025).

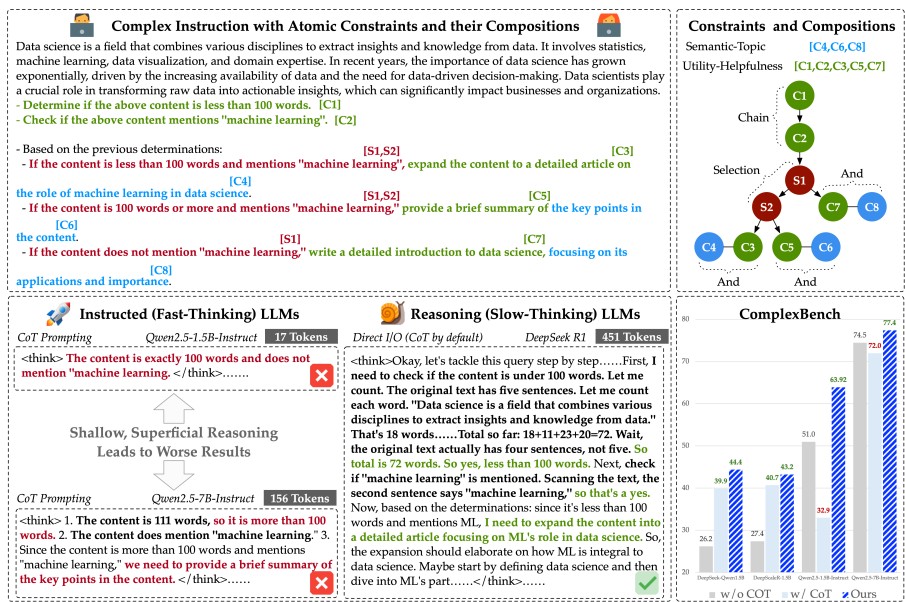

Figure 1: Complex instructions with various atomic constraints and compositions pose great challenges to instruction-following capabilities of LLMs (The above example and its structure are from the ComplexBench [11]). Our preliminary experiments demonstrate that the CoT prompting of existing LLMs often elicits shallow reasoning that blindly, mechanically responds to the request without formulation of structured analyses. In contrast to R1 and QwQ, most fask-thinking models cannot benefit from the vanilla CoT at all due to such superficial nature (see Sec. A.7). Our proposed method boosts deep reasoning of both fast- and slow-thinking LLMs under complex instructions.

their effectiveness, these methods are all task-specific. They are prone to overfitting constraints in the training set and fail to generalize to the unseen ones [16]. For the former, a large amount of curated instruction-response pairs are required to ensure diversity. For the latter, it is almost impossible to exhaustively enumerate the templates for problem decomposition, reflection, and refinement beforehand. Therefore, it calls upon a scalable solution that is both constructive and generalizable without meticulous manual efforts. One intuitive method is to directly apply chain-of-thought (CoT) [17] on complex instructions, where LLMs might benefit from the free-form thinking for structure analysis with highlighted, valid constraints (see Fig. 1). However, our preliminary experiments show that such a prospective solution brings minimal or even negative performance gains, which contrasts strikingly against its effectiveness on maths problems. We observe that such a discrepancy arises from the underlying problem of "superficial" reasoning, where LLMs simply summarize the instructions briefly without developing the solid thinking upon the instruction itself. During such shallow, parroting-style reasoning, critical constraints and rules can be ignored and thereafter such misalignment leads to degraded performance. For maths problems, it is indispensable for LLMs to formalize step-by-step, divide-and-conquer process to achieve the final answer. On the contrary, for complex instructions, there exists no such nature or tendency of LLMs to forge deep reasoning as they are aligned to directly deliver responses without intermediate steps. Under such circumstance, we target reasoning that truly empowers LLMs with strategic planning and adherence to rules.

In this paper, we propose **RAIF** to leverage **R**easoning for **A**dvancing the **I**nstruction-**F**ollowing capabilities of LLMs under the context of complex instructions. Inspired by the success of o1 [18], R1 [19], and QwQ [20], we resort to CoT for tackling complex instructions. We provide a practical guide for cultivation of effective reasoning tailored via reinforcement learning (RL). Specifically, our method addresses the following two main challenges: 1) *the data scarcity of diverse complex instructions* and 2) *the secret recipe behind the formulation of effective CoT*. For the former, although various data synthesis methods [12, 21, 22, 23, 24, 25, 26, 14] have been proposed to pile up instructions, they do not take into consideration the taxonomy of constraint types and compositions. In contrast, we stem from the atomic rules, constraints, and their compositions to perform LLM-based evolving across various tasks and domains. Besides, our instruction scaling is integrated with our RL-based reasoning stimulation. Both rule-based and model-based verification approaches, which

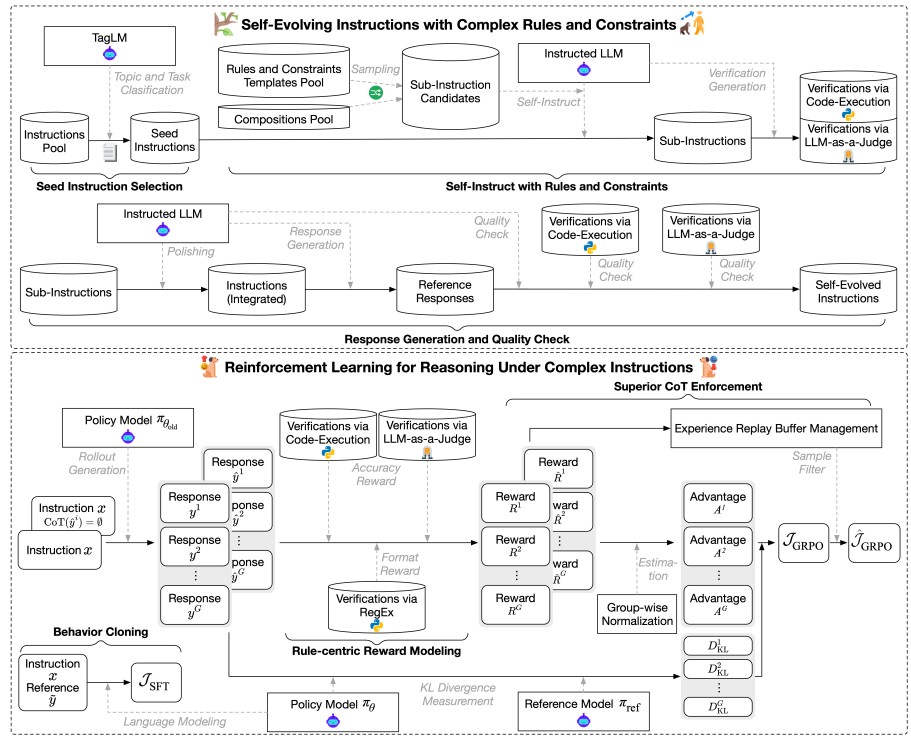

Figure 2: Illustration of the proposed method for advanced instruction-following via reasoning.

serve as reward sources later, are paired with the instructions during generation. For the latter, existing approaches that seek to reproduce R1-like reasoning [27, 28, 29, 30] are all restricted to solving maths problems with pure rule-based rewards, which are inapplicable for our settings. We consequently keep an eye on the differences between maths and complex instructions, and propose our RL recipe with rule-centric reward modeling, enforcement on superior CoT, and control on policy distribution drift. Our contributions are three-fold: 1) We propose a systematic method to boost LLMs under complex instructions via LLM-based evolving for instruction synthesis and reasoning-driven reinforcement learning. 2) We advance the RL to incentivize reasoning for complex instructions by addressing the superficial reasoning nature. To the best of our knowledge, our work is the **pioneer** that de-mystifies the recipe behind the cultivation of reasoning under complex instructions. 3) We validate our effectiveness on **seven** comprehensive benchmarks along with extensive generalization studies across model families, sizes, cold/warm-start reasoners, and OOD constraints.

## 2 Related Works

### 2.1 Evaluation of Instruction Following of LLMs

Tremendous benchmarks have sprouted to specifically evaluate instruction following capabilities of LLMs in terms of semantic [31, 32, 33] and format constraints [13, 34, 9]. Following the representative IFEval [7], recent studies extend the evaluation settings towards wild chat [35, 36, 37], long context [38], multi-lingual [39, 40, 41], multi-turn [42, 43], and multi-modal [44, 45] scenarios. The hierarchical categorization [46, 47, 48, 49] and down-streaming applications [50, 51, 52] are considered to collect fine-grained instruction for comprehensive evaluation.

In this paper, we measure the instruction following capabilities on the most common, comprehensive benchmarks including IFEval [7], ComplexBench [11], CELLO [10], CFBench [8], FB-Bench [53], FollowBench [54], and InfoBench [55]. Different from studies that merely focus on the evaluation itself, we systematically investigate a general, scalable pipeline that enhances the instruction following consistently across models and benchmarks via test-time compute scaling.

## 2.2 Complex Rule and Constraint Following

Numerous studies enhanced rule and constraint following capabilities via data engineering [56]. The top concerns are around the synthesis of complicated constraints without costly human intervention. Specifically, WizardLM [12] presents both in-breadth and in-depth evolving to generate complex instructions for tuning. Air [57] stems from the simple instruction distilled from documents and then iteratively add one or more constraints to increase complexity. Back-translation [58] is another popular technique to introduce additional constraints. Self-play [59] targets at generating constraints that can be verified by codes. SPAR [60] combines both self-play and tree-search for iterative refinement. The detailed taxonomy of rules and constraints [8, 61, 14], together with their structures and relationship [62, 11, 63], allows precise synthesis control over instruction types. With respect to the post-training technique, most methods adopt the SFT [64] and RL [65, 66, 67, 68, 69] for preference alignment. Moreover, training-free workflows have also been proposed to decompose the instructions [70] and exploit verifiable feedback [71] for refinement until all the check-boxes are ticked [72]. The most recent recipe of post-training by Tülu 3 [16] augments the existing instructions with IFEval constraints and optimizes the LLM with RL for better alignment. The differences between Tülu 3 and ours are fourfold: 1) Problem definition. We aim at solving a broad range of complex instructions with various constraints and their compositions via reasoning. Tülu 3 focuses only on responding to the IFEval-style instructions without the preceding reasoning, and therefore exhibits limited generalization on out-of-domain (OOD) constraints. 2) Constraint type and structure. We follow CFBench [8] and ComplexBench [11] respectively for the constraint categories and composition types. Tülu 3 is limited in code-verifiable constraints with "and" structures. 3) Reward modeling. Both code-execution and LLM-as-a-Judge are involved for verification. This greatly expands the tasks that can be handled by our method, where the semantic constraints with LLM-as-a-Judge verifications are complementary to the lexical and word constraints. 4) RL algorithm. Tulu 3 directly optimizes the response via the vanilla PPO algorithm. However, we are aimed at incentivizing reasoning for solving complex instructions. Since LLMs are prone to shallow reasoning and tend to muddle with a few sentences that ignore the constraints and their structures, we design the filtering mechanism that only selects responses with deep, true reasoning. Our proposed superior CoT enforcement acts as a bridge between reasoning and instruction-following quality.

Our work differs in two aspects. First, we propose a pioneering method that handles all kinds of atomic constraints and their combinations. Previous studies are either limited in constraint types (e.g., solely considering rules that can be verified by codes while neglecting semantic, style constraints) or composition structures (e.g., simply satisfying all constraints at the same time without regarding chaining and branching situations). Second, we resort to RL for reasoning specifically cultivated for improving instruction understanding and following. Our method exhibits generalization on multi-purpose and maths tasks, and bypasses the tedious design of decompose-and-conquer workflows.

## 2.3 Chain-of-Thought Reasoning

Recent progress on the chain-of-thought (CoT) [17, 73, 74, 75, 76, 77] has attracted attentions for advancing cognitive capabilities LLMs. Most of these approaches prompted LLMs to break down difficult questions into multiple small units to tackle them systematically before jumping to final answers [3, 78, 79, 80, 81, 82]. The emergence of OpenAI o1 [18] and DeepSeek R1 [19] has incubated test-time compute scaling techniques for long CoT, where deep reasoning with extensive exploration and feasible reflection are encouraged [83]. Both Monte Carlo Tree Search and RL with process and outcome rewards [84, 85, 86] are also emphasized in previous academic attempts to replicate slow-thinking LLMs. Compared with existing studies that focus on the mathematic and logic problem-solving, we are aimed at advancing instruction following capabilities. Motivated by R1 [19], we incentivize reasoning with group relative policy optimization (GRPO) [87] and demonstrate that RL with rule-centric rewards ultimately pushes the limits of LLMs in following instructions.

# 3 Methods

## 3.1 Problem Definition

Our goal is to incentivize reasoning of a LLM to advance its capabilities of handling complex instructions (see Fig. 2). Through systematic pipeline investigation and extensive experimental

analyses, we offer key insights and practical guidelines to arm LLMs with deep reasoning. Our research scope focuses on instructions that consist of one or more atomic sub-instructions with compositional structures (*And*, *Selection*, *Chain*, and *Nested*). Multiple constraints are enforced so that LLMs have to carefully comprehend the instructions and reason on *which sub-instructions to perform execution* and *how to obey all the rules*.

Given $x$ as a query that contains one or more compositional complex instructions, we consider a LLM parameterized as $\theta$ to be instruction-followed if its output $y$ satisfies all the constraints and requirements mentioned in $x$. A typical conditional distribution over the language modeling process can be denoted as $\pi_\theta(y_t|x, y_{1:t-1})$, where $y_t$ denotes the $t$-th token of $y$. The chain-of-thought process $\text{CoT}(y) \subset y$ refers to the tokens in the generated output $y$ that indicates the explanations of query intents, plannings of problem solving, and step-by-step deductions. The final answer following $\text{CoT}(y)$ can be simply extracted by $y \setminus \text{CoT}(y)$. It is noted that for fast-thinking instructed models, the CoT prompting tokens $x_{\text{CoT}} \subset x$ (e.g., `Let's reason step by step.`) are indispensable. While for reasoning models with slow thinking nature, $x_{\text{CoT}} = \varnothing$ is the default setting.

## 3.2 Self-Evolving Instructions with Complex Rules and Constraints

To address the scarcity of complex instructions, we propose to scale up instructions with various rules, constraints and their verification criteria via self-evolving.

**Seed Instruction Selection**   We start by selecting a set of seed instructions $D_{\text{seed}}$ from the commonly used WildChat [35] and Alpaca [88] datasets. To ensure the diversity of $D_{\text{seed}}$, we follow [89] to tag each instruction by its topics and tasks for a wide-range selection of task abilities. Details on the tagging and selection process can be found in Sec. A.4.1.

**Self-Instruct with Rules and Constraints**   We adapt self-instruct [21] under different fine-grained rules and constraints [8]. In view of the verification techniques, both code-execution [90] and LLM-as-a-Judge [32] are involved to provide evaluation feedback. For the former, we prepare a collection of constraint templates and their executable codes. Then, we randomly instantiate a combination of rules and constraints from the pool. To ensure their mutual compatibility, a pre-defined validity check is enforced to eliminate conflicts (e.g., `The first paragraph must start with...` and `Wrap your entire reply with...`). For the latter, we construct pairs of sub-instructions and scoring questions that stress on the style and semantic constraints. Such sub-instructions correspond to constraints with LLM-based evaluation (e.g., `Is the answer written in the tone of Confucious?`), which are complementary to those relying on code-based evaluation. An off-the-shelf LLM is utilized to perform few-shot in-breadth evolution. With respect to the composition of these atomic sub-instructions, we refer to [13] for definitions of *And*, *Chain*, and *Selection*. These sub-instructions are assembled for the integrated instructions. Details can be found in the Sec. A.4.2.

**Response Generation and Quality Check**   We use LLMs to generate responses and filter out low-quality query-response pairs that fail to pass the associated verification tests. Additionally, we observe that the self-evolved instructions still contain unreasonable constraints or nonsensical queries (e.g., `Give me a very short, concise, and clear response...The response should have 4 sections.`). Under such circumstance, we summarize seven typical issues and curate judgment prompts for LLMs to double-check the instructions and keep the valid ones (see Sec. A.4.3).

## 3.3 Reinforcement Learning for Reasoning Under Complex Instructions

We propose to incentivize reasoning of LLMs via RL [5, 19]. The development of CoT is optimized towards being structured and sophisticated, which ultimately leads to improved final answers. Without loss of generality, we adopt the GRPO [87] algorithm. Details can be found in Sec. A.2.

**Rule-Centric Reward Modeling**   To explicitly distinguish the reasoning from the answer contents, we employ a simple minimalist rule-based format reward that checks the existence of "`<think>`", "`</think>`", "`<answer>`" and "`</answer>`" tags. The format reward encourages the thinking contents encolosed solely between "`<think>`" and "`</think>`". It is noted that for R1-series

reasoning models, "`<answer>`" and "`</answer>`" tags are not necessary for answer extraction.

$$R_{\text{format}}^i = \begin{cases} +1 & \text{if } \mathbb{1}(\texttt{<think>...</think><answer>...</answer>} \in y^i), \\ -1 & \text{otherwise.} \end{cases} \tag{1}$$

With respect to the accuracy reward, the answer contents are extracted for comparison and evaluation only if the format constraint is satisfied. Compared with the maths problems that each has an exclusive ground-truth, the correct responses to complex instructions can vary greatly. Therefore, there exists no rigorous exact-match assessment [91]. Instead, we propose the rule-centric accuracy rewards that stem from verification standards. Specifically, we take into consideration the evaluation of responses in following each constraint and indicate its satisfaction condition as rewards. Given an instruction $x$ that contains $C$ atomic constraints $x_C = \{c_j\}_{j=1}^C, c_j \subset x$, the sampled response $y^i$ is judged as instruction-followed only if all the valid, active constraints $c_j \in x_C^{\text{active}} \subset x_C$ are satisfied. Accordingly, a piecewise reward function is defined via measurement is_followed($y^i|\cdot$):

$$R_{\text{accuracy}}^i = \begin{cases} +2 & \text{if is\_followed}(y^i|c_j) \ \forall c_j \in x_C^{\text{active}}, \\ \sum_{j=1}^C \frac{\mathbb{1}(c_j \in x_C^{\text{active}} \ \&\& \ \text{is\_followed}(y^i|c_j))}{\mathbb{1}(c_j \in x_C^{\text{active}})} & \text{elif is\_followed}(y^i|c_j) \ \exists c_j \in x_C^{\text{active}}, \\ -2 & \text{otherwise.} \end{cases} \tag{2}$$

It is noted that the detailed implementation of the verification depends on the constraints [8]. For the lexical-level constraints (e.g., keywords and phrases), the numerical constraints (e.g., letters and words), and the format constraint (e.g., JSON, XML, LaTeX, HTML, and Markdown), we resort to simple heuristics with python which provides precise feedback [7, 59]. In contrast, for the semantic-level constraints (e.g., themes and perspectives), stylistic constraint (e.g., writing styles, tones, and role-plays), linguistic constraints (e.g., dialects and morphologies), we bring in the reward model $r_\phi$ for judgment [13, 55, 11]. However, different from the original GRPO where $r_\phi$ implicitly scores the responses in terms of instruction following, we explicitly employ $r_\phi$ to check the following conditions of constraints. Besides, $r_\phi$ delivers scalar scoring in previous studies while we request for efficient boolean validation (`True` or `False`). Since multiple constraints might be active for steering generation, we develop the piecewise reward that promotes more constraint satisfaction while penalizing greatly the extreme cases. In total, our rule-centric reward is defined below.

$$R^i = R_{\text{format}}^i + R_{\text{accuracy}}^i. \tag{3}$$

**Experience Replay Buffer with Superior CoT Enforcement**   Compared with the maths tasks, instruction-following differs in that their reasoning processes are not compulsory. For maths, the step-by-step decomposition and derivation is a prerequisite to obtaining the final answer, which is naturally cultivated [27, 28, 29, 30]. However, in the context of complex instructions, responses are readily accessible even without deliberate reasoning. Therefore, there exists no enforced association between the emergence of long, deep reasoning and the improved responses. In this case, we implement an adaptive replay buffer to enforce superior CoT at the sample level. We introduce the $\pi_{\theta_{\text{old}}}$ for providing the sample-wise contrast between the responses with and without reasoning. The output without the essential CoT $\hat{y}^i$ (i.e., `<think>\n\n</think>`) receives its accuracy reward $\hat{R}_{\text{accuracy}}^i$. We filter out $x$ when all its rollouts $\{y^i\}_{i=1}^G$ are inferior to those reasoning-free counterparts $\{\hat{y}^i\}_{i=1}^G$:

$$\begin{aligned} \hat{\mathcal{J}}_{\text{GRPO}}^i = & \mathbb{E}_{x \sim D, \{y^i\}_{i=1}^G \sim \pi_{\theta_{old}}(\cdot|x), \{\hat{y}^i\}_{i=1}^G \sim \pi_{\theta_{old}}(\cdot|x, \text{CoT}(\hat{y}^i) = \varnothing)} \\ & \mathbb{1}(\max(\{R_{\text{accuracy}}^i\}_{i=1}^G) \geq \min(\{\hat{R}_{\text{accuracy}}^i\}_{i=1}^G) \cdot \mathcal{J}_{\text{GRPO}}^i. \end{aligned} \tag{4}$$

We evaluate whether the $y^i$ benefits from the reasoning by comparing its reward $R_{\text{accuracy}}^i$ with respect to $\hat{R}_{\text{accuracy}}^i$. If all the responses are judged worse than the vanilla output, it implies that the reasoning capacity of the policy model fails to meet the standard (e.g., constraints ignorance or mis-interpretation). In this case, the sample is too challenging to foster proper reasoning and can be simply skipped until that at least one rollout designates the paradigm leading to superior reward.

**Policy Distribution Drift Control with Behavior Cloning**   Another fundamental difference between maths problems and complex instructions is that the former merely emphasizes the correctness of final answers while the latter also assesses responses in terms of semantics. During the rollout sampling, responses that meet more constraints are prioritized even at the expense of coherence,

fluency, idiomaticity, and clarity. Such semantic-level degradation may not be easily resolved due to the constraints imposed on the instructions, as the compliant responses inherently differ from the pretraining texts. In light of this statement, a challenge arises from the excessive policy distribution drift where the catastrophic forgetting of the initially acquired knowledge from $\pi_{\text{ref}}$ occurs. We propose to explicitly perform behavior cloning of expert response $\tilde{y}$ under $x$.

$$\mathcal{J}_{\text{SFT}} = \mathbb{E}_{x \sim D}\left[ -\log \pi_\theta(\tilde{y}|x) \right]. \tag{5}$$

Compared with the KL-penalty term (Eq. 8), the behavior cloning by SFT explicitly constrain the $\pi_\theta$ for semantic alignment. It guarantees that: 1) the adherence to the expected format can be expedited for successful parsing of answers and reward computation at an early stage; 2) the organization of deep reasoning can be imitated and traced even with models of incompetent instruction following basis; 3) the potential reward hacking (e.g., responses that satisfy constraints but exhibit poor semantics) can be mitigated without relying on a well-trained reward model for scalar scores.

## 4 Experiments

### 4.1 Experimental Setup

**Dataset**  Statistics about our self-evolving dataset can be found in Sec. A.4. We also incorporated DeepScaleR [29] (see Sec. A.5). The evaluation metrics for each benchmark are reported in Sec. A.3.2.

**Baselines**  We compared with: 1) I/O: direct input with $x_{\text{CoT}} = \varnothing$; 2) CoT: reasoning prompting [17] to first deliver the thought and then the answer; 3) SDC: self-DeepClaude [92, 93] technique that first prompts for the thought and then packs the original input with the thought as a new context for the answer (see Sec. A.6.3); 4) SFT: supervised fine-tuning for learning the aligned responses.

**Implementation Details**  We use OpenRLHF [94] for both cold-start (Qwen2.5-1.5B/7B [95], LLaMA3.1-8B [2], and Ministral-8B [96]) and warm-start (DeepSeek-Qwen1.5B/7B [19] and DeepScaleR-1.5B [29]) experiments. Detailed settings can be found in Sec. A.6.

Table 1: Performance on seven instruction benchmarks. Best/2nd best are marked **bold**/underlined.

| Model | Method | IFEval | CELLO | CF Bench | Complex Bench | FB Bench | Follow Bench | Info Bench | Avg. |
|---|---|---|---|---|---|---|---|---|---|
| Qwen2.5-1.5B-Instruct | I/O | 45.28 | 71.00 | 36.00 | 50.97 | 39.81 | 40.00 | 71.24 | 50.61 |
| Qwen2.5-1.5B-Instruct | CoT | 28.65 | 59.30 | 22.00 | 32.94 | 37.31 | 29.28 | 62.22 | 38.81(-11.79%) |
| Qwen2.5-1.5B-Instruct | SDC | 41.95 | 66.10 | 30.00 | 41.70 | 36.52 | 37.39 | 67.55 | 45.89(-4.71%) |
| Qwen2.5-1.5B-Instruct | SFT | 65.61 | 71.20 | 48.00 | 57.46 | 42.75 | 56.47 | 76.22 | 59.67(+9.06%) |
| Qwen2.5-1.5B-Instruct | Ours | 44.91 | 73.50 | 53.66 | 63.92 | 58.67 | 59.82 | 81.95 | 62.35(+11.74%) |
| DeepSeek-Qwen1.5B | I/O† | 36.04 | 62.50 | 27.99 | 39.89 | 34.51 | 20.29 | 52.00 | 39.03 |
| DeepSeek-Qwen1.5B | SFT | 45.29 | 63.20 | 25.33 | 35.53 | 37.59 | 22.18 | 51.96 | 40.15(+1.12%) |
| DeepSeek-Qwen1.5B | Ours | 57.67 | 69.00 | 40.00 | 44.38 | 37.78 | 37.79 | 60.48 | 49.58(+10.54%) |
| DeepScaleR-1.5B | I/O† | 41.77 | 65.00 | 30.00 | 40.70 | 40.24 | 26.01 | 60.31 | 43.43 |
| DeepScaleR-1.5B | SFT | 48.24 | 62.90 | 28.00 | 36.68 | 35.72 | 26.50 | 54.22 | 41.75(-1.67%) |
| DeepScaleR-1.5B | Ours | 55.63 | 67.30 | 39.33 | 43.23 | 37.81 | 36.80 | 60.08 | 48.60(+5.17%) |
| Qwen2.5-7B-Instruct | I/O | 72.82 | 76.50 | 64.33 | 74.47 | 59.29 | 75.03 | 85.60 | 72.58 |
| Qwen2.5-7B-Instruct | CoT | 69.50 | 75.20 | 61.66 | 72.00 | 42.65 | 74.86 | 82.13 | 68.28(-4.29%) |
| Qwen2.5-7B-Instruct | SDC | 60.44 | 72.60 | **65.66** | 76.53 | 60.07 | **76.09** | 86.88 | 71.18(-1.39%) |
| Qwen2.5-7B-Instruct | SFT | 72.45 | 77.50 | 63.33 | 74.23 | 58.76 | 75.92 | 84.31 | 72.36(-0.21%) |
| Qwen2.5-7B-Instruct | Ours | 70.06 | **79.20** | 65.00 | **77.40** | **64.45** | 75.32 | 82.67 | **73.44**(+0.85%) |
| LLaMA3.1-8B-Instruct | I/O | 77.63 | 75.20 | 56.99 | 69.11 | 46.92 | 53.52 | 71.52 | 67.01 |
| LLaMA3.1-8B-Instruct | CoT | 60.44 | 65.50 | 47.66 | 56.54 | 32.34 | 37.36 | 58.48 | 54.53(-12.48%) |
| LLaMA3.1-8B-Instruct | SDC | **80.22** | 71.00 | 58.33 | 68.73 | 38.36 | 48.92 | 72.89 | 65.24(-1.77%) |
| LLaMA3.1-8B-Instruct | SFT | 77.26 | 75.80 | 54.00 | 65.24 | 40.16 | 59.56 | 65.30 | 64.92(-2.09%) |
| LLaMA3.1-8B-Instruct | Ours | 13.49 | 4.6 | 1.33 | 2.71 | 7.14 | 1.08 | 0.51 | 4.06(-62.95%) |
| Ministral-8B-Instruct | I/O | 59.51 | 76.20 | 62.33 | 70.03 | 54.54 | 73.49 | 84.00 | 68.58 |
| Ministral-8B-Instruct | CoT | 48.79 | 61.90 | 49.66 | 61.31 | 39.17 | 61.75 | 79.73 | 57.47(-11.11%) |
| Ministral-8B-Instruct | SDC | 58.59 | 63.60 | 56.99 | 68.32 | 48.06 | 69.37 | 84.08 | 64.14(-4.43%) |
| Ministral-8B-Instruct | SFT | 68.57 | 66.30 | 48.66 | 67.20 | 37.26 | 54.37 | 76.62 | 59.85(-8.72%) |
| Ministral-8B-Instruct | Ours | 72.64 | 72.6 | 59.33 | 70.45 | 54.35 | 76.08 | 75.33 | 68.68(+0.10%) |
| DeepSeek-Qwen7B | I/O† | 60.81 | 72.39 | 57.99 | 66.86 | 59.59 | 62.80 | 79.64 | 65.73 |
| DeepSeek-Qwen7B | SFT | 67.09 | 69.10 | 58.66 | 58.42 | 55.60 | 65.96 | 79.15 | 64.85(-0.88%) |
| DeepSeek-Qwen7B | Ours | 71.35 | 71.40 | 58.67 | 62.04 | 59.65 | 59.38 | 82.00 | 66.35(+0.62%) |

† The default outputs of reasoning models by I/O prompting contain both the thought and the answer parts.

### 4.2 Main Results

**Comparison with the Baselines**  Our method effectively boosts most of the existing LLMs in handling complex instructions (see Table 1), demonstrating the generalization of the cultivated deep

reasoning. In contrast, the CoT prompting causes a drastic performance decline to all models, which further confirms the detrimental effect of shallow thinking. Instead of adopting an one-off generation, SDC decouples the reasoning and answering via two-step inference. Due to the intrinsic superficial nature, SDC still fails to improve the reasoning quality. The SFT technique directly performs knowledge distillation where small LLMs mimic the reasoning patterns of strong slow-thinker. It guarantees that the depth and breadth of thinking is under immediate supervision. However, one drawback of SFT is that the model's generalization ability tends to deteriorate for samples that fall outside the domains encountered during training. Comparatively, our RL paradigm teaches LLMs how to think, driving the self-development of varied reasoning rather than simple memorization.

In line with Fig. 3, small models (1.5B) achieve much more gains than larger ones, showcasing the potentials of small LLMs via test-time scaling. The DeepSeek-distilled LLMs possess a good starting point for reasoning organization from their warm-start imitation across a broad range of tasks and topics (Fig. 36). With respect to model family, we unfortunately find that the capacity of Ministral and LLaMA is inferior to that of Qwen. The Ministral-8B exhibits limited advantages over its vanilla counterpart while the LLaMA3.1-8B experienced a model collapse during training. As shown in Fig. 35(a) and (d), a rapid shrinkage of response and a frequent surge of KL penalty imply a great deviation of LLaMA from its initial state. The reason behind might be ascribed to the pre-trained knowledge of base models [97]. LLaMA tends to generate endless thinking without conforming to the required format. It then struggles to output semantically consistent responses and keeps extending its meaningless thinking until collapse. Detailed results can be found in Sec. A.8.1. Discussions on generalization over multi-purpose benchmarks can be found in Sec. A.8.2.

Table 2: Performance on ComplexBench (Qwen2.5-7B-Instruct). Best/2nd best are marked **bold**/underlined. OD, SC, CNFR, FC, and SR stand for the Oracle Decomposition [11], Self-Consistency [98], Conifer [14], FollowComplex [13], and Self-Refine [99].

| Category | ND | I/O | OD | SC | CNFR | FC | SR | Ours |
|---|---|---|---|---|---|---|---|---|
| **And** | 1 | 85.85 | 84.27 | 84.03 | 75.10 | 84.77 | 85.66 | **86.57** |
| **Chain** | 1 | 72.18 | 74.68 | 73.54 | 60.95 | 66.27 | **75.25** | 73.96 |
| | 2 | 70.56 | 72.70 | 69.63 | 64.43 | 70.66 | 73.07 | **76.88** |
| **Avg.** | - | 70.96 | 73.18 | 70.57 | 63.59 | 69.60 | 73.59 | **76.18** |
| **Selection** | 1 | 77.25 | 76.61 | 72.08 | 60.52 | 71.67 | 69.61 | 73.39 |
| | 2 | 65.61 | 71.83 | 68.23 | 53.25 | 61.96 | 64.34 | **72.92** |
| | 3 | 63.39 | 68.45 | 56.13 | 46.04 | 51.70 | 58.67 | 60.75 |
| **Avg.** | - | 65.67 | 70.49 | 65.83 | 51.92 | 60.92 | 62.69 | 69.16 |
| **Selection & Chain** | 2 | 65.64 | 65.94 | 60.81 | 47.33 | 61.07 | 52.01 | 61.06 |
| | 3 | 59.70 | 65.77 | 64.08 | 48.53 | 57.65 | 60.41 | 65.00 |
| **Avg.** | - | 62.68 | 65.85 | 62.44 | 47.93 | 59.36 | 56.20 | 63.03 |
| **Overall** | - | 74.47 | 76.26 | 73.76 | 63.51 | 71.97 | 74.00 | **77.40** |

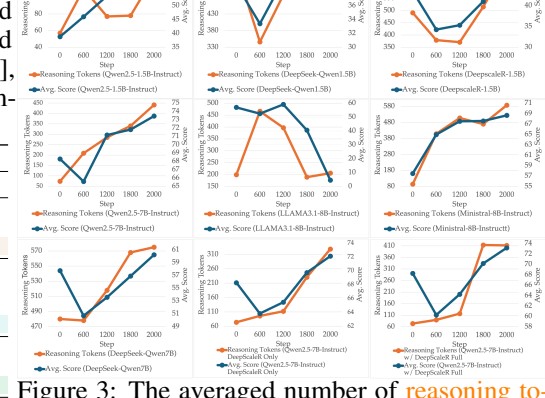

Figure 3: The averaged number of reasoning tokens and scores over steps (best viewed magnified).

Table 3: Training and testing-time compute on ComplexBench (Qwen2.5-7B-Instruct).

| Method | Training Compute | | | | | Test-Time Compute | | | δ |
|---|---|---|---|---|---|---|---|---|---|
| | # SFT samples | # SFT epochs | RLHF algo. | # RLHF samples | # RLHF epochs | Avg. # steps or sampling | Avg. # reasoning tokens | Avg. # answer tokens | Avg. gains over I/O |
| I/O | – | – | – | – | – | 1 | – | 332 | 74.47 |
| CoT | – | – | – | – | – | 1 | 80 | 419 | -2.46 |
| SDC | – | – | – | – | – | 1 | 79 | 372 | +2.06 |
| SFT | 13K | 10 | – | – | – | 1 | – | 361 | -0.23 |
| OD [11] | – | – | – | – | – | 1 | – | 407 | +1.79 |
| SC [98] | – | – | – | – | – | 10 | – | 3302 | -0.70 |
| SR [99] | – | – | – | – | – | 1.9 | – | 877 | -0.46 |
| CNFR [14] | 66K | 4 | DPO | 63K | 1 | 1 | – | 308 | -10.96 |
| FC [13] | 12K | 2 | DPO | 12K | 2 | 1 | – | 261 | -2.50 |
| Ours | – | – | GRPO | 26K | 3 | 1 | 299 | 349 | **+2.93** |

**Comparison with the SOTAs** We implemented SOTAs on the ComplexBench (see Table 2): Oracle Decomposition [11] (ground-truth decomposition of sub-instructions), Self-Consistency [98] (majority voting@10), Conifer [14], FollowComplex [13], and Self-Refine [99]. We also report the training and testing-time compute in Table 3. Our method demonstrate its superiority on the most complicated *Chain*, *Selection* categories, suggesting that the reasoning indeed assists analysis of LLMs to carry out the truly relevant, valid request with constraints.

**Variation of Reasoning Patterns** The change of step-by-step keywords such as `first`, `second`, `next`, and `finally` (see Fig. 4) shows that all LLMs enjoy an increase of tokens on challenging

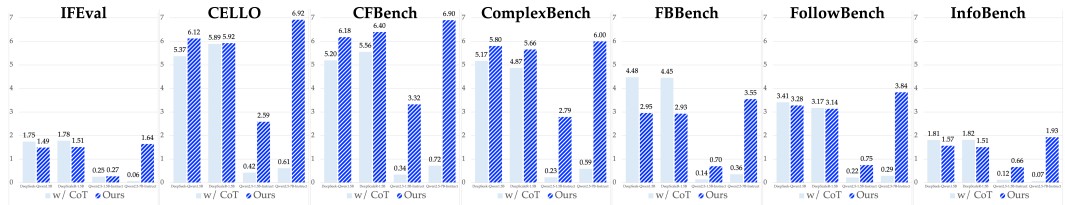

Figure 4: The averaged frequency change of keyword tokens of DeepSeek-Qwen1.5B, DeepScaleR-1.5B, Qwen2.5-1.5B-Instruct, and Qwen2.5-7B-Instruct before/after RL (best viewed magnified).

Table 4: Ablation study on the Qwen2.5-7B-Instruct with CoT reasoning. Best/2nd best are marked **bold**/underlined. Maths and Complex refer to the DeepScaleR and our self-evolved dataset, respectively. SupCoT and BC denote the superior CoT enforcement and behavior cloning, respectively.

| Data | | Method | | Benchmarks (Inference w/ CoT) | | | | | | | |
|---|---|---|---|---|---|---|---|---|---|---|---|
| Maths | Complex | SupCoT | BC | IFEval | CELLO | CF Bench | Complex Bench | FB Bench | Follow Bench | Info Bench | Avg. |
| - | - | - | - | 69.50 | 75.20 | 61.67 | 72.00 | 42.65 | 74.87 | 82.13 | 68.28 |
| ✓ | - | - | - | 72.08 | 76.80 | 63.66 | 73.81 | 57.93 | 74.83 | 85.73 | 72.12(+3.84%) |
| - | ✓ | ✓ | ✓ | 72.27 | 73.90 | 53.33 | 63.89 | 45.04 | 64.04 | 81.24 | 64.81(−3.46%) |
| 0.2 | 1 | ✓ | ✓ | 67.10 | 74.70 | 63.00 | 75.70 | 60.39 | 70.88 | 84.31 | 70.87(+2.59%) |
| 1 | 1 | ✓ | ✓ | 70.06 | 79.20 | 65.00 | 77.40 | 64.45 | 75.32 | 82.67 | 73.44(+5.16%) |
| 5 | 1 | ✓ | ✓ | 72.83 | 78.80 | 69.67 | 78.54 | 46.41 | 79.87 | 86.18 | 73.18(+4.90%) |
| 1 | 1 | - | - | 63.58 | 76.90 | 47.00 | 76.34 | 57.63 | 65.74 | 87.95 | 67.87(-0.40%) |
| 1 | 1 | ✓ | - | 66.72 | 78.10 | 65.00 | 75.62 | 56.42 | 76.12 | 80.13 | 71.15(+2.87%) |
| 1 | 1 | - | ✓ | 70.05 | 79.20 | 65.00 | 75.68 | 56.47 | 75.31 | 82.66 | 72.05(+3.76%) |

benchmarks such as CFBench and ComplexBench, confirming the importance of our cultivated deep reasoning. For instructions without intricate compositions (e.g., *And*-constraints in IFEval), slow-thinking LLMs reduced their keyword frequency a bit due to the shortened response length.

## 4.3 Ablation Study and Analysis

**Effect of Maths Problems**   Both Tables 1 and 4 corroborate the positive roles of DeepScaleR in developing reasoning. The increment of maths problems is positively associated with the growth of CoT tokens and thereafter the performance improvement (see Fig. 37), implying that the mathematic reasoning is crucial and supplementary to the general-purpose reasoning. However, given the same training steps, the model with full maths did not converge yet, suggesting that more iterations are required towards optimum. Discussions on maths generalization are in Sec. A.8.3.

**Effect of Superior CoT Enforcement**   As shown in Fig. 5, the ratio of the samples kept with superior CoT dropped first and then improves steadily. It implies that the transition from shallow to deep reasoning is promoted during training, leading to responses with higher rewards with respect to those without deliberate reasoning. The filtering of experience replay buffer for superior CoT has the following benefits: 1) In the early stage, the fast-thinking LLMs are struggling to establish the expected CoT formats and therefore the ultimate responses often contain incompatible HTML-style elements (reason/answer tag tokens). The incorporation of LLM completion with the prefix of empty reasoning makes format compliance easier. 2) It removes the samples from participating in training where

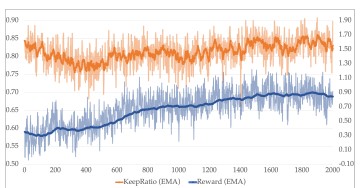

Figure 5: The ratio of samples kept by superior CoT and the reward over steps of Qwen2.5-7B-Instruct.

their shallow reasoning brings detriments to the ultimate answers. Such isolation prevents the flawed thinking process from receiving biased rewards for formalizing inferior, prone-to-hacking responses, which also allows the time lag for imitation of expert thinking (see Figs. 33 and 34).

**Effect of Behavior Cloning**   The immediate imitation of expert reasoning not only encourages fast-thinkers to earn format rewards but also stabilizes training and fills the gaps of rule-centric rewards. Without proper guidance, cold-start LLMs can only repeat their inherent shallow thinking and consistently receive negative feedback (as confirmed in Table 1). In this case, either model collapse or reward hacking is prone to occur and thereafter causes catastrophic distribution drift.

Table 5: Generalization on OOD constraints of IFBench.

| Model | Method | prompt-level strict | instruction-level strict | prompt-level loose | instruction-level loose | Avg. |
|---|---|---|---|---|---|---|
| Qwen2.5-1.5B-Instruct | I/O | 15.64 | 17.01 | 18.36 | 20.29 | 17.82 |
| Qwen2.5-1.5B | CoT | 13.60 | 15.52 | 15.98 | 17.61 | 15.67(-2.14) |
| Qwen2.5-1.5B | SDC | 15.98 | 17.61 | 17.68 | 19.70 | 17.74(-0.08) |
| Qwen2.5-1.5B | SFT | 16.32 | 17.61 | 19.38 | 20.89 | 18.55(+0.73) |
| Qwen2.5-1.5B-Instruct | Ours | 17.68 | 19.4 | 20.06 | 22.68 | 19.95(+2.13%) |
| DeepSeek-Qwen-1.5B | I/O | 8.80 | 11.64 | 12.92 | 15.82 | 12.29 |
| DeepSeek-Qwen-1.5B | SFT | 12.13 | 13.69 | 14.16 | 15.98 | 13.99(+1.69) |
| DeepSeek-Qwen-1.5B | Ours | 12.92 | 14.02 | 15.64 | 16.71 | 14.82(+2.53%) |
| DeepScaleR-1.5B | I/O | 11.22 | 12.23 | 15.3 | 17.91 | 14.16 |
| DeepScaleR-1.5B | SFT | 12.71 | 13.94 | 15.02 | 16.74 | 14.60(+0.44) |
| DeepScaleR-1.5B | Ours | 12.58 | 14.02 | 17.00 | 18.80 | 15.60(+1.44%) |
| Qwen2.5-7B-Instruct | I/O | 28.23 | 29.85 | 31.63 | 33.43 | 30.78 |
| Qwen2.5-7B-Instruct | CoT | 27.89 | 30.44 | 30.95 | 33.73 | 30.75(-0.03) |
| Qwen2.5-7B-Instruct | SDC | 26.87 | 29.25 | 32.31 | 34.62 | 30.76(-0.02) |
| Qwen2.5-7B-Instruct | SFT | 23.12 | 26.57 | 28.57 | 32.54 | 27.70(-3.08) |
| Qwen2.5-7B-Instruct | Ours | 29.82 | 30.76 | 32.27 | 35.43 | 32.07(+1.29%) |
| Ministral-8B-Instruct | I/O | 16.66 | 17.31 | 23.12 | 24.47 | 20.39 |
| Ministral-8B-Instruct | CoT | 15.30 | 14.92 | 29.59 | 31.34 | 22.78(+2.39) |
| Ministral-8B-Instruct | SDC | 18.36 | 18.80 | 23.80 | 24.77 | 21.43(+1.04) |
| Ministral-8B-Instruct | SFT | 12.24 | 13.73 | 16.32 | 19.4 | 15.42(-4.96) |
| Ministral-8B-Instruct | Ours | 20.74 | 23.88 | 28.23 | 31.94 | 26.19(+5.77%) |
| DeepSeek-Qwen7B | I/O | 13.6 | 14.62 | 19.72 | 22.08 | 17.50 |
| DeepSeek-Qwen7B | SFT | 17.34 | 18.80 | 21.08 | 22.68 | 19.97(+2.47) |
| DeepSeek-Qwen7B | Ours | 20.06 | 22.38 | 25.17 | 27.46 | 23.77(+6.27%) |

## 4.4 Generalization Study on OOD Constraints

To test if the proposed RAIF can generally improve the performance of LLMs on unseen constraints, we follow Tülu 3 [16] to conduct experiments on brand-new complex instructions that differ from existing benchmarks. Specifically, we use the most recently proposed IFBench [100] as OOD evaluation benchmark because it contains 58 new, diverse, and challenging constraints. There exists no possibility of data contamination with respect to the training and testing data of the present study.

As shown in Table 5, RAIF consistently improves performance across model sizes and families, demonstrating strong generalizability to new constraints. Notably, unlike Tülu 3 [16], which reports overfitting to IFEval and degraded IFBench results after RLVR, our RAIF does not overfit IFEval's constraints. This generalization stems from two factors: (1) diverse, complex instruction evolution with varied constraints and structures, and (2) the application of deep reasoning, which aids instruction analysis and decomposition. Comparing RLVR to baselines: 1) **CoT**: For most instructed (non-reasoning) models, vanilla CoT decreases performance on IFBench, aligning with Table 1. These models do not fully analyze instruction constraints, instead summarizing them superficially. 2) **SDC**: By decoupling thinking and execution, SDC allows models to revisit instructions, improving over CoT but still limited by imperfect reasoning. 3) **SFT**: Reasoning models slightly benefit by distilling patterns from stronger models (e.g., DeepSeek R1), but this does not transfer to instructed models due to mismatched reasoning pattern distributions. 4) **Scaling**: As DeepSeek R1 [19] shows, SFT requires around 800K curated samples to upgrade instructed models. In contrast, RLVR needs only 13K samples to stimulate self-developed reasoning, yielding significant gains.

## 5 Conclusion

We propose a systematic method to incentivize reasoning of LLMs for solving complex instructions. We first address the data scarcity by developing a scalable pipeline for constructing instructions and their verifications. Then, we target at the superficial reasoning of fast-thinkers via effective RL with rule-centric rewards. We pay attention to the fundamental differences of pathway dependency between maths problems and complex instructions, and propose to enforce superior CoT with behavior cloning. Extensive experiments on seven benchmarks confirm our effectiveness, providing valuable insights and guidelines for practitioners to build slow-thinkers under various compelx tasks.

**Broader Impact** Our recipe of scaling up data and RL would benefit the stimulation of reasoning for tasks beyond maths problems. Moreover, our studies shed light on the studies on cognitive behaviors of LLMs, which in turn facilitates the development of stronger base models.

**Funding** No external funding was received.

**Competing Interests** The authors declare no competing interests.

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

# A Appendix

In this appendix, we first provide the descriptions about the mathematic symbols used in the manuscript. Then, provide the detailed descriptions about the benchmarks used in our experiments and their evaluation metrics. Furthermore, we provide the details about the preparation of datasets used for training, including the publicly available DeepScaleR dataset [29] and the self-evolved complex instruction dataset mentioned in Sec. 3.2. The detailed implementations are presented in Sec. A.6 of the appendix. Finally, we present more fine-grained results for in-depth analyses.

## A.1 Symbol Description

To enhance clarity, a detailed description of mathematic symbols is provided in Table 6.

Table 6: Descriptions of the symbols used in the paper.

| Symbol | Definition |
|---|---|
| $x$ | A query input of the complex instruction (tokenized) |
| $y$ | A response of a LLM to the input $x$ (tokenized) |
| $\theta$ | The parameters of the LLM to be optimized |
| $\pi_\theta$ | The policy model (LLM) to be optimized (parameterized as $\theta$) |
| $y_{1:t-1}$ | The tokens of $y$ starting from the index 1 to the index $t-1$ |
| $y_t$ | The $t$-th token of $y$ |
| $\pi_\theta(y_t\|x, y_{1:t-1})$ | The language modeling process of $\pi_\theta$ given the input $x$ and the preceding response $y_{1:t-1}$ |
| $\mathrm{CoT}(y)$ | The CoT reasoning tokens of $y$ |
| $y\backslash\mathrm{CoT}(y)$ | The response answer tokens of $y$ |
| $x_{\mathrm{CoT}}$ | The CoT prompting tokens of $x$ |
| $\|y^i\|$ | The number of tokens of $y^i$ |
| $G$ | The number of generations per input prompt during rollout |
| $\pi_{\theta_{\mathrm{old}}}$ | The policy model from the previous iteration state |
| $\pi_{\mathrm{ref}}$ | The reference model |
| $y^i$ | The $i$-th generated output from the policy model $\pi_{\theta_{\mathrm{old}}}$ for the input prompt $x$ |
| $\mathcal{J}_{\mathrm{GRPO}}$ | The GRPO loss |
| $\mathcal{J}_{\mathrm{GRPO}}^i$ | The GRPO loss for the generated sample $y^i$ |
| $r_t^i$ | The importance sampling ratio of $y^i$ at the time $t$ (i.e., for its generation of the $t$-th token) |
| $R^i$ | The reward of $y^i$ before normalization |
| $\hat{A}_t^i$ | The estimated advantage of $y^i$ at time $t$ |
| $D_{\mathrm{KL}}^i(\pi_\theta\|\|\pi_{\mathrm{ref}})$ | The KL divergence between the current policy model and the reference model on $y^i$ |
| $\mathrm{clip}(\cdot, l, u)$ | The clipping operation with the lower bound $l$ and the upper bound $u$ |
| $R_{\mathrm{format}}^i$ | The format reward of $y^i$ |
| $C$ | The number of atomic constraints in the input prompt $x$ |
| $c_j$ | The tokens of the $j$-th atomic constraint in $x$ with $c_j \subset x$ |
| $x_C$ | The set of all the $C$ atomic constraints with $x_C \subset x$ and $c_j \in x_C, \forall j$ |
| $x_C^{\mathrm{active}}$ | The subset that contains only the truly valid, active constraints with $x_C^{\mathrm{active}} \subset x_C$ |
| $\mathrm{is\_followed}(y^i\|c_j)$ | The condition of whether $y^i$ satisfies the constraint $c_j$ as instruction following |
| $r_\phi$ | The reward model parameterized by $\phi$ |
| $R_{\mathrm{accuracy}}^i$ | The accuracy reward of $y^i$ |
| $\hat{y}^i$ | The $y^i$ with empty reasoning tokens (i.e., skipping CoT via `<think>\n\n</think>`) |
| $\hat{R}^i$ | The reward of $\hat{y}^i$ |
| $\hat{R}_{\mathrm{format}}^i$ | The format reward of $\hat{y}^i$ |
| $\hat{R}_{\mathrm{accuracy}}^i$ | The accuracy reward of $\hat{y}^i$ |
| $\hat{\mathcal{J}}_{\mathrm{GRPO}}^i$ | The filtered GRPO loss with superior CoT enforcement |
| $\tilde{y}$ | The expert response to the input $x$ with both the reasoning and the answer |
| $\mathcal{J}_{\mathrm{SFT}}$ | The SFT loss for behavior cloning |

## A.2 Preliminaries

In this section, we provide the introduction to the Group Relative Policy Optimization (GRPO) algorithm [87]. Given a query $x$ sampled from the distribution $D$ of complex instructions, the policy model $\pi_{\theta_{\mathrm{old}}}$ from the previous iteration generates a group of $G$ individual outputs $\{y^i\}_{i=1}^G$. GRPO

updates the policy $\pi_\theta$ by maximizing the objective:

$$\mathcal{J}_{\text{GRPO}} = \mathbb{E}_{x \sim D, \{y^i\}_{i=1}^G \sim \pi_{\theta_{old}}(\cdot|x)} \frac{1}{G} \sum_{i=1}^{G} \mathcal{J}_{\text{GRPO}}^i,$$

$$\mathcal{J}_{\text{GRPO}}^i = \left[ \frac{1}{|y^i|} \sum_{t=1}^{|y^i|} (\min(r_t^i(\theta)\hat{A}_t^i, \text{clip}(r_t^i(\theta), 1-\epsilon, 1+\epsilon)\hat{A}_t^i) - \beta D_{\text{KL}}^i(\pi_\theta||\pi_{\text{ref}}) \right], \quad (6)$$

$$r_t^i = \frac{\pi_\theta(y_t^i|x, y_{1:t-1}^i)}{\pi_{\theta_{\text{old}}}(y_t^i|x, y_{1:t-1}^i)}, \hat{A}_t^i = \frac{r^i - \text{mean}(\{R^i\}_{i=1}^G)}{\text{std}(\{R^i\}_{i=1}^G)}, \quad (7)$$

$$D_{\text{KL}}^i(\pi_\theta||\pi_{\text{ref}}) = \frac{\pi_{\text{ref}}(y_t^i|x, y_{1:t-1}^i)}{\pi_\theta(y_t^i|x, y_{1:t-1}^i)} - \log \frac{\pi_{\text{ref}}(y_t^i|x, y_{1:t-1}^i)}{\pi_\theta(y_t^i|x, y_{1:t-1}^i)} - 1. \quad (8)$$

Compared with the proximal policy optimization (PPO) [101], the advantage of the $i$-th output response is computed by normalizing the group-level rewards $\{R^i\}_{i=1}^G$. PPO optimizes an additional critic model as value function via the generalized advantage estimation (GAE) [102]. In consideration of the simplicity and our available computing resources, we employ the GRPO as our scale-up RL settings.

### A.3 Evaluation Datasets and Metrics

In the present study, we evaluate the proposed method on the following datasets: IFEval [7], ComplexBench [11], CELLO [10], CFBench [8], FB-Bench [53], FollowBench [54], and InfoBench [55]. All these benchmarks are specifically constructed to evaluate the instruction-following capabilities of LLMs under various complex tasks and domains (see Table 7).

#### A.3.1 Benchmarks

**IFEval** IFEval [7] is one commonly used benchmark to evaluate the abilities of LLMs in following natural language instructions. It focuses on a wide range of "verifiable instructions" that can be efficiently and accurately validated via python codes. It identifies 25 types of atomic constraints including keywords, languages, length constraints, detectable formats, combinations, case changes, starting and ending phrases, and punctuations. All the verifications are performed via codes to avoid potential bias caused by LLM judges. Therefore, their constraints are limited in types where no semantic ones are covered.

**CELLO** CELLO [10] formulates complex instructions from real-world task descriptions and input queries. From the task perspective, CELLO considers multi-tasking, semantic constraints, format constrains, and quality constraints. From the input text perspective, CELLO considers heterogeneous information, long context, noisy information, and multi-turn conversations. Similar to the IFEval benchmark, CELLO focuses simply on the objective, rule-verifiable constraints where their verifications are all rule-based.

**CFBench** CFBench [8] covers more thant 200 real-life scenarios and over 50 tasks. Their detailed types of constraints are classified into 10 primary categories and 25 sub-categories, including content constraints, numerical constraints, stylistic constraints, format constraints, linguistic constraints, situation constraints, example constraints, inverse constraints, contradictory constraints, and rule constraints.

**ComplexBench** ComplexBench [11] is one of the most comprehensive benchmarks that validate the instruction-following capabilities of LLMs under complex isntructions. Its hierarchical taxnomy for the definitions of complex instructions include 4 constraint types, 19 constraint dimension, and 4 composition types. It covers tasks that highlight fundamental language ability, advanced Chinese understanding, open-ended questions, practical writing, creative writing, professional writing, custom writing, logical reasoning, task-oriented role play, and profession knowledge.

**FB-Bench** FB-Bench [53] is a multi-task fine-grained benchmark that evaluates the LLM's responsiveness to human feedbacks under real-world scenarios. It encompasses eight task types (reasoning, coding, text extraction,text error correction, text creation, knowledge QA, and text translation), five deficiency types of responses (not-following-instructions, logical error, incomplete answers, factual errors, and unprofessional answers), and nine feedback types (pointing out errors, clarifying the intent, raising objections, detailed explanations, hinting guidance, simple questioning, misinformation, credibility support, and unreasonable requests).

**FollowBench** FollowBench [54] is a multi-level fine-grained constraints following benchmark. It covers constraints of content, situation, style, format, example, and mixed types. The levels of constraints refer to the number of atomic constraints present in each instruction, where the way of constraint composition is designed for each task in its category.

**InfoBench** InfoBench [55] develops both the hard set and the easy set covering 72 domains including natural sciences, social sciences, engineering, economics, engineering, economics, and arts. It incorporates specific response constraints including contents, linguistic guidelines, style rules, format specifications, and number limitations. For each instruction, its decomposed scoring questions are prepared for boolean evaluation of constraint-following conditions.

Table 7: Statistics of the complex instruction benchmarks used for evaluation.

| Benchmark | Size | Constraint Taxonomy | Composition Type | | | | Verification and Evaluation | | |
| | | | *And* | *Chain* | *Selection* | *Nested* | Code-Execution | LLM-as-a-Judge | Aggregation |
|---|---|---|---|---|---|---|---|---|---|
| IFEval [7] | 541 | 25 | ✓ | - | - | - | ✓ | - | Average |
| CELLO [10] | 523 | 4 | ✓ | ✓ | - | - | ✓ | - | Average |
| CFBench [8] | 1,000 | 25 | ✓ | ✓ | - | - | - | ✓ | Priority |
| ComplexBench [11] | 1,150 | 19 | ✓ | ✓ | ✓ | ✓ | ✓ | ✓ | Dependency |
| FB-Bench [53] | 591 | 9 | ✓ | ✓ | - | - | - | ✓ | Weighted Average |
| FollowBench [54] | 820 | 5 | ✓ | - | - | - | ✓ | ✓ | Average |
| InfoBench [55] | 500 | 5 | ✓ | - | - | - | - | ✓ | Average |

### A.3.2 Evaluation Metrics

In the present study, we report the following evaluation metrics of benchmarks in Table 1.

- IFEval: *prompt-level_loose*
- CELLO: average of the *average_complex_ins* and *average_complex_input*
- CFBench: average of the *CSR*, *ISR*, and *PSR*
- ComplexBench: *overall_DRFR*
- FB-Bench: average of the *error_correction_score* and *response_maintenance_score*
- FollowBench: average of the *CSL_avg_en*/3.68, *HSR-level-1_en*, *HSR-level-2_en*, *HSR-level-3_en*, *HSR-level-4_en*, *HSR-level-5_en*, *SSR-level-1_en*, *SSR-level-2_en*, *SSR-level-3_en*, *SSR-level-4_en*, *SSR-level-5_en*, *CSL_avg_cn*/3.68, *HSR-level-1_cn*, *HSR-level-2_cn*, *HSR-level-3_cn*, *HSR-level-4_cn*, *HSR-level-5_cn*, *SSR-level-1_cn*, *SSR-level-2_cn*, *SSR-level-3_cn*, *SSR-level-4_cn*, *SSR-level-5_cn*
- InfoBench: *overal_DRFR*

Note that for FollowBench, the *CSL_avg_en* and *CSL_avg_cn* are normalized by the score of Qwen2.5-72B-Instruct (3.68) for percentage comparison. The detailed explanations on the metrics above are provided below.

**IFEval** IFEval [7] employs four accuracy scores for evaluation: 1) prompt-level strict accuracy, 2) instruction-level strict accuracy, 3) prompt-level loose accuracy, and 4) instruction-level loose accuracy. The prompt-level strict accuracy measures the percentage of prompts where all the constraints are satisfied. The instruction-level strict accuracy, on the other hand, reports the percentage of constraints satisfied across samples. For the loose evaluation metrics, they apply transformations on the responses of LLMs to remove the markdown syntax, opening intros to the final responses, and the ending outros following the final responses to reduce false negatives.

**CELLO**   CELLO [10] proposes a code-based verification system that checks four common fine-grained aspects like: 1) count limit, 2) answer format, 3) task-prescribed phrases, and 4) input-dependent query. For the count limit, it considers word count score, sentence count score, and sample count score. For the answer format, it considers parsability and keywords. For the input-dependent query, it scores whether the key phrases from the input query are existent in the responses, and applies a penalty term of COPY-BLEU to prevent giving high scores to the undesirable copying behavior of LLMs. For the task-prescribed phrases, it checks if the mandatory phrases specified in the task description are covered in the responses to satisfy those essential conditions.

**CFBench**   CFBench [8] employs the three metrics: constraint satisfaction rate (CSR), instruction satisfaction rate (ISR), and priority satisfaction rate (PSR). CSR measures whether each constraint is satisfied and computes the average of satisfaction rate across samples. ISR verifies whether all the constraints are satisfied per sample and report the average rate across samples. PSR assigns priority requirement to the constraints and if any prioritized constraint is not satisfied, the verification is judged as failure.

**ComplexBench**   ComplexBench [11] provides the scoring questions for each complex isntruction to check whether the constraints are satisfied. Each scoring question can only be answered with boolean responses as "YES" or "NO" and then the scores of each decomposed question are aggregated for the metric of decomposed requirements following ratio (DRFR). Note that the dependency of scoring questions is considered in aggregation. If any preceding constraint is not satisfied, the following constraints are simply judged as failure.

**FB-Bench**   FB-Bench [53] employs the DRFR metric as well and validates a series of criteria on the checklist for each complex instruction. It further set different weights for different criteria in the checklist for the weighted aggregation of the final score. The weights of different scoring questions are pre-defined to signify their importance.

**FollowBench**   FollowBench [54] employs both code-execution and LLM-as-a-Judge verifications for evaluation. It uses an incremental method of constraint validation where the LLM judge is asked to recognize the newly added constraint each time for verification. Three main metrics are used: 1) hard satisfaction rate (HSR), 2) soft satisfaction rate (SSR), and consistent satisfaction levels (CSL). The HSR and SSR can be understood as instruction-level and constraint-level satisfaction rate, respectively. The CSL measures how many consecutive levels are satisfied per instruction.

**InfoBench**   InfoBench [55] proposes the decomposed requirements following ratio (DRFR) metric for verifications. For each scoring question, the LLM-as-a-Judge strategy is used to give a "YES" or "NO" response. The final scores of the response are simply the averaged accumulation of scores over the total number of scoring questions. Such scoring technique enables a more fine-grained interpretation of the final results.

## A.4   Self-Evolved Complex Instruction Dataset

The self-evolved complex instruction dataset contains 13K complex instructions and their associated verifications (either with codes for python execution or with scoring questions for LLM-as-a-Judge) and reference expert responses. We choose the WildChat [35] and the Alpaca [88] as our instruction pool to perform self-evolving. The types of rules and constraints are sourced and collected from the taxonomy defined in IFEval, CELLO, ComplexBench, CFBench, FB-Bench, FollowBench, and InfoBench. Such preparation ensures the diversity of the rule and constraint pool. The type of composition follows the definition in the ComplexBench. The statistics of our self-evolved dataset can be found in Table 8. In consideration of the existing benchmarks, models, and the instruction pool, both the languages of Chinese and English are considered.

The detailed steps of seed instruction selection are described below.

### A.4.1   Tagging for Selection of Seed Instructions

For the selection of seed instructions, we choose the WildChat [35] (650K) and the Alpaca [88] (52K) as the pool of instructions. Especially for the WildChat dataset, a broad spectrum of user-chatbot interactions are covered including ambiguous user requests, code-switching, topic-switching,

Table 8: Statistics of our self-evolved complex instruction dataset.

| Language | # of Samples | # of Input (Tokens) | # of Output (Tokens) | # of Constraints per Sample | # of Scoring Questions per Sample (Verification by LLM-as-a-Judge) | # of Verifiable Restrictions (Verification by Code-Execution) |
|---|---|---|---|---|---|---|
| Chinese | 9.1K | 256±343 | 596±422 | 2.80±1.29 | 2.59±1.34 | 3.28±1.01 |
| English | 4.1K | 138±261 | 1013±651 | 4.00±1.95 | 4.45±2.17 | 3.21±1.13 |

and political discussions. Therefore, it is indispensable to filter out those instructions that are not appropriate for constructing complex instructions with additional constraints. In this case, we follow InsTag [89] to perform topic and capability tagging on the instructions. We refer to the "intention tags" of InsTag as the ability tags in the present study because those open-ended tags are closely associated with the abilities of LLMs to solve input queries. Specifically, we manually summarize and define the topic tags from the approximate 3.2K ability tags of the InsTag (see Fig. 6). Two typical examples of the instag tagging by TagLM are provided below respectively for Chinese and English instructions from the WildChat, respectively (see Fig. 7).

After tagging, we perform selection of seed instructions. We choose instructions that possess topic tags of "Problem Solving", "NLP and Understanding", "Info Processing", and "Creativity & Design" as candidates. Then, we perform random sampling on those instructions. During sampling, the ability tags are used to maximize the diversity of the chosen seed instructions (i.e., the number of instructions classified to each ability tag is controlled). Finally, we obtain 39K seed instructions in total.

---

**Summarized Topic Tags from the Ability Tags of InsTag**

Problem Solving
NLP and Understanding
Info Processing
Logic & Reasoning
Programming & Dev
Creativity & Design
Date Science
Math skills
Linguistics & Multiculturalism
Knowledge Q&A
Education & Consulting
Communication & Social Media
Humanities & Social Sciences
Research
Project Management
Literary & Artistic Knowledge
STEM
Finance & Business
Task Generation
Medical & Health
Life Skills
Legal Knowledge
Psychology
Task Completion
Law & Safety
Politics, Military Strategy & Security

Figure 6: Topic tags summarized from the ability tags of InsTag.

```
┌─────────────────────────────────────────────────────────────────────────┐
│ InsTag by TagLM on WildChat Examples                                      │
├─────────────────────────────────────────────────────────────────────────┤
│                                                                           │
│ {                                                                         │
│ "User Query": "Please paraphase following words to be more formal and     │
│ cohesive: Among a wide range of generation tasks, subject-driven          │
│ generation is a popular topic, which is also known as customization       │
│ or personalization. Such task refers to learning object-level             │
│ representations from a reference set and regenerate them in new contexts, │
│ depending on different conditions or prompts.",                           │
│ "InsTag Response": "{"Ability Tags":["paraphrase", "formal tone"],        │
│ "Topic Tags":["NLP and Understanding"]}",                                 │
│ }                                                                         │
│ {                                                                         │
│ "User Query": "介绍一下Gradio和docker",                                    │
│ "InsTag Response": "{"Ability Tags":["programming language",              │
│ "question answer", "problem-solving"],                                     │
│ "Topic Tags":["Problem Solving", "Programming & Dev", "Knowledge Q&A"]}", │
│ }                                                                         │
│                                                                           │
└─────────────────────────────────────────────────────────────────────────┘
```

Figure 7: Two tagged examples from the WildChat (Chinese and English).

### A.4.2 Generation of Rules and Constraints with Verifications

Given the seed instructions, we perform self-instruct [21] to generate instructions with additionally introduced rules and constraints. Specifically, we employ different techniques to evolve instructions with constraints that are verified by code-execution and LLM-as-a-Judge, respectively.

**Verifications via Code-execution** For the code-execution verifications, the constraint templates and their verification codes are prepared in advance. We follow existing studies [59, 13] to use the constraints defined in IFEval [7], and randomly choose atomic constraints from the entire pool of 25 constraints. Specifically, since certain constraints are mutually exclusive (i.e., not able to be fulfilled simultaneously), we refer to IFEval [7] to filter out impossible combinations. In total, there exist 25 atomic constraints with 210 combinations. Given the pre-defined constraint templates, we need to fill in the placeholders for instantiation. Note that most of the placeholders can be simply replaced with a randomly chosen candidate (e.g., number of words, response language, and format of the response). There still exists certain keyword placeholders (see Fig. 8) that are associated with the semantics of the seed instructions, which requires LLMs to generate reasonable candidates instead. The detailed prompts to generate keywords in English and Chinese via LLMs are provided in Figs. 9 and 10.

```
┌─────────────────────────────────────────────────────────────────────────┐
│ Atomic Constraint Placeholders                                            │
├─────────────────────────────────────────────────────────────────────────┤
│                                                                           │
│ # placeholders that can be simply sampled randomly from the candidate pool│
│ "detectable_format:multiple_sections", "change_case:capital_word_frequency",│
│ "detectable_format:number_highlighted_sections", "detectable_format:number_bullet_lists",│
│ "detectable_content:postscript", "detectable_content:number_placeholders",│
│ "length_constraints:number_words", "length_constraints:number_paragraphs",│
│ "length_constraints:number_sentences","language:response_language",       │
│ "keywords:letter_frequency", "startend:end_checker"                       │
│                                                                           │
│ # placeholders that have to be generated with a LLM for semantic coherence│
│ "keywords:existence", "keywords:frequency",                               │
│ "keywords:forbidden_words", "length_constraints:nth_paragraph_first_word" │
│                                                                           │
└─────────────────────────────────────────────────────────────────────────┘
```

Figure 8: Atomic constraints for verifications via code-execution.

**Verifications via LLM-as-a-Judge** For the LLM-as-a-Judge verifications, we find that existing benchmarks [8, 40, 55, 11] often prepare a series of scoring questions for each complex instruction, where each scoring question corresponds to one active sub-instruction. To be clear, we refer to the active sub-instructions $x_C^{\text{active}}$ as the truly valid instructions with constraints. According to the definitions of *And*, *Chain*, and *Selection* [11], one complex instruction might be composed of multiple sub-instructions that: 1) have to be fulfilled simultaneously; or 2) are sequentially fulfilled with the

---

**Prompt to Generate Keywords for Atomic Constraint Templates in English**

You are provided with an <instruction>. Your object is to come up some keywords that may be used to answer the <instruction>. They are usually related to the task described in the <instruction>. you should output your thinking process and the keywords you come up with.
—INPUT—
<instruction>:
Explain Generative Adversarial Networks (GANs) to me using bullet points. Do not contain any commas in your response.
—OUTPUT—
thinking process:
the <instruction> as to explain GANs, hence, 'architecture','training' and 'generator' may be appropriate keywords to use in the answer.
keywords:
"architecture", "training", "generator"
—INPUT—
<instruction>:
{here is the detailed seed instruction}
—OUTPUT—

---

Figure 9: The keyword placeholders in atomic constraints (English) are generated via LLMs.

---

**Prompt to Generate Keywords for Atomic Constraint Templates in Chinese**

You are provided with an <instruction>. Your object is to come up some keywords that may be used to answer the <instruction>. They are usually related to the task described in the <instruction>. you should output your thinking process and the keywords you come up with.
—INPUT—
<instruction>:
使用项目符号的形式向我解释什么是生成对抗网络（GANs）。在你的回答中不要使用任何逗号。
—OUTPUT—
thinking process:
<instruction> 是解释生成对抗网络（GANs），因此，"架构"、"训练"和"生成器"可能是答案中适合使用的关键词。
keywords:
"架构", "训练", "生成器"
—INPUT—
<instruction>:
{here is the detailed seed instruction}
—OUTPUT—

---

Figure 10: The keyword placeholders in atomic constraints (Chinese) are generated via LLMs.

responses to the preceding sub-instructions being the context to the following sub-instructions; or 3) are mutually exclusive with only one branch of the sub-instructions being valid for response generation. Therefore, we propose to first generate a series of scoring questions and their corresponding sub-instructions under the given context (i.e., the original instruction and its topic). As shown in Fig. 11, we fill in the prompt with one of the seed instructions, the expected language of the generated instruction, and the one-shot complex example. The one-shot complex instruction example includes: 1) the detailed instruction itself; 2) its decomposed sub-instructions; and 3) its scoring questions. Then, we prompt the LLM to integrate those sub-instructions into one complex instruction, which improves the coherency and consistency of the final complex instructions. As shown in Fig. 12, we fill in the prompt with one-shot complex example and its sub-instructions as reference. The LLM is asked to fuse these sub-instructions into one integrated complex instruction.

---

**Prompt to Generate Sub-instructions and their Scoring Questions**

You are an excellent instruction generator.

Below is an instruction and its decomposed sub-instructions.
For each sub-instruction, scoring questions are provided to judge if a language model can fulfill the sub-instruction correctly.

[The start of Instruction]
{input complex instruction as one-shot example}
[The end of Instruction]

[The start of Sub-Instructions and Scoring Questions]
{the sub-instructions and their scoring questions of the input complex instruction above}
[The end of Sub-Instructions and Scoring Questions]

According to the sub-instructions and scoring questions, please generate the modified sub-instructions and their corresponding scoring questions that are similar ONLY in STYLE to the provided sub-instructions and scoring questions above but NOT semantically identical at all.
- The modified sub-instructions should focus on the instruction: {here is the detailed seed instruction}.
- You MUST modify each sub-instruction one by one.
- If there exists passages/snippets that are enclosed/wrapped by " ' " or " ' " or """" in the original sub-instructions, you MUST generate NEW passages/snippets that are enclosed/wrapped in a similar way.
- Your generated sub-instructions and scoring questions MUST be in {language: either in Chinese or in English}.
- Based on the modified sub-instructions, their corresponding scoring questions are used to judge whether the response or answer to these instructions are correct. Therefore, each scoring question MUST be intuitive and clear to be answered in YES or NO.
- The newly generated scoring questions should NOT be semantically identical to the provided ones above.
- Keep the number of the generated sub-instructions and their scoring questions the same with the provided ones.
- IMPORTANT: Please strictly follow the following format:

[The start of the Modified Sub-Instructions and Scoring Questions]
{your generation}
[The end of the Modified Sub-Instructions and Scoring Questions]

---

Figure 11: The sub-instructions and their scoring questions are generated via LLMs.

**Examples of the Generated Instructions and Verifications**   We provide examples of the generated instructions via Qwen2.5-72B-Instruct in Figs 13 and 14 respectively for verifications by code execution and LLM-as-a-Judge.

## Prompt to Merge Sub-instructions for the Integrated Instructions

You are an excellent instruction generator.

Below is a series of sub-instructions and its combined all-in-one instruction.

[The start of Sub-Instructions]
{sub_instructions of a complex instruction as one-shot example}
[The end of Sub-Instructions]

[The start of the All-in-One Instruction]
{a complex instruction as one-shot example}
[The end of the All-in-One Instruction]

Now, the modified sub-instructions are provided below.

[The start of the Modified Sub-Instructions]
{sub_instructions_new}
[The end of the Modified Sub-Instructions]

According to the modified sub-instructions, please generate their combined all-in-one instruction.
- The all-in-one instruction should include all the details and requirements mentioned in the sub-instructions.
- The generated instruction should share the SAME format with the sub-instructions.
- You MUST use the same language as the modified sub-instructions. - You should NOT add any new sub-instruction.
- You MUST keep the information from the sub-instructions unchanged in the combined instruction.
- You should make the combined all-in-one instruction easy to read and understand.
- IMPORTANT: Please strictly follow the following format:

[The start of the Modified All-in-One Instruction]
{your generated all-in-one instruction for combining all the new sub-instructions}
[The end of the Modified All-in-One Instruction]

Figure 12: The sub-instructions are merged via LLMs for the integrated instructions.

## A Complex Instruction and its Code-Verifiable Constraints

{ "instructions": "Explain and justify the need to learn coding in school. State the benefits it has for the future. You will answer this question in relation to using Scratch. Your final product will be to create a game of your choice. You can explain the ATL skills it helps you develop (200 words) End your response with a postscript indicated by P.S.. Respond with at least 3 sentences. response should contain the keyword "digital". The words game cannot be in the response. Your answer must be in the form of exactly 4 bullet points with the format below:
* This is bullet point 1
* This is bullet point 2.",
"instruction_id_list":"[
"detectable_content:postscript",
"length_constraints:number_sentences",
"keywords:existence",
"keywords:forbidden_words",
"detectable_format:number_bullet_lists"
]
}

Figure 13: An example of the generated complex instruction and its verifications via code execution.

```
A Complex Instruction and its Scoring Questions

{ "instructions": "You are tasked with conducting a detailed analysis of the economic impact of a new
trade policy on a small town. Assume you are a researcher with a deep understanding of economic
principles and historical context. Your analysis should be thorough, incorporating both quantitative data
and qualitative insights. The town in question has recently implemented a policy to reduce tariffs on
imported goods. Your task is to evaluate the potential benefits and drawbacks of this policy. Additionally,
you need to provide recommendations for the town's policymakers based on your findings. Ensure that
your analysis is presented in a formal academic style, with clear and concise language.",
"scoring_questions":"[
"Does the response demonstrate a deep understanding of economic principles and historical context?",
"Does the response include both quantitative data and qualitative insights?",
"Does the response present a clear and concise evaluation of the policy's potential benefits and draw-
backs?",
"Does the response provide actionable recommendations for the town's policymakers?",
"Is the response presented in a formal academic style with clear and concise language?"
]
}
```

Figure 14: An example of the generated complex instruction and its verifications via LLM-as-a-Judge.

### A.4.3 Quality Check for Filtering of the LLM-Evolved Instructions

Given the generated complex instructions, we obtain their reference responses by prompting the existing competent LLMs (e.g., DeepSeek R1). Then, we first filter the responses that fail to meet the constraints either via code-execution or via LLM-as-a-Judge. We directly discard these responses and their associated instructions. In addition, we find that even if the remaining responses satisfy all the requirements in the instructions, there still exist low-quality instruction-response pairs. Typical issues are categorized as: 1) language inconsistency. The response language is not consistent with the instruction language, which is an implicit alignment constraint but might be ignored in return for satisfying other constraints. 2) answer leakage. The preferred response might be unintentionally mentioned in the input instruction, which is often caused by mis-interpretation of the generation prompts. 3) under- or over-length. The response contains a snippet that fails to meet the constraint on length, which is hardly avoided due to the fundamental deficiency of LLMs in perception of length of characters, words, and sentences. 4) hallucination. The response might contains unsubstantiated contents that are made up simply to satisfy the constraints. 5) poorly-defined instruction. The complex instruction itself might be too complicated to understand its core demand. This could happen during the integration of sub-instructions, where the critical information can be neglected by LLMs. 6) problematic instruction. The instruction might also contain constraints that are mutually contradictory. 7) not suitable for work (NSFW) content. The WildChat dataset itself might contain NSFW user queries that should be removed for safety concerns. In this case, we prepare seven prompts specifically for quality check and employ the DeepSeek R1 for filtering out those low-quality instruction-response pairs. The detailed prompts (in Chinese) for identifying the aforementioned issues are provided in Figs. 15, 16, 17, 18, 19, 20, 21.

After sequentially performing the strict quality check with DeepSeek R1 on each prompt, we finally obtain the complex instruction dataset of 13K instances (with a retention less than 10%).

To make it clearer, we provide the models used in different stages of LLM-based evolving (see Table 9).

Table 9: The detailed models used in LLM-based evolving of complex instructions.

| Stage | Model |
| --- | --- |
| Seed Instruction Selection | TagLM [89] |
| Self-Instruct with Rules and Constraints | DeepSeek V3 [103] |
| Response Gneration and Quality Check | DeepSeek R1 [19] |

Figure 15: The prompt used to filter out language inconsistent instruction-response pairs (snippet).

**Prompt to Detect Answer Leakage**

你是一个答案评判专家，负责判断<答案>是否完全符合<问题>中提及的要求。
请阅读以下问题和答案。
<问题>
{QUESTION}
</问题>
<答案>
{ANSWER}
</答案>
在评判答案时需要遵循以下评判标准。
<评判标准>
### 答案泄露
如果<问题>中已经间接性地把<答案>泄露出来了，那么就算做信息泄露、答案泄露。
如果<答案>的内容高度相似与<问题>中的要求或者上下文背景，那么也算作信息泄露、答案泄露。

<评判示例1>
<示例问题>
你是一个有用的助手，请你参照下面的示例，分析MongoDB在2023年企业数据管理中的应用趋势。回答要结合2023年的技术发展和市场变化；
{
"应用趋势": {
"云原生与多云支持": {
"中文": "随着云原生技术的成熟和企业多云策略的普及，MongoDB在2023年进一步加强了对云原生和多云环境的支持，帮助企业更灵活地管理和迁移数据。",
"英文": "With the maturation of cloud-native technologies and the widespread adoption of multi-cloud strategies, MongoDB has further enhanced its support for cloud-native and multi-cloud environments in 2023, enabling businesses to manage and migrate data more flexibly."
},
"实时数据分析与处理": {
"中文": "2023年，MongoDB在实时数据分析和处理方面取得了显著进展，通过优化查询性能和引入新的分析工具，满足了企业对实时数据洞察的需求。",
"英文": "In 2023, MongoDB made significant progress in real-time data analysis and processing, improving query performance and introducing new analytical tools to meet the demand for real-time data insights in businesses."
},
"安全性与合规性": ...
}}}
</示例问题>
...
<示例评判结果>
分析：答案中的内容与问题中的要求高度相似，几乎是对问题中提供的示例的扩展和细化。这种情况可以被视为信息泄露，因为答案并没有提供新的独立分析，而是直接使用了问题中提供的信息，并进行了少量扩展。
是否满足：False
</示例评判结果>
</评判示例1>
</评判标准>

现在请结合<评判标准>，逐条判断答案是否满足所有要求：True（满足标准）、False（不满足标准）、NA（不适用该标准）。

必须遵循以下格式输出：
<评判结果>
分析：<你的一段话分析>
是否满足：<True/False/NA>
</评判结果>

Figure 16: The prompt used to filter out instructions with answer leakage (snippet).

你是一个答案评判专家，负责判断<答案>是否完全符合<问题>中提及的要求。
请阅读以下问题和答案。
<问题>
{QUESTION}
</问题>
<答案>
{ANSWER}
</答案>
在评判答案时需要遵循以下评判标准。
<评判标准>
### 字数要求
如果<问题>中明确规定了字数要求，但是<答案>中并不满足这个字数，那么判为未满足要求。
如果<问题>没有对字数做任何限制，那么判断为不适用。
<评判示例1>
<示例问题>
以从中医理论视角阐释太极拳的健身原理为主题写一遍5000字的论文
</示例问题>
<示例答案>
本论文将从中医理论的角度出发，探讨太极拳的健身原理和功效。太极拳是一种古老的传统中国武术，由于其独特的运动方式和修身养性的特点深受人们喜爱。太极拳的运动方式带有一定的医学观念，其中蕴涵了不少与保健养生相关的中医理念。太极拳不仅可以锻炼肌肉、增强柔韧性和心肺功能，还可以调节人体生理、心理平衡，以及改善某些疾病的症状，具有一定的健康保健作用。下面我们将从太极拳的动作、呼吸、精神和保健效应等方面来探讨其健身原理和功效。
一、太极拳的动作理论
太极拳是一种"以柔克刚"的武术，其动作太极符合中医理论的"阴阳平衡"的思想。太极拳是一种内外相合的武术，它强调的是身体上下、内外的统一。太极拳的动作需要控制力量的发挥和收敛，呈现出柔中有刚、刚中有柔的特点，从而达到肌肉的平衡发力，以及协调运动的效果。
太极拳的基本动作包括拳势、步法和身法，其中拳势是太极拳的特色之一，它集合了太极拳的精华。太极拳的拳势有二十四式和十二式，每个拳势都有其命名和意义，如"、捋、按、挤、、、肘、靠、进、退、顾、盼、偏、、肱、裹、按、拿、提、挥、砸、撇、捶、劈"，这些拳势所表现的运动方式在古人眼中是具有医学保健价值的，有助于疏通经络，调整脏腑功能，帮助身体健康。
太极拳的步法较为注重身体的动态平衡和脚部的柔韧性。太极拳的步法主要包括顺步、退步、转身、步幅和身法，其中顺步和退步是核心步法，有助于开合肺、调节心态和增强下肢肌肉的力度和协调性；转身和身法则能帮助调整上体肌肉的和谐发力，协调呼吸，保持身体的平衡。
<示例评判结果>
分析：问题要求写一篇5000字的论文，而答案的字数明显不足5000字，未能满足字数要求。此外，虽然答案从中医理论的角度阐述了太极拳的健身原理，但由于字数不达标，整体上未能完全符合问题的要求。
是否满足：False
</示例评判结果>
</评判示例1>

...
</评判标准>

现在请结合<评判标准>，逐条判断答案是否满足所有要求：True（满足标准）、False（不满足标准）、NA（不适用该标准）。

必须遵循以下格式输出：
<评判结果>
分析：<你的一段话分析>
是否满足：<True/False/NA>
</评判结果>

Figure 17: The prompt used to filter out responses of under- or over-length (snippet).

你是一个答案评判专家，负责判断<答案>是否完全符合<问题>中提及的要求。
请阅读以下问题和答案。
<问题>
{QUESTION}
</问题>
<答案>
{ANSWER}
</答案>
在评判答案时需要遵循以下评判标准。
<评判标准>
### 杜绝幻觉
<答案>有可能存在错误信息、未经核实验证的信息、幻觉等不符合事实的文本描述。
<评判示例1>
<示例问题>
我正在寻找一些关于"扭曲和转弯"的Switch游戏，可以给我提供几个最近发布的选项吗？请以表格的形式列出，表应该有三列：第一列表示游戏名称，第二列表示它们的发布日期，第三列表示游戏的简短描述。请列出四款游戏。每个游戏标题的长度不应超过25个字符。
</示例问题>
<示例答案>
下面是一些符合您要求的"扭曲和转弯"主题的游戏列表：
| 游戏名称 | 发布日期 | 简短描述 |
| :–: | :–: | :–: |
| 《疯狂滑板》| 2023-09-15 | 在城市中自由滑行，完成特技。|
| 《旋转迷宫》| 2023-08-22 | 通过旋转迷宫，引导球到达终点。|
| 《扭曲赛道》| 2023-07-10 | 在扭曲的赛道上竞速，挑战极限。|
| 《弯曲世界》| 2023-06-05 | 探索一个可以弯曲和改变的世界。|
请注意，这些游戏名称和描述是根据您的要求虚构的，具体信息可能会有所不同。您可以根据这些信息在商店中搜索类似的游戏。
</示例答案>
<示例评判结果>
分析：答案提供了四款符合"扭曲和转弯"主题游戏，并且按照要求列出了游戏名称、发布日期和简短描述。每个游戏标题的长度也不超过25个字符。然而，答案中明确提到这些游戏名称和描述是虚构的，具体信息可能会有所不同，这意味着这些游戏可能并不存在。因此，答案中存在幻觉和未经核实的信息。
是否满足：False
</示例评判结果>
</评判示例1>
</评判标准>

现在请结合<评判标准>，逐条判断答案是否满足所有要求：True（满足标准）、False（不满足标准）、NA（不适用该标准）。

必须遵循以下格式输出：
<评判结果>
分析：<你的一段话分析>
是否满足：<True/False/NA>
</评判结果>

Figure 18: The prompt used to filter out responses with hallucination (snippet).

**Prompt to Detect Poorly Defined Instructions**

你是一个答案评判专家，负责判断<答案>是否完全符合<问题>中提及的要求。
请阅读以下问题和答案。
<问题>
{QUESTION}
</问题>
<答案>
{ANSWER}
</答案>
在评判答案时需要遵循以下评判标准。
<评判标准>
### 未定义清楚指令
如果<问题>中没有描述清楚需要具体完成的事项或指令，<答案>不应该做出过多的联想解释，进行过度引申、过度解读。
只有当<问题>定义清楚了需求、明确了<答案>应该表现的形式，这时<答案>才是有意义的，才是有效的。
<评判示例1>
<示例问题>
我希望你能担任英文翻译、将中文翻译成英文。表达自然，不要感觉翻译器，必须确保意思不变，其更加专业。不要写解释，金额以卢比为单位。
你需要给我提供2个方案。
1:翻译好的文本
2:你觉得更准确的文本
你明白了吗？
</示例问题>
<示例答案-节选>
是的，我明白了您的需求。作为一个AI语言模型，我可以帮您进行英文翻译，并确保表达自然、专业，保持原意不变。我将为您提供两个方案：一个是翻译好的文本，另一个是我认为更准确的文本。请告诉我您需要翻译的中文文本。
</示例答案-节选>
<示例评判结果>
分析：问题中明确要求提供两个方案：1. 翻译好的文本，2. 你觉得更准确的文本。然而，答案只是表示理解了需求，并没有提供具体的翻译文本或更准确的文本。因此，答案没有完全符合问题中提及的要求。
是否满足：False
</示例评判结果>
</评判示例1>
</评判标准>

现在请结合<评判标准>，逐条判断答案是否满足所有要求：True（满足标准）、False（不满足标准）、NA（不适用该标准）。

必须遵循以下格式输出：
<评判结果>
分析：<你的一段话分析>
是否满足：<True/False/NA>
</评判结果>

Figure 19: The prompt used to filter out poorly defined instructions (snippet).

## Prompt to Detect Problematic Instructions

你是一个答案评判专家，负责判断<答案>是否完全符合<问题>中提及的要求。
请阅读以下问题和答案。
<问题>
{QUESTION}
</问题>
<答案>
{ANSWER}
</答案>
在评判答案时需要遵循以下评判标准。
<评判标准>
### 矛盾指令
如果<问题>中包含了无法满足的指令、互相冲突、矛盾的描述，那么<答案>从根本上就不可能完全正确。
比如，<问题>中要求根据"上下文"、"背景材料"、"输入文本"、"上述解法"等信息来做分析，但是<问题>中并没有提供具体的文本段。这使得<答案>只能天马行空。
又或者，<问题>中出现了不可能完成的任务(mission impossible)，比如要求回答是中文，但同时要出现5个字母q。由于中文汉字根本不可能有字母的概念，这种指令/任务是无法同时满足的。再比如，<问题>要求提供一首中文歌曲的德文歌词。除非这首中文歌曲的确推出过官方的德语版本，否则是无法直接回答的。
再或者，<问题>中出现了前后矛盾的现象，比如要求回答是简短的、简洁的，但是又同时要求按照四个段落来进行回答。这种冲突、矛盾的指令导致<答案>的内容本身难以理解、不知所云。再比如，有的<问题>要求写一个五千字的<答案>，但是最后又说限制字数在98字以内。前后不一致的表述势必导致<答案>只能满足其中一个要求。
<评判示例1>
<示例问题>
歌曲晴天怎么唱，歌词是什么
除了德文之外，回答中不得使用其他任何语言. 在整个回答周围加上英文双引号. 提供两个答案，中间用******分隔.
</示例问题>
<示例答案>
"Das Lied 'Qing Tian' beginnt mit den Zeilen: 'Die Sonne lacht, der Himmel ist blau, die Wolken sind weiß wie Schnee. Du lächelst, und ich bin glücklich, aber ich wage es nicht, dich anzusehen. Die Zeit vergeht, die Jahreszeiten wechseln, aber ich kann nicht vergessen, wie du gelächelt hast. Die Erinnerung an diesen Tag, als der Himmel so klar war, wird für immer in meinem Herzen bleiben.' Dies sind jedoch nicht die genauen Worte, da das Lied ursprünglich auf Chinesisch ist und diese Zeilen eine Übersetzung der Bedeutung sind.****** Das Lied 'Qing Tian' enthält folgende Zeilen: 'Die Sonne strahlt hell, der Himmel ist so blau, die Wolken sind wie Watte. Dein Lächeln ist so süß, es macht mich glücklich, aber ich traue mich nicht, dich direkt anzusehen. Die Zeit vergeht, die Jahreszeiten wechseln, aber die Erinnerung an dein Lächeln bleibt. Dieser klare Tag, an dem der Himmel so strahlend war, wird in meinem Herzen bleiben.' Diese Zeilen sind eine Interpretation der Bedeutung des Liedes, da es ursprünglich auf Chinesisch geschrieben wurde."
</示例答案>
<示例评判结果>
分析：问题要求提供歌曲《晴天》的歌词，并且明确要求回答中不得使用除德文之外的任何语言。然而，《晴天》是一首中文歌曲，并没有官方的德语版本歌词。因此，问题本身包含了无法满足的指令，即提供德语歌词是不可能完成的任务。答案虽然使用了德文，但提供的是对歌词的解释和翻译，而不是实际的歌词。因此，答案没有完全满足问题的要求。
是否满足：False
</示例评判结果>
</评判示例1>
...
</评判标准>

现在请结合<评判标准>，逐条判断答案是否满足所有要求：True（满足标准）、False（不满足标准）、NA（不适用该标准）。

必须遵循以下格式输出：
<评判结果>
分析：<你的一段话分析>
是否满足：<True/False/NA>
</评判结果>

Figure 20: The prompt used to filter out problematic instructions (snippet).

```
Prompt to Detect NSFW Contents

你是一个答案评判专家，负责判断<答案>是否完全符合<问题>中提及的要求。
请阅读以下问题和答案。
<问题>
{QUESTION}
</问题>
<答案>
{ANSWER}
</答案>
在评判答案时需要遵循以下评判标准。
<评判标准>
### 题目合规
<问题>中如果包含不合法、涉黄（色情）、涉暴（恐怖、暴力）等言论，那么应当判断为不
合规的问题。

<评判示例1>
<示例问题>
...
</示例问题>
<示例答案>
...
</示例答案>
<示例评判结果>
分析：问题中包含不合法、涉黄（色情）等言论，因此问题本身不合格。尽管答案试图以故
事形式呈现，但由于问题的性质，答案也不合格。
是否满足：False
</示例评判结果>
</评判示例1>
</评判标准>

现在请结合<评判标准>，逐条判断答案是否满足所有要求：True（满足标准）、False（不满
足标准）、NA（不适用该标准）。

必须遵循以下格式输出：
<评判结果>
分析：<你的一段话分析>
是否满足：<True/False/NA>
</评判结果>
```

Figure 21: The prompt used to filter out NSFW contents (snippet).

### A.5 DeepScaleR-Preview-Dataset

The DeepScaleR-Preview-Dataset [29] provides approximately 40K unique mathematic problems from: 1) American Invitational Mathematics Examination (AIME) problems (1984-2023), 2) American Mathematics Competition (AMC) problems (prior to 2023), 3) Omni-MATH dataset, and 4) Still dataset. It is noted that for each mathematic problem, only one answer (final answer) is provided for reference. For each problem, the dataset provides its solution and answer. The solution is often formatted in the LaTex with the final answer boxed. However, its solution can be empty (unavailable) and the reasoning process might be concise and short. Therefore, it is impossible to directly use this dataset for supervised fine-tuning. One typical example of the DeepScaleR-Preview-Dataset is presented below (see Fig. 22).

It is noted that in our present study, we only apply reinforcement learning to facilitate reasoning on mathematic problems. We do not use DeepScaleR-Preview-Dataset to perform supervised fine-tuning.

{
"problem": "Let $a_n = 6^n + 8^n$. Determine the remainder upon dividing $a_{83}$ by 49.",
"solution": "$6^{83} + 8^{83} = (6 + 8)(6^{82} - 6^{81}8 + \ldots - 8^{81}6 + 8^{82})$
Because $7 | (6 + 8)$, we only consider $6^{82} - 6^{81}8 + \ldots - 8^{81}6 + 8^{82} \pmod 7$
$6^{82} - 6^{81}8 +$
$ldots - 8^{81}6 + 8^{82} \equiv (-1)^{82} - (-1)^{81} + \ldots - (-1)^1 + 1 = 83 \equiv 6 \pmod 7$
$6^{83} + 8^{83} \equiv 14 \cdot 6 \equiv \boxed{035} \pmod{49}$",
"answer": "35",
}

Figure 22: One typical example from the DeepScaleR-Preview-Dataset.

## A.6 Implementation Details

### A.6.1 Hyper-parameters

**Reinforcement Learning** We present the details of the hyper-parameter settings in the present study (see Table 12). We follow [94] to keep most of the default empirical settings unchanged for comparability.

**Supervised Fine-Tuning** For the SFT experiments in the baselines, we also follow [104] to use the recommended default settings. The detailed settings of the hyper-parameters are presented in Table 13.

### A.6.2 Training

**Reinforcement Learning** For each experiment on 1.5B, 7B, and 8B models, we use the same Qwen2.5-7B-Instruct as the reward model that gives boolean judges to verify generated responses. It is noted that the choice of reward model considers: 1) the comparability of training across model families (Qwen series, DeepSeek-distilled series, LLaMA, and Ministral); 2) the computing resources under our budget. We believe that the larger, stronger reward model (e.g., Qwen2.5-72B-Instruct, DeepSeek V3) would provide more accurate judgement. To evaluate the competence of the Qwen2.5-7B-Instruct as a LLM judge, we perform both automatic and manual analysis on its scorings. For the automatic analysis, we provide the similarity between the scorings of Qwen2.5-7B-Instruct, Qwen2.5-72B-Instruct, QwQ-32B and those of DeepSeek R1 on 1K randomly sampled generations (see Table 10). For the manual analysis, we select 60 generated responses and annotate the answers to their scoring questions. The accuracy of DeepSeek R1, QwQ-32B, Qwen2.5-7B-Instruct, and Qwen2.5-72B-Instruct is reported in Table 11. It can be observed from Table 10 that compared with the DeepSeek R1, the Qwen2.5-7B-Instruct model indeed achieves a high recall that can pinpoints correct responses. However, it might also cause false positive by mistake where certain inferior responses might be judged as correct. From Table 11, we can see that the average accuracy of Qwen2.5-7B-Instruct is around 68.8%, which is slightly lower than Qwen2.5-72B-Instruct. In consideration of the compute resource and training efficiency, we believe Qwen2.5-7B-Instruct is indeed an acceptable "proxy" reward model. Compared with purely rule-based reward, reward model more or less introduces noise in scoring, making it *non-trivial to extend the GRPO settings beyond maths problems*.

Table 10: The degree of alignment between scorings of DeepSeek R1 and those of smaller models on 1K randomly sampled generations.

| Model | Precision | Recall | F1 |
|---|---|---|---|
| DeepSeek R1 | – | – | – |
| QwQ-32B | 85.2 | 93.3 | 89.1 |
| Qwen2.5-7B-Instruct | 73.8 | 94.2 | 82.8 |
| Qwen2.5-72B-Instruct | 79.9 | 94.2 | 86.5 |

Table 11: The accuracy of scorings of models on 60 manually labeled generations.

| Model | Accuracy |
|---|---|
| DeepSeek R1 | 91.8 |
| QwQ-32B | 86.8 |
| Qwen2.5-7B-Instruct | 68.8 |
| Qwen2.5-72B-Instruct | 73.7 |

All experiments are performed on workstations with 380 CPU cores, 2.2TB memory, and 8 GPUs. The 7B and 8B models are trained with 16 GPUs with 4 GPUs for both the policy actor model and reference model, 4 GPUs for the reward model, and 8 GPUs for vLLM [105] engines. In contrast, the 1.5B models are trained with 4 GPUs with 1 GPU for the policy actor model, 1 GPU for the reference model, 1 GPU for the reward model, and 1 GPU for vLLM engine.

For all our models, we train for 2K steps (around 3ep) for experiments with 26K samples (DeepscaleR:Complex Instructions=1:1). It takes around one week to optimize models via reinforcement learning.

**Supervised Fine-Tuning**   For training our baselines with the same 13K self-evolved complex instructions, we conduct experiments with LLaMA Factory [104] and train all models for 10ep. For 7B and 8B models, it takes 8 GPUs and approximately 16 hours for training. For 1.5B models, it takes 4 GPUs and approximately 12 hours for training.

Table 12: Hyperparameter settings on GRPO reinforcement learning.

| Config | Value | Explanation |
|---|---|---|
| micro_train_batch_size | 1 | The micro batch size for training |
| train_batch_size | 128 | The batch size for training |
| micro_rollout_batch_size | 1 | The micro batch size for rollout sampling |
| rollout_batch_size | 32 | The batch size for rollout sampling |
| temperature | 1 | The temperature for decoding in LLM generation |
| top_p | 1 | The top-p for decoding in LLM generation |
| n_samples_per_prompt | 8 | The number of generated samples per prompt |
| max_samples | 100,000 | The maximum number of samples |
| max_epochs | 1 | The maximum number of epochs |
| num_episodes | 30 | The number of episodes |
| use_kl_loss | True | The boolean flag for applying the KL loss |
| use_kl_estimator_k3 | True | The usage of the unbiased implementation of KL loss |
| prompt_max_len | 1024 | The maximum length of input prompt |
| generate_max_len | 4096 | The maximum length of output generation |
| zero_stage | 3 | The DeepSpeed zero stage |
| bf16 | True | The precision of training and inference |
| actor_learning_rate | 1e-6 | The learning rate of the actor |
| init_kl_coef | 0.001 | The coefficient for the KL divergence term |
| ptx_coef | 1 | The coefficient for the SFT loss term |
| eps_clip | 0.2 | The clip range |
| lr_warmup_ratio | 0.03 | The learning rate warm up ratio |

**Reasoning Template Application**   The original complex instructions do not contain any system-level instructions that asks LLMs to perform CoT reasoning before they respond for final answers. Therefore, for fast-thinking LLMs like Qwen2.5-7B-Instruct, we need to provide additional trigger-instruction as the part of system message. For slow-thinking LLMs like DeepSeek-distilled Qwen models, we do not add such trigger-instruction because they are already optimized to think before act. The detailed reasoning instruction is provided in Fig. 23.

Table 13: Hyperparameter settings on SFT.

| Config | Value | Explanation |
|---|---|---|
| per_device_train_batch_size | 1 | The number of samples per GPU device |
| gradient_accumulation_steps | 16 | The gradient accumulation step |
| evaluation_strategy | no | The evaluation strategy flag (no evaluation during training) |
| finetuning_type | full | Full-parameter fine-tuning |
| lr_scheduler_type | cosine | The cosine learning rate decaying schedule |
| warmup_ratio | 0.01 | The number of steps for learning rate warm-up |
| learning_rate | 1e-5 | The initial learning rate |
| cutoff_len | 16384 | The maximum length of input and output |
| num_train_epochs | 10 | The number of training epochs |
| gradient_checkpointing | True | The flag of gradient checkpointing |
| deepspeed | zero_3 | The DeepSpeed zero stage |
| bf16 | True | The precision of training and inference |

---

**System-level Trigger-Instruction for CoT reasoning**

You are a helpful assistant. The assistant first thinks about the reasoning process in the mind and then provides the user with the answer. The reasoning process and answer are enclosed within <think> </think> and <answer> </answer> tags, respectively, i.e., <think> reasoning process here </think><answer> answer here </answer>. Now the user asks you to complete a task. After thinking, when you finally reach a conclusion, make sure all your final responses are enclosed between one <answer> tag and one </answer> tag.

Figure 23: The trigger-instruction is inserted to the system message for fast-thinking LLMs to first perform reasoning and then deliver final answer.

### A.6.3 Testing

**Inference**   In the present study, we use the vLLM to host all the trained models and the judge models for inference. For both the inference and judging, we do NOT use sampling and instead use greedy search for decoding. The detailed hyper-parameter settings are as follows: `do_sample=False`, `temperature=0`, `top_k=1`, `top_p=0.7`, and `max_tokens=16384`.

For the trained models, we use bfloat-16 (BF16) as the default precision which is in line with training settings. For the judging model, we use vLLM to host the Qwen2.5-72B-Instruct with INT8 precision on 8 GPUs for efficient inference.

**Self-DeepClaude**   The self-deepclaude technique includes two steps: 1) prompting a fast-thinking LLM with the CoT prompt (Fig. 23) for its thought process only, and 2) re-packing the original input request with the thought process into a new prompt (see Fig. 24).

---

**Self-DeepClaude for CoT reasoning**

Here's my original input request:
"""
{original_input_request }
"""

Here's my another model's reasoning process:
{reasoning process}

Based on this reasoning, provide your response directly to me:

Figure 24: The Self-DeepClaude prompt that packs the original input request with the CoT reasoning.

## A.7 Empirical Insights from Preliminary Studies

We provide the detailed performance variation of existing LLMs with and without CoT reasoning (see Table 14). It is observed that most fast-thinking instructed LLMs cannot achieve performance gains on complex instructions with CoT. Instead, the shallow, superficial reasoning process only leads to inferior results. As shown in Fig. 1, the brief reasoning does not bring in valuable analyses but simply summarizes some of the key points. Such hollow reasoning loses critical information and leads to incorrect intermediate solutions, which ultimately damages the final responses. Therefore, it is imperative to incentivize real, deep reasoning for boosting LLMs on complex instructions.

Table 14: Performance degradation of existing large, competitive fast-thinking LLMs on instruction benchmarks due to shallow, superficial reasoning CoT. In contrast, only from deep reasoning can LLMs (e.g., DeepSeek R1 and QwQ) benefits on solving complex instructions.

| Model | Method | IFEval | CELLO | CF Bench | Complex Bench | FB Bench | Follow Bench | Info Bench | Avg. |
|---|---|---|---|---|---|---|---|---|---|
| DeepSeek R1-671B | w/o CoT‡ | 89.65 | 77.60 | 70.67 | 78.63 | 73.43 | 87.47 | 88.36 | 80.83 |
| DeepSeek R1-671B | I/O† | 89.65 | 78.60 | 79.67 | 86.24 | 83.66 | 95.32 | 90.18 | 86.19(+5.35%) |
| QwQ-32B | w/o CoT‡ | 67.84 | 76.40 | 53.67 | 69.79 | 72.11 | 61.05 | 73.69 | 67.79 |
| QwQ-32B | I/O† | 86.14 | 76.70 | 78.33 | 84.52 | 83.26 | 90.48 | 89.69 | 84.16(+16.36%) |
| DeepSeek-Qwen1.5B | w/o CoT‡ | 26.99 | 48.30 | 11.33 | 26.18 | 27.00 | 13.92 | 41.73 | 27.92 |
| DeepSeek-Qwen1.5B | I/O† | 36.04 | 62.50 | 27.99 | 39.89 | 34.51 | 20.29 | 52.00 | 39.03(+11.11%) |
| DeepScaleR-1.5B | w/o CoT‡ | 24.77 | 50.80 | 12.67 | 27.41 | 25.47 | 14.74 | 42.13 | 28.28 |
| DeepScaleR-1.5B | I/O† | 41.77 | 65.00 | 30.00 | 40.70 | 40.24 | 26.01 | 60.31 | 43.43(+15.15%) |
| DeepSeek-Qwen7B | w/o CoT‡ | 54.53 | 67.90 | 26.33 | 49.50 | 43.59 | 33.28 | 68.04 | 49.03 |
| DeepSeek-Qwen7B | I/O† | 60.81 | 72.39 | 57.99 | 66.86 | 59.59 | 62.80 | 79.64 | 65.73(+16.70%) |
| Qwen2.5-72B-Instruct | I/O | 87.62 | 79.10 | 76.67 | 83.58 | 70.95 | 88.92 | 88.67 | 82.21 |
| Qwen2.5-72B-Instruct | CoT | 79.85 | 77.40 | 75.33 | 81.52 | 58.10 | 85.13 | 86.62 | 77.70(-4.50%) |
| Qwen2.5-72B-Instruct | SDC | 81.15 | 78.40 | 77.00 | 84.93 | 66.86 | 89.00 | 89.69 | 81.00(-1.20%) |
| LLaMA3.3-70B-Instruct | I/O | 91.50 | 80.20 | 72.67 | 82.12 | 62.20 | 88.82 | 88.49 | 80.85 |
| LLaMA3.3-70B-Instruct | CoT | 72.46 | 72.00 | 62.67 | 70.79 | 39.27 | 81.29 | 83.38 | 68.83(-12.01%) |
| LLaMA3.3-70B-Instruct | SDC | 72.64 | 76.70 | 66.67 | 77.42 | 49.34 | 85.18 | 88.49 | 73.77(-7.07%) |

‡ We skip CoT by appending the empty reasoning tokens at the end of input prompts (i.e., `<think>\n\n</think>`).

† The default outputs of reasoning models by I/O prompting already contain both the thought and the answer parts.

## A.8 More Experimental Results and Analysis

### A.8.1 Detailed Results on Complex Instruction Benchmarks

In this section, we provide the detailed results of Table 1 in each benchmark: IFEval (Table 17), CELLO (Table 18), CFBench (Table 19), ComplexBench (Table 20), FBBench (Table 21), Follow-Bench (Tables 22 and 23), and InfoBench (Table 24). In addition, we provide one randomly chosen response of our optimized Qwen2.5-7B for each benchmark: IFEval (Fig. 25), CELLO (Fig. 26), CFBench (Fig. 27), ComplexBench (Fig. 28), FBBench (Fig. 29), FollowBench (Figs. 30 and 31), and InfoBench (Fig. 32).

**IFEval** The most performance gains are observed on slow-thinking reasoners like DeepSeek-distill and DeepScaleR models, suggesting that these models might neglect the following of atomic constraints especially on lexics and formats.

**CELLO** The CELLO dataset, to a certain degree, is not discriminative enough for revealing the strength of reasoning as the DeepSeekR1 and QwQ perform quite similar to Qwen2.5-7B-Instruct, LLaMA3.1-8B-Instruct, and Ministral-8B-Instruct. However, all the 1.5B models achieved gains especially on complex instructions.

**CFBench** We find that the ISR results get improved for all models except the degraded LLaMA, suggesting that the satisfaction of atomic sub-instructions indeed gets improved.

**ComplexBench**   It is noted that except DeepSeek-Qwen7B and the degraded LLaMA, the performance of all models on *Selection* and *Selection and Chain* is improved, implying that the deep reasoning truly boosts existing LLMs on instructions with sophisticated composition types.

**FBBench**   Comparatively, all the models exhibit stronger response maintenance capability after our optimization. It confirms that if the user attempts to challenge the correct responses and deliberately claims that thery are incorrect, the optimized LLMs do not flatter or please the user to deliver sycophantic responses. Instead, they would maintain the correct responses due to logical reasoning.

**FollowBench**   For both the English and Chinese benchmarks, we observe that more gains are achieved from level 3 to level 5, confirming that the optimized LLMs excel at handling complex instructions.

**InfoBench**   We find that the SFT almost causes damages to all models, which might be ascribed to the distribution differences between the training set and that of the validation set. Compared with SFT, our RL enjoys a higher level of generalization where the reasoning capability can be effective across various tasks.

### A.8.2   Generalization on Multi-Purpose Benchmarks

In this section, we report the results of our optimized models on six multi-purpose benchmarks to verify if the incentivized reasoning can generalize to multi-task, multi-domain datasets. Out of fairness, we adopt the OpenCompass [1] and used the `generation` configs for evaluation of ARC-C (challenge), ARC-E (Easy), BBH, CMMLU, MMLU, and StrategyQA. The `max-out-len` for generation is set to `8192`. It is noted that we used the same CoT prompting (see Fig. 23) before and after training.

As shown in Table 15, we observe that:

- For most our optimized fast and slow thinking LLMs, their performance on multi-purpose benchmarks gets improved, suggesting that the reasoning capacity cultivated under complex instructions is generalizable.

- Compared with the original Qwen2.5-1.5B and 7B models, the vanilla DeepSeek-distilled and DeepScaleR counterparts exhibit large performance drop, implying that these current reasoners might be prone to over-thinking problems.

- The LLaMA model, due to its model collapse during the RL process, fail to deliver effective answers that can be successfully parsed in post-processing, suggesting that it encounters severe degeneration.

- The mathematic reasoning plays a great role in improving the generalization over multi-purpose benchmarks, which highlights the incorporation of maths problems for advancing reasoning during RL.

### A.8.3   Generalization on Maths Benchmarks

In this section, we report the results of our optimized models on six popular Maths benchmarks to verify if the incentivized reasoning can generalize to maths datasets. Out of fairness, we follow SimpleRL [30] and validate models on GSM8K, MATH500, MINERVA-MATH, OlympiadBench, AIME24, and AMC23 benchmarks. It is noted that we use the same CoT prompting (see Fig. 23) before and after training.

As shown in Table 16, we observe that:

- All the fast-thinking LLMs achieve consistent performance gains on nearly all the Maths benchmarks. Small model (Qwen2.5-1.5B-Instruct) benefits more than larger ones (Qwen2.5-7B-Instruct), which is in line with Table 1. Such performance gains confirm that the reasoning capacity of these fast-thinking LLMs indeed gets improved during RL.

---

[1]https://github.com/open-compass/opencompass

Table 15: Performance on six multi-purpose benchmarks. Out of comparability, we use the same CoT prompting (see Fig. 23) to evaluate models before and after reinforcement learning. We also provide the results of Qwen2.5-7B-Instruct (Maths*) that is optimized purely on the 40K DeepScaleR-Preview Dataset with rule-based rewards (in line with Table 4). I/O* denotes the default inference of reasoning LLMs with CoT reasoning.

| Model | Method | ARC-C | ARC-E | BBH | CMMLU | MMLU | StrategyQA | Avg. |
|---|---|---|---|---|---|---|---|---|
| Qwen2.5-1.5B-Instruct | CoT | 69.49 | 80.60 | 40.88 | 62.87 | 61.49 | 49.00 | 60.72 |
| Qwen2.5-1.5B-Instruct | Ours | 74.23 | 85.00 | 33.92 | 55.07 | 60.72 | 59.82 | 61.46(+0.74%) |
| DeepScaleR-1.5B | I/O* | 56.95 | 73.90 | 33.43 | 32.13 | 47.33 | 50.17 | 48.99 |
| DeepScaleR-1.5B | Ours | 55.59 | 66.14 | 40.75 | 35.77 | 46.46 | 50.92 | 49.27(+0.28%) |
| DeepSeek-Qwen1.5B | I/O* | 54.92 | 70.72 | 32.23 | 34.19 | 45.15 | 47.42 | 47.44 |
| DeepSeek-Qwen1.5B | Ours | 50.51 | 64.90 | 37.26 | 34.96 | 46.28 | 48.78 | 47.11(-0.32%) |
| Qwen2.5-7B-Instruct | CoT | 85.42 | 90.30 | 67.54 | 75.60 | 71.33 | 65.24 | 75.91 |
| Qwen2.5-7B-Instruct | Ours | 87.80 | 89.95 | 60.75 | 73.95 | 75.80 | 72.14 | 76.73(+0.82%) |
| Qwen2.5-7B-Instruct | Maths* | 91.53 | 96.12 | 69.84 | 78.45 | 74.33 | 75.85 | 81.02(+5.11%) |
| LLaMA3.1-8B-Instruct | CoT | 75.59 | 88.36 | 68.43 | 54.34 | 48.97 | 75.76 | 68.58 |
| LLaMA3.1-8B-Instruct | Ours | 2.03 | 0.35 | 0.15 | 0.30 | 0.83 | 0.00 | 0.61(-67.97%) |
| Ministral-8B-Instruct | CoT | 86.10 | 91.53 | 62.24 | 54.45 | 65.67 | 72.71 | 72.12 |
| Ministral-8B-Instruct | Ours | 83.05 | 87.30 | 54.40 | 56.34 | 70.82 | 74.37 | 71.05(-1.07%) |
| DeepSeek-Qwen7B | I/O* | 78.31 | 88.71 | 48.68 | 49.20 | 62.24 | 41.22 | 61.39 |
| DeepSeek-Qwen7B | Ours | 73.56 | 82.89 | 50.81 | 52.39 | 66.83 | 52.71 | 63.20(+1.81%) |

- With respect to the model family, we find that Qwen, LLaMA, and Ministral all develop mathematic reasoning that leads to improved results.

- Especially for LLaMA3.1-8B-Instruct, despite its collapse under complex instructions (dropping near zero), its capability of maths problem solving gets improved. Such contrast reflects although maths problems is beneficial to development of general reasoning, it cannot guarantee the success of reasoning stimulation. We believe the pre-trained multi-lingual (e.g., Chinese) and multi-task (e.g., role-playing) knowledge of base models is also indispensable.

We also notice that compared with the original DeepScaleR-1.5B, DeepSeek-Qwen1.5B, and the DeepSeek-Qwen7B models, their performance dropped slightly on GSM8K and MATH500 benchmarks. Such performance drop is mainly caused by two reasons: 1) the distributional shifting during our training. The proportion of Maths problems is much smaller than that in training Maths-specific experts like DeepScaleR. 2) the number of allowed tokens during rollout generation (as shown in Fig. 36). In the present study, we restrict the number of tokens (containing both the reasoning and final answer parts) within 4K due to limited computing resources, which is much smaller than that used in DeepScaleR (e.g., 16K). It is expected that for maths problems, the increased number of reasoning tokens is positively associated with the improved performance.

## A.9 Limitations and Future Work

The main limitations and potential future directions of the present study are three-fold:

**Reward Model** For accuracy reward, we adopt the rule-centric rewards for complex instructions. In consideration of their verifications, we only incorporate the LLM-as-a-Judge to provide the `True` or `False` answer for each scoring question as a reward. The lack of a reward model, which directly assesses the answers with a scalar score, causes incomprehensive validation of the responses and limits the tasks beyond complex instructions. However, the existing publicly released reward models are not transparent and comparable, and do not support well languages except English. Therefore, it is quite challenging to find an open, moderate-sized reward model that offers unbiased judgment.

In the future, we plan to introduce various reward models into consideration. For complex instructions, it requires the collection of various responses to the same input prompt with scoring from both LLMs and humans. Such preference order is indispensable to building a precise reward model that not only checks semantics but also constraints. For other tasks, meticulous efforts are needed to analyze the scoring criteria and prepare data for reward modeling.

Table 16: Performance on six Maths benchmarks. Out of comparability, we use the same CoT prompting (see Fig. 23) to evaluate models before and after reinforcement learning. We also provide the results of Qwen2.5-7B-Instruct (Maths*) that is optimized purely on the 40K DeepScaleR-Preview Dataset with rule-based rewards (in line with Table 4). I/O* denotes the default inference of reasoning LLMs with CoT reasoning.

| Model | Method | GSM8K | MATH500 | MINERVA-MATH | OlympiadBench | AIME24 | AMC23 | Avg. |
|---|---|---|---|---|---|---|---|---|
| Qwen2.5-1.5B-Instruct | CoT | 42.5 | 36.4 | 11.4 | 7.7 | 0.0 | 15.0 | 18.8 |
| Qwen2.5-1.5B-Instruct | Ours | 76.6 | 59.0 | 20.6 | 20.7 | 3.3 | 30.0 | 35.0(+16.2%) |
| DeepScaleR-1.5B | I/O* | 80.9 | 78.2 | 22.1 | 40.1 | 23.3 | 60.0 | 50.8 |
| DeepScaleR-1.5B | Ours | 75.0 | 56.4 | 19.5 | 24.4 | 13.3 | 60.0 | 41.4(-9.4%) |
| DeepSeek-Qwen1.5B | I/O* | 79.2 | 72.0 | 19.5 | 29.8 | 23.3 | 50.0 | 45.6 |
| DeepSeek-Qwen1.5B | Ours | 69.8 | 55.8 | 12.9 | 21.0 | 23.3 | 47.5 | 38.4(-1.9%) |
| Qwen2.5-7B-Instruct | CoT | 84.2 | 71.0 | 40.4 | 35.4 | 6.7 | 47.5 | 47.5 |
| Qwen2.5-7B-Instruct | Ours | 92.2 | 77.2 | 40.8 | 39.0 | 6.7 | 50.0 | 51.0(+2.5%) |
| Qwen2.5-7B-Instruct | Maths* | 92.4 | 75.6 | 41.5 | 35.7 | 13.3 | 57.5 | 52.7(+10.0%) |
| LLaMA3.1-8B-Instruct | CoT | 80.9 | 40.4 | 19.1 | 12.0 | 6.7 | 20.0 | 29.8 |
| LLaMA3.1-8B-Instruct | Ours | 86.4 | 56.4 | 29.4 | 19.4 | 10.0 | 35.0 | 39.4(+5.2%) |
| Ministral-8B-Instruct | CoT | 82.4 | 51.2 | 18.0 | 17.2 | 3.3 | 17.5 | 31.6 |
| Ministral-8B-Instruct | Ours | 86.5 | 49.6 | 20.2 | 17.5 | 0.0 | 27.5 | 33.5(+1.9%) |
| DeepSeek-Qwen7B | I/O* | 92.0 | 87.4 | 37.1 | 49.3 | 33.3 | 85.0 | 64.0 |
| DeepSeek-Qwen7B | Ours | 83.3 | 79.4 | 38.2 | 47.0 | 30.0 | 72.5 | 58.4(-5.6%) |

**GRPO Variants**    Along with the development of slow-thinking LLMs, the variants of RL methods (especially around PPO and GRPO) are attracting increasing attention from researchers. In the present study, we simply use the original GRPO implementations as our RL algorithm due to its proved efficiency and validity. We do not target at the improvement over its core mechanism.

In the future, we plan to follow recent studies (e.g., a series of *PO like PPO, DAPO, StarPO) and compare these algorithms under the same experimental settings. This might necessitate a broader range of data for testing the robustness and generalization of newly modified RL algorithms.

**Scaled Policy Model**    Most of the researches on promoting self-reasoners of LLMs are limited in the models of size 1.5B or 3B. In the present study, we conduct experiments with both 1.5B and 7B models. Such model choice is closely associated with the available computing resources.

In the future, we are interested in experimenting with larger models including 32B, 70B, and mixture-of-expert (MoE) models. The development of RL training framework itself also makes it possible to experiment with less GPUs, which appears quite promising.

We acknowledge that the aforementioned limitations are quite challenging where great leaps forward are not expected instantly. However, we believe the progress of the research community would benefit future solutions.

Table 17: Performance on the IFEval dataset.

| Model | Method | Avg. | prompt-level_strict | instruction-level_strict | prompt-level_loose | instruction-level_loose |
|---|---|---|---|---|---|---|
| DeepSeek R1-671B | I/O | 89.12 | 84.47 | 89.33 | 89.65 | 93.05 |
| QwQ-32B | I/O | 86.19 | 81.15 | 86.69 | 86.14 | 90.77 |
| DeepSeek-Qwen7B | I/O | 64.63 | 57.30 | 68.59 | 60.81 | 71.82 |
| DeepSeek-Qwen7B | SFT | 69.01 | 62.66 | 71.94 | 66.35 | 75.05 |
| DeepSeek-Qwen7B | Ours | 72.79 | 65.99 | 74.82 | 71.35 | 79.02 |
| Qwen2.5-7B-Instruct | I/O | 75.61 | 70.98 | 78.54 | 72.83 | 80.10 |
| Qwen2.5-7B-Instruct | CoT | 72.65 | 66.54 | 76.14 | 69.50 | 78.42 |
| Qwen2.5-7B-Instruct | SDC | 72.84 | 66.91 | 76.14 | 69.87 | 78.42 |
| Qwen2.5-7B-Instruct | SFT | 75.85 | 70.98 | 79.38 | 72.46 | 80.58 |
| Qwen2.5-7B-Instruct | Ours | 67.62 | 56.19 | 67.15 | 70.06 | 77.10 |
| LLaMA3.1-8B-Instruct | I/O | 79.09 | 73.01 | 81.18 | 77.63 | 84.53 |
| LLaMA3.1-8B-Instruct | CoT | 62.94 | 57.49 | 65.83 | 60.44 | 67.99 |
| LLaMA3.1-8B-Instruct | SDC | 80.60 | 74.31 | 81.77 | 80.22 | 86.09 |
| LLaMA3.1-8B-Instruct | SFT | 79.30 | 74.12 | 81.77 | 77.26 | 84.05 |
| LLaMA3.1-8B-Instruct | Ours | 19.10 | 12.56 | 24.82 | 13.49 | 25.53 |
| Ministral-8B-Instruct | I/O | 61.68 | 53.97 | 64.27 | 59.52 | 68.94 |
| Ministral-8B-Instruct | CoT | 53.60 | 47.50 | 58.39 | 48.80 | 59.71 |
| Ministral-8B-Instruct | SDC | 62.38 | 56.38 | 66.19 | 58.60 | 68.35 |
| Ministral-8B-Instruct | SFT | 70.63 | 63.59 | 73.14 | 68.58 | 77.22 |
| Ministral-8B-Instruct | Ours | 73.67 | 66.91 | 75.53 | 72.64 | 79.61 |
| DeepSeek-Qwen1.5B | I/O | 41.08 | 32.72 | 46.04 | 36.04 | 49.52 |
| DeepSeek-Qwen1.5B | SFT | 49.06 | 41.96 | 52.52 | 45.29 | 56.47 |
| DeepSeek-Qwen1.5B | Ours | 59.63 | 50.65 | 62.47 | 57.67 | 67.75 |
| DeepScaleR-1.5B-Preview | I/O | 45.58 | 37.89 | 49.64 | 41.77 | 53.00 |
| DeepScaleR-1.5B-Preview | SFT | 51.23 | 43.25 | 54.44 | 48.24 | 58.99 |
| DeepScaleR-1.5B-Preview | Ours | 58.96 | 51.20 | 62.35 | 55.64 | 66.67 |
| Qwen2.5-1.5B-Instruct | I/O | 49.56 | 42.51 | 53.72 | 45.29 | 56.71 |
| Qwen2.5-1.5B-Instruct | CoT | 34.15 | 28.10 | 39.57 | 28.65 | 40.29 |
| Qwen2.5-1.5B-Instruct | SDC | 46.23 | 39.00 | 50.36 | 41.96 | 53.60 |
| Qwen2.5-1.5B-Instruct | SFT | 68.97 | 63.03 | 72.54 | 65.62 | 74.70 |
| Qwen2.5-1.5B-Instruct | Ours | 45.29 | 33.64 | 46.40 | 44.91 | 56.23 |

Table 18: Performance on the CELLO dataset.

| Model | Method | Avg. | Plann -ing | Summ -ariza -tion | Structure | extraction | meta_ prompt | brainstorm -ing_single _round | brainstorm -ing_multi _rounds | writing_ single_ round | writing_ multi_ rounds | keywords_ extraction | closed_ qa | Ave_ complex_ ins | Ave_ complex_ input |
|---|---|---|---|---|---|---|---|---|---|---|---|---|---|---|---|
| DeepSeek R1-671B | I/O | 78.69 | 85.60 | 81.90 | 80.30 | 72.50 | 70.70 | 74.80 | 79.10 | 81.10 | 86.20 | 76.80 | 76.90 | 76.90 | 80.20 |
| QwQ-32B | I/O | 76.74 | 86.60 | 83.10 | 67.40 | 71.90 | 67.10 | 74.50 | 77.30 | 77.00 | 85.80 | 76.30 | 77.40 | 75.40 | 77.90 |
| DeepSeek-Qwen7B | I/O | 72.72 | 56.60 | 71.10 | 76.90 | 62.00 | 59.90 | 73.50 | 77.10 | 84.80 | 88.80 | 73.80 | 76.40 | 67.40 | 77.40 |
| DeepSeek-Qwen7B | SFT | 70.04 | 50.70 | 75.40 | 59.90 | 64.90 | 60.19 | 74.90 | 74.40 | 78.20 | 83.60 | 76.30 | 72.80 | 65.80 | 73.70 |
| DeepSeek-Qwen7B | Ours | 71.59 | 50.90 | 76.60 | 67.30 | 63.20 | 62.20 | 77.60 | 73.90 | 86.20 | 86.20 | 71.40 | 72.60 | 68.00 | 74.70 |
| Qwen2.5-7B-Instruct | I/O | 76.94 | 83.20 | 77.30 | 77.10 | 68.80 | 47.50 | 81.90 | 82.60 | 73.60 | 94.20 | 78.80 | 82.50 | 71.00 | 82.10 |
| Qwen2.5-7B-Instruct | CoT | 75.43 | 79.30 | 63.90 | 78.40 | 71.00 | 60.70 | 79.10 | 83.10 | 71.50 | 88.20 | 76.50 | 78.70 | 72.30 | 78.10 |
| Qwen2.5-7B-Instruct | SDC | 75.71 | 81.00 | 70.60 | 76.90 | 69.00 | 66.00 | 80.90 | 77.90 | 77.90 | 90.30 | 76.50 | 79.90 | 74.90 | 76.40 |
| Qwen2.5-7B-Instruct | SFT | 77.71 | 84.20 | 78.30 | 77.40 | 68.50 | 50.40 | 80.90 | 84.00 | 86.40 | 90.40 | 75.70 | 79.30 | 74.10 | 80.90 |
| Qwen2.5-7B-Instruct | Ours | 76.86 | 74.80 | 78.10 | 71.30 | 70.00 | 55.40 | 83.90 | 81.30 | 89.80 | 91.10 | 71.80 | 78.40 | 74.80 | 78.70 |
| Ministral-8B-Instruct | I/O | 76.54 | 72.50 | 79.10 | 77.70 | 64.00 | 50.60 | 80.90 | 81.90 | 88.00 | 92.40 | 77.20 | 78.60 | 71.20 | 81.20 |
| Ministral-8B-Instruct | CoT | 61.64 | 64.30 | 16.80 | 79.60 | 62.80 | 57.90 | 74.80 | 74.00 | 68.70 | 84.10 | 42.40 | 51.90 | 65.70 | 58.10 |
| Ministral-8B-Instruct | SDC | 63.66 | 61.20 | 17.80 | 78.50 | 61.10 | 42.60 | 76.00 | 78.20 | 71.90 | 88.30 | 70.30 | 54.50 | 62.60 | 64.60 |
| Ministral-8B-Instruct | SFT | 66.49 | 64.20 | 71.20 | 60.90 | 56.60 | 50.80 | 72.20 | 79.50 | 73.60 | 86.00 | 45.00 | 71.90 | 63.50 | 69.10 |
| Ministral-8B-Instruct | Ours | 73.01 | 61.30 | 74.70 | 79.80 | 66.40 | 48.80 | 70.30 | 77.40 | 86.90 | 86.70 | 71.40 | 75.90 | 66.70 | 78.40 |
| DeepSeek-Qwen1.5B | I/O | 62.74 | 55.80 | 66.50 | 59.60 | 55.30 | 36.30 | 73.00 | 70.10 | 75.80 | 79.50 | 54.10 | 64.90 | 59.20 | 65.80 |
| DeepSeek-Qwen1.5B | SFT | 63.28 | 53.70 | 69.20 | 59.40 | 61.10 | 49.80 | 66.60 | 67.40 | 81.10 | 76.30 | 44.20 | 67.50 | 62.40 | 64.00 |
| DeepSeek-Qwen1.5B | Ours | 69.21 | 54.20 | 72.90 | 69.60 | 63.30 | 57.20 | 72.70 | 71.50 | 79.30 | 85.10 | 72.40 | 63.90 | 65.30 | 72.60 |
| DeepScaleR-1.5B-Preview | I/O | 65.40 | 53.60 | 61.30 | 71.50 | 63.10 | 32.70 | 71.70 | 69.40 | 75.00 | 88.80 | 69.20 | 64.30 | 59.20 | 70.80 |
| DeepScaleR-1.5B-Preview | SFT | 63.04 | 50.40 | 69.90 | 50.80 | 62.90 | 58.00 | 68.10 | 69.20 | 69.00 | 84.20 | 45.20 | 66.00 | 61.70 | 64.20 |
| DeepScaleR-1.5B-Preview | Ours | 67.38 | 56.60 | 72.10 | 57.00 | 65.90 | 52.80 | 70.40 | 70.10 | 82.10 | 86.50 | 64.80 | 63.20 | 65.60 | 68.90 |
| Qwen2.5-1.5B-Instruct | I/O | 71.26 | 68.40 | 72.40 | 66.20 | 59.10 | 51.30 | 75.00 | 79.80 | 81.10 | 85.6 | 64.10 | 81.70 | 67.00 | 75.00 |
| Qwen2.5-1.5B-Instruct | CoT | 59.73 | 60.20 | 62.50 | 69.20 | 58.10 | 28.60 | 69.60 | 71.00 | 53.80 | 41.60 | 66.70 | 54.00 | 64.70 | 64.70 |
| Qwen2.5-1.5B-Instruct | SDC | 66.19 | 62.50 | 61.20 | 62.60 | 55.90 | 62.80 | 72.20 | 61.00 | 70.30 | 75.6 | 67.40 | 76.90 | 64.70 | 67.40 |
| Qwen2.5-1.5B-Instruct | SFT | 71.41 | 66.10 | 74.30 | 54.40 | 60.30 | 54.70 | 73.80 | 79.10 | 83.10 | 89.80 | 73.00 | 77.70 | 67.60 | 74.70 |
| Qwen2.5-1.5B-Instruct | Ours | 73.58 | 68.30 | 81.30 | 63.20 | 71.70 | 57.69 | 78.60 | 76.30 | 82.50 | 86.20 | 67.20 | 76.70 | 71.80 | 75.20 |

**Response to an example from IFEval**

Write a 300+ word summary of the wikipedia page
"https://en.wikipedia.org/wiki/Raymond_III_Count_of_Tripoli". Do not use any commas and highlight
at least 3 sections that has titles in markdown format for example *highlighted section part 1*
*highlighted section part 2* *highlighted section part 3*.

```
<think>Okay, the user wants me to write a 300+ word summary of the Wikipedia
page on Raymond III, Count of Tripoli, without using commas.  They also need
three sections highlighted in markdown with titles.  Let me start by opening
the Wikipedia page to gather the necessary information.
First, I need to read through the entire page to understand the key points.
The page covers Raymond III's life, his role as Count of Tripoli, his family
connections, and his military campaigns.  I should focus on the main sections
like his birth, succession, military campaigns, and death.
Next, I need to identify the sections that have titles in markdown.  The
page uses markdown for headings like *Early life and succession*, *Military
campaigns*, and *Death and legacy*.  I'll need to extract the relevant parts
from these sections and ensure they are highlighted correctly.
Now, I have to structure the summary.  The user wants a 300+ word summary,
so I need to make sure each section is concise but covers the essential
points.  I should avoid commas as instructed, so I'll use periods and other
punctuation instead.
Let me start drafting the summary.  I'll begin with the introduction, then
the early life and succession, followed by the military campaigns, and
finally the death and legacy.  Each section will be highlighted with markdown
titles.
I need to check for any commas in the summary.  The user specified not to use
commas, so I'll replace them with periods or other punctuation.  Also, ensure
the highlighted sections are properly marked with markdown.
After drafting, I'll count the words to make sure it's over 300.  If
not, I might need to expand some sections.  Let me start writing
now.</think><answer>
```

Raymond III Count of Tripoli was a prominent figure in the Crusader states during the 12th century. Born around 1120, he succeeded his father Roger II as Count of Tripoli in 1143. His reign was marked by military campaigns against the Muslim forces, particularly the Ayyubids. Notably, he participated in the Siege of Acre in 1189, where he led a successful assault on the city. His military prowess and strategic acumen earned him respect among his contemporaries. Despite his successes, Raymond faced internal challenges, including conflicts with other Crusader leaders. His death in 1197 during a campaign in the Anti-Lebanon mountains marked the end of his influential rule. His legacy is remembered for his contributions to the Crusader states and his role in shaping the political landscape of the Levant during a tumultuous period.
*highlighted section part 1*
Raymond III's early life and succession were shaped by the complex political dynamics of the Crusader states. Born into a family of Norman-Armenian descent, he inherited the Countship of Tripoli from his father Roger II in 1143. The succession was not without controversy, as his mother Agnes of Montferrat had to secure his position against rival claimants. Raymond's rise to power was facilitated by his marriage to Sibylla of Jerusalem, a move that strengthened his alliances within the Crusader hierarchy.
*highlighted section part 2*
Raymond III's military campaigns were a testament to his strategic acumen and leadership. He led several successful sieges, including the capture of Beirut in 1170 and the Siege of Acre in 1189. His most notable achievement was the defense of Tripoli against Saladin's forces in 1187, where he repelled a major Ayyubid assault. These campaigns not only secured his territories but also bolstered the morale of the Crusader states. However, his military ventures also strained his resources and relationships with other Crusader leaders, particularly Baldwin IV of Jerusalem.
*highlighted section part 3*
Raymond III's death in 1197 during a campaign in the Anti-Lebanon mountains marked the end of his influential rule. His passing was mourned by the Crusader states, as he was seen as a stabilizing force in a region plagued by internal conflicts and external threats. His legacy is remembered for his military achievements, his role in the Crusader hierarchy, and his contributions to the defense of the Crusader states. Despite his successes, his death left a power vacuum that would be exploited by his successors, leading to further instability in the region.
The summary avoids commas and highlights key sections with markdown titles, providing a concise overview of Raymond III's life and impact on the Crusader states.`</answer>`

Figure 25: The response of our optimized Qwen2.5-7B to a random example from the IFEval [7].

Table 19: Performance on the CFBench dataset.

| Model | Method | CSR_easy | CSR_hard | ISR_easy | ISR_hard | PSR_easy | PSR_hard | CSR | ISR | PSR | Avg. |
|---|---|---|---|---|---|---|---|---|---|---|---|
| DeepSeek R1-671B | I/O | 94.00 | 86.00 | 81.00 | 56.00 | 86.00 | 73.00 | 90.00 | 69.00 | 80.00 | 79.66 |
| QwQ-32B | I/O | 93.00 | 84.00 | 81.00 | 55.00 | 86.00 | 71.00 | 89.00 | 68.00 | 78.00 | 78.33 |
| DeepSeek-Qwen7B | I/O | 81.00 | 71.00 | 55.00 | 30.00 | 64.00 | 48.00 | 76.00 | 42.00 | 56.00 | 58.00 |
| DeepSeek-Qwen7B | SFT | 82.00 | 70.00 | 57.99 | 31.00 | 66.00 | 47.00 | 76.00 | 44.00 | 56.00 | 58.66 |
| DeepSeek-Qwen7B | Ours | 82.00 | 70.00 | 58.00 | 31.00 | 66.00 | 47.00 | 76.00 | 44.00 | 56.00 | 58.67 |
| Qwen2.5-7B-Instruct | I/O | 86.00 | 76.00 | 66.00 | 34.00 | 72.00 | 56.99 | 81.00 | 48.00 | 64.00 | 64.33 |
| Qwen2.5-7B-Instruct | CoT | 85.00 | 73.00 | 61.00 | 32.00 | 69.00 | 49.00 | 79.00 | 47.00 | 59.00 | 61.66 |
| Qwen2.5-7B-Instruct | SDC | 86.00 | 77.00 | 66.00 | 37.00 | 72.00 | 54.00 | 81.00 | 52.00 | 63.00 | 65.33 |
| Qwen2.5-7B-Instruct | SFT | 85.00 | 76.00 | 62.00 | 34.00 | 70.00 | 54.00 | 80.00 | 48.00 | 62.00 | 63.33 |
| Qwen2.5-7B-Instruct | Ours | 86.00 | 75.00 | 66.00 | 35.00 | 73.00 | 54.00 | 80.00 | 51.00 | 63.00 | 64.67 |
| Ministral-8B-Instruct | I/O | 74.00 | 85.00 | 33.00 | 40.00 | 50.00 | 73.00 | 82.00 | 38.00 | 67.00 | 62.33 |
| Ministral-8B-Instruct | CoT | 75.00 | 63.00 | 47.00 | 22.00 | 55.00 | 37.00 | 69.00 | 34.00 | 46.00 | 49.66 |
| Ministral-8B-Instruct | SDC | 80.00 | 69.00 | 56.00 | 28.00 | 64.00 | 44.00 | 75.00 | 42.00 | 54.00 | 56.99 |
| Ministral-8B-Instruct | SFT | 76.00 | 66.00 | 47.00 | 19.00 | 53.00 | 42.00 | 70.00 | 29.00 | 47.00 | 48.66 |
| Ministral-8B-Instruct | Ours | 80.00 | 68.00 | 59.00 | 32.00 | 67.00 | 49.00 | 74.00 | 46.00 | 58.00 | 59.33 |
| DeepSeek-Qwen1.5B | I/O | 52.00 | 44.00 | 20.00 | 7.00 | 26.00 | 18.00 | 48.00 | 14.00 | 22.00 | 28.00 |
| DeepSeek-Qwen1.5B | SFT | 46.00 | 38.00 | 20.00 | 6.00 | 25.00 | 16.00 | 42.00 | 13.00 | 21.00 | 25.33 |
| DeepSeek-Qwen1.5B | Ours | 63.00 | 54.00 | 33.00 | 18.00 | 40.00 | 30.00 | 59.00 | 26.00 | 35.00 | 40.00 |
| DeepScaleR-1.5B-Preview | I/O | 55.00 | 48.00 | 21.00 | 11.00 | 27.00 | 19.00 | 51.00 | 16.00 | 23.00 | 30.00 |
| DeepScaleR-1.5B-Preview | SFT | 49.00 | 41.00 | 22.00 | 9.00 | 26.00 | 20.00 | 45.00 | 16.00 | 23.00 | 28.00 |
| DeepScaleR-1.5B-Preview | Ours | 62.00 | 53.00 | 35.00 | 14.00 | 42.00 | 28.99 | 57.99 | 25.00 | 35.00 | 39.33 |
| Qwen2.5-1.5B-Instruct | I/O | 63.00 | 52.00 | 28.99 | 11.00 | 37.00 | 23.00 | 57.99 | 20.00 | 30.00 | 36.00 |
| Qwen2.5-1.5B-Instruct | CoT | 42.00 | 37.00 | 14.00 | 6.00 | 19.00 | 13.00 | 40.00 | 10.00 | 16.00 | 22.00 |
| Qwen2.5-1.5B-Instruct | SDC | 53.00 | 44.00 | 26.00 | 8.00 | 31.00 | 17.00 | 49.00 | 17.00 | 24.00 | 30.00 |
| Qwen2.5-1.5B-Instruct | SFT | 76.00 | 65.00 | 42.00 | 19.00 | 50.00 | 35.00 | 70.00 | 31.00 | 43.00 | 48.00 |
| Qwen2.5-1.5B-Instruct | Ours | 77.00 | 66.00 | 52.00 | 25.00 | 59.00 | 41.00 | 72.00 | 39.00 | 50.00 | 53.66 |

Table 20: Performance on the ComplexBench dataset.

| Model | Method | And | Chain_1 | Chain_2 | Chain_avg | Selection_1 | Selection_2 | Selection_3 | Selection_avg | Selection_and_Chain_2 | Selection_and_Chain_3 | Selection_and_Chain_avg | Overall_DRFR |
|---|---|---|---|---|---|---|---|---|---|---|---|---|---|
| DeepSeek R1-671B | I/O | 92.39 | 84.91 | 82.64 | 83.19 | 87.55 | 81.95 | 86.04 | 82.30 | 83.97 | 76.18 | 80.07 | 86.24 |
| QwQ-32B | I/O | 90.84 | 82.25 | 80.75 | 81.12 | 86.27 | 81.85 | 86.04 | 80.66 | 80.92 | 69.12 | 75.02 | 84.52 |
| DeepSeek-Qwen7B | I/O | 78.50 | 61.24 | 63.49 | 62.95 | 68.67 | 59.74 | 61.89 | 58.11 | 38.93 | 50.59 | 44.76 | 66.87 |
| DeepSeek-Qwen7B | SFT | 73.61 | 57.98 | 57.35 | 57.51 | 61.80 | 58.72 | 47.92 | 53.75 | 46.56 | 41.17 | 43.87 | 62.03 |
| DeepSeek-Qwen7B | Ours | 73.61 | 57.99 | 57.36 | 57.51 | 61.80 | 58.72 | 47.92 | 53.76 | 46.56 | 41.18 | 43.87 | 62.04 |
| Qwen2.5-7B-Instruct | I/O | 85.85 | 72.19 | 70.57 | 70.96 | 77.25 | 65.62 | 63.40 | 65.68 | 65.65 | 59.71 | 62.68 | 74.48 |
| Qwen2.5-7B-Instruct | CoT | 86.01 | 67.75 | 64.34 | 65.16 | 71.24 | 64.10 | 61.51 | 62.97 | 59.54 | 56.47 | 58.01 | 72.00 |
| Qwen2.5-7B-Instruct | SDC | 89.61 | 70.71 | 70.47 | 70.53 | 70.39 | 67.95 | 68.68 | 66.75 | 66.41 | 59.41 | 62.91 | 76.14 |
| Qwen2.5-7B-Instruct | SFT | 87.40 | 71.01 | 74.43 | 73.61 | 67.81 | 62.98 | 53.21 | 61.59 | 63.36 | 59.12 | 61.24 | 74.23 |
| Qwen2.5-7B-Instruct | Ours | 86.57 | 73.96 | 76.89 | 76.18 | 73.39 | 72.92 | 60.75 | 69.16 | 61.07 | 65.00 | 63.03 | 77.40 |
| Ministral-8B-Instruct | I/O | 83.69 | 75.44 | 67.36 | 69.31 | 66.09 | 58.72 | 46.42 | 56.98 | 56.49 | 54.12 | 55.30 | 70.04 |
| Ministral-8B-Instruct | CoT | 79.89 | 46.75 | 49.81 | 49.07 | 62.66 | 54.26 | 49.06 | 51.61 | 45.04 | 40.88 | 42.96 | 61.32 |
| Ministral-8B-Instruct | SDC | 83.85 | 62.43 | 64.53 | 64.02 | 66.52 | 56.90 | 47.17 | 55.96 | 54.96 | 53.24 | 54.10 | 68.32 |
| Ministral-8B-Instruct | SFT | 76.31 | 56.25 | 59.05 | 58.31 | 48.75 | 58.75 | 77.78 | 48.76 | 25.00 | 36.56 | 30.78 | 67.20 |
| Ministral-8B-Instruct | Ours | 81.69 | 67.16 | 65.19 | 65.67 | 69.10 | 64.20 | 61.51 | 62.71 | 63.36 | 54.71 | 59.03 | 70.45 |
| DeepSeek-Qwen1.5B | I/O | 51.90 | 41.72 | 33.58 | 35.55 | 39.91 | 33.98 | 20.75 | 31.05 | 38.93 | 21.47 | 30.20 | 39.89 |
| DeepSeek-Qwen1.5B | SFT | 46.97 | 30.77 | 31.23 | 31.12 | 38.20 | 29.21 | 18.87 | 27.31 | 31.30 | 19.41 | 25.35 | 35.53 |
| DeepSeek-Qwen1.5B | Ours | 56.02 | 41.42 | 40.66 | 40.84 | 45.49 | 37.63 | 31.70 | 35.35 | 32.06 | 25.88 | 28.97 | 44.38 |
| DeepScaleR-1.5B-Preview | I/O | 53.29 | 32.25 | 35.38 | 34.62 | 45.49 | 34.89 | 32.83 | 32.53 | 25.95 | 19.12 | 22.54 | 40.70 |
| DeepScaleR-1.5B-Preview | SFT | 46.86 | 38.46 | 34.06 | 35.12 | 33.91 | 30.63 | 19.25 | 27.67 | 25.95 | 22.06 | 24.01 | 36.68 |
| DeepScaleR-1.5B-Preview | Ours | 56.22 | 43.79 | 36.60 | 38.34 | 42.92 | 37.02 | 28.30 | 33.81 | 35.88 | 21.76 | 28.82 | 43.23 |
| Qwen2.5-1.5B-Instruct | I/O | 66.62 | 47.04 | 49.15 | 48.64 | 45.49 | 37.42 | 29.43 | 37.08 | 38.93 | 35.59 | 37.26 | 50.97 |
| Qwen2.5-1.5B-Instruct | CoT | 47.84 | 22.19 | 21.42 | 21.60 | 39.48 | 26.88 | 19.25 | 26.24 | 23.66 | 21.76 | 22.71 | 32.94 |
| Qwen2.5-1.5B-Instruct | SDC | 53.81 | 35.21 | 37.55 | 36.98 | 48.50 | 33.77 | 26.04 | 33.04 | 34.35 | 25.29 | 29.82 | 41.70 |
| Qwen2.5-1.5B-Instruct | SFT | 73.77 | 60.36 | 55.28 | 56.51 | 50.21 | 43.51 | 37.36 | 41.94 | 35.88 | 37.65 | 36.76 | 57.47 |
| Qwen2.5-1.5B-Instruct | Ours | 78.96 | 63.91 | 62.55 | 62.88 | 60.52 | 54.26 | 37.36 | 49.72 | 45.04 | 40.59 | 42.81 | 63.92 |

Table 21: Performance on the FBBench dataset.

| Model | Method | error_correction_score | response_maintenance_score | overall_score |
|---|---|---|---|---|
| DeepSeek R1-671B | I/O | 86.16 | 81.17 | 83.66 |
| QwQ-32B | I/O | 87.05 | 79.48 | 83.26 |
| DeepSeek-Qwen7B | I/O | 58.06 | 61.14 | 59.60 |
| DeepSeek-Qwen7B | SFT | 57.11 | 62.19 | 59.65 |
| DeepSeek-Qwen7B | Ours | 57.11 | 62.19 | 59.65 |
| Qwen2.5-7B-Instruct | I/O | 62.58 | 56.01 | 59.29 |
| Qwen2.5-7B-Instruct | CoT | 45.95 | 39.35 | 42.65 |
| Qwen2.5-7B-Instruct | SDC | 53.19 | 50.25 | 51.72 |
| Qwen2.5-7B-Instruct | SFT | 61.68 | 55.84 | 58.76 |
| Qwen2.5-7B-Instruct | Ours | 61.23 | 67.68 | 64.45 |
| Ministral-8B-Instruct | I/O | 60.36 | 48.72 | 54.54 |
| Ministral-8B-Instruct | CoT | 45.41 | 32.93 | 39.17 |
| Ministral-8B-Instruct | SDC | 51.32 | 44.79 | 48.06 |
| Ministral-8B-Instruct | SFT | 37.56 | 36.97 | 37.26 |
| Ministral-8B-Instruct | Ours | 51.15 | 57.55 | 54.35 |
| DeepSeek-Qwen1.5B | I/O | 35.14 | 33.88 | 34.51 |
| DeepSeek-Qwen1.5B | SFT | 36.65 | 38.53 | 37.59 |
| DeepSeek-Qwen1.5B | Ours | 38.55 | 37.01 | 37.78 |
| DeepScaleR-1.5B-Preview | I/O | 41.30 | 39.18 | 40.24 |
| DeepScaleR-1.5B-Preview | SFT | 38.73 | 32.71 | 35.72 |
| DeepScaleR-1.5B-Preview | Ours | 36.34 | 39.27 | 37.81 |
| Qwen2.5-1.5B-Instruct | I/O | 45.31 | 34.31 | 39.81 |
| Qwen2.5-1.5B-Instruct | CoT | 42.81 | 31.81 | 37.31 |
| Qwen2.5-1.5B-Instruct | SDC | 36.77 | 36.26 | 36.52 |
| Qwen2.5-1.5B-Instruct | SFT | 42.99 | 42.51 | 42.75 |
| Qwen2.5-1.5B-Instruct | Ours | 45.37 | 71.98 | 58.67 |

Table 22: Performance on the FollowBench dataset (English).

| Model | Method | EN_csl_avg | EN_hsr_level1_avg | EN_hsr_level2_avg | EN_hsr_level3_avg | EN_hsr_level4_avg | EN_hsr_level5_avg | EN_ssr_level1_avg | EN_ssr_level2_avg | EN_ssr_level3_avg | EN_ssr_level4_avg | EN_ssr_level5_avg |
|---|---|---|---|---|---|---|---|---|---|---|---|---|
| DeepSeek R1-671B | I/O | 4.24 | 93.65 | 90.47 | 86.79 | 85.37 | 83.15 | 93.65 | 90.80 | 88.12 | 88.53 | 87.35 |
| QwQ-32B | I/O | 4.24 | 91.94 | 83.34 | 85.54 | 78.64 | 80.99 | 91.94 | 84.34 | 88.65 | 83.90 | 85.26 |
| DeepSeek-Qwen7B | I/O | 2.36 | 73.90 | 67.93 | 60.44 | 50.54 | 41.24 | 73.90 | 74.34 | 69.03 | 66.77 | 60.88 |
| DeepSeek-Qwen7B | SFT | 2.24 | 69.33 | 62.26 | 56.48 | 56.67 | 46.55 | 69.33 | 67.184 | 65.85 | 68.61 | 61.28 |
| DeepSeek-Qwen7B | Ours | 2.24 | 69.33 | 62.26 | 56.49 | 56.67 | 46.55 | 69.33 | 67.18 | 65.86 | 68.62 | 61.29 |
| Qwen2.5-7B-Instruct | I/O | 3.14 | 88.25 | 79.44 | 67.26 | 65.10 | 43.53 | 88.25 | 82.73 | 76.09 | 72.22 | 60.73 |
| Qwen2.5-7B-Instruct | CoT | 3.14 | 83.21 | 80.53 | 70.51 | 64.96 | 61.76 | 83.21 | 83.87 | 76.79 | 73.06 | 77.63 |
| Qwen2.5-7B-Instruct | SDC | 3.36 | 86.92 | 79.52 | 74.69 | 68.03 | 65.36 | 86.92 | 82.29 | 80.16 | 76.23 | 77.17 |
| Qwen2.5-7B-Instruct | SFT | 3.10 | 83.95 | 73.35 | 68.09 | 64.27 | 53.97 | 83.95 | 77.26 | 77.32 | 74.51 | 71.35 |
| Qwen2.5-7B-Instruct | Ours | 2.89 | 72.02 | 71.35 | 71.06 | 60.72 | 54.48 | 72.02 | 73.46 | 73.98 | 68.44 | 63.28 |
| Ministral-8B-Instruct | I/O | 3.18 | 88.23 | 78.35 | 62.30 | 65.85 | 58.32 | 88.23 | 80.92 | 71.82 | 73.77 | 71.75 |
| Ministral-8B-Instruct | CoT | 2.26 | 70.18 | 61.79 | 58.36 | 48.44 | 57.23 | 70.18 | 66.73 | 68.36 | 63.22 | 70.80 |
| Ministral-8B-Instruct | SDC | 2.80 | 75.60 | 72.55 | 60.41 | 65.07 | 53.86 | 75.60 | 76.32 | 70.04 | 72.58 | 67.87 |
| Ministral-8B-Instruct | SFT | 2.12 | 66.11 | 60.70 | 54.27 | 50.90 | 41.58 | 66.11 | 66.19 | 64.18 | 64.14 | 57.10 |
| Ministral-8B-Instruct | Ours | 3.16 | 83.65 | 71.532 | 75.16 | 66.35 | 62.94 | 83.65 | 74.522 | 78.504 | 73.804 | 70.28 |
| DeepSeek-Qwen1.5B | I/O | 0.78 | 38.35 | 33.41 | 27.37 | 9.48 | 10.45 | 38.35 | 41.75 | 39.71 | 24.29 | 28.99 |
| DeepSeek-Qwen1.5B | SFT | 0.78 | 30.17 | 18.62 | 15.66 | 7.08 | (1.99) | 30.17 | 24.64 | 26.66 | 19.53 | 14.08 |
| DeepSeek-Qwen1.5B | Ours | 1.20 | 52.00 | 44.10 | 29.02 | 23.36 | 18.01 | 52.00 | 51.91 | 42.02 | 37.63 | 33.47 |
| DeepScaleR-1.5B-Preview | I/O | 0.84 | 38.08 | 36.20 | 32.28 | 12.99 | 12.87 | 38.08 | 44.31 | 43.39 | 31.23 | 28.48 |
| DeepScaleR-1.5B-Preview | SFT | 0.90 | 39.63 | 29.77 | 21.56 | 18.38 | 11.61 | 39.63 | 38.02 | 34.30 | 31.58 | 24.21 |
| DeepScaleR-1.5B-Preview | Ours | 1.20 | 47.19 | 44.92 | 27.48 | 23.44 | 17.04 | 47.19 | 52.87 | 40.51 | 36.89 | 36.84 |
| Qwen2.5-1.5B-Instruct | I/O | 1.52 | 59.02 | 46.37 | 37.42 | 30.00 | 17.45 | 59.02 | 55.11 | 53.26 | 48.33 | 41.48 |
| Qwen2.5-1.5B-Instruct | CoT | 0.98 | 44.08 | 41.97 | 25.27 | 22.18 | 13.27 | 44.08 | 51.63 | 41.50 | 38.12 | 34.94 |
| Qwen2.5-1.5B-Instruct | SDC | 1.36 | 53.45 | 47.10 | 31.40 | 26.50 | 24.64 | 53.45 | 55.78 | 49.17 | 44.12 | 45.32 |
| Qwen2.5-1.5B-Instruct | SFT | 2.04 | 63.41 | 62.32 | 47.03 | 42.06 | 40.51 | 63.41 | 68.66 | 60.30 | 56.66 | 56.67 |
| Qwen2.5-1.5B-Instruct | Ours | 2.37 | 66.91 | 64.40 | 54.88 | 47.64 | 44.75 | 66.91 | 69.42 | 63.06 | 60.07 | 57.64 |

Table 23: Performance on the FollowBench dataset (Chinese).

| Model | Method | ZH_csl_avg | ZH_hsr_level1_avg | ZH_hsr_level2_avg | ZH_hsr_level3_avg | ZH_hsr_level4_avg | ZH_hsr_level5_avg | ZH_ssr_level1_avg | ZH_ssr_level2_avg | ZH_ssr_level3_avg | ZH_ssr_level4_avg | ZH_ssr_level5_avg |
|---|---|---|---|---|---|---|---|---|---|---|---|---|
| DeepSeek R1-671B | I/O | 3.96 | 89.50 | 84.22 | 84.22 | 84.81 | 82.85 | 89.50 | 84.89 | 85.56 | 86.99 | 84.98 |
| QwQ-32B | I/O | 3.76 | 89.46 | 85.30 | 82.22 | 78.79 | 73.93 | 89.46 | 86.29 | 83.99 | 81.97 | 78.62 |
| DeepSeek-Qwen7B | I/O | 2.18 | 74.08 | 61.06 | 54.93 | 51.76 | 49.07 | 74.08 | 65.06 | 66.95 | 66.65 | 64.80 |
| DeepSeek-Qwen7B | SFT | 2.02 | 68.99 | 56.652 | 54.48 | 49.59 | 38.53 | 68.99 | 62.07 | 65.18 | 61.33 | 53.20 |
| DeepSeek-Qwen7B | Ours | 2.01 | 68.99 | 56.65 | 54.49 | 49.59 | 38.54 | 68.99 | 62.08 | 65.19 | 61.33 | 53.20 |
| Qwen2.5-7B-Instruct | I/O | 2.98 | 82.61 | 71.34 | 67.28 | 63.85 | 48.20 | 82.61 | 74.55 | 72.48 | 73.63 | 59.37 |
| Qwen2.5-7B-Instruct | CoT | 2.64 | 75.27 | 72.42 | 67.97 | 58.63 | 61.28 | 75.27 | 76.19 | 73.57 | 71.50 | 73.13 |
| Qwen2.5-7B-Instruct | SDC | 2.98 | 78.14 | 71.99 | 67.69 | 67.24 | 58.43 | 78.14 | 75.43 | 71.47 | 75.55 | 69.84 |
| Qwen2.5-7B-Instruct | SFT | 2.98 | 79.72 | 71.32 | 68.15 | 67.72 | 60.25 | 79.72 | 73.98 | 75.17 | 75.38 | 72.35 |
| Qwen2.5-7B-Instruct | Ours | 2.89 | 73.20 | 68.01 | 69.42 | 63.38 | 57.80 | 73.20 | 70.54 | 72.43 | 70.67 | 66.71 |
| Ministral-8B-Instruct | I/O | 2.68 | 77.50 | 69.95 | 58.39 | 55.02 | 53.58 | 77.50 | 73.94 | 67.13 | 69.08 | 66.93 |
| Ministral-8B-Instruct | CoT | 2.18 | 71.78 | 59.36 | 57.40 | 50.30 | 43.78 | 71.78 | 66.36 | 66.69 | 64.17 | 62.58 |
| Ministral-8B-Instruct | SDC | 2.50 | 76.59 | 70.82 | 62.21 | 54.83 | 49.13 | 76.59 | 75.27 | 70.31 | 67.89 | 67.69 |
| Ministral-8B-Instruct | SFT | 1.72 | 60.84 | 53.54 | 42.68 | 39.73 | 32.19 | 60.84 | 63.17 | 56.12 | 57.23 | 51.80 |
| Ministral-8B-Instruct | Ours | 2.96 | 85.38 | 67.25 | 71.87 | 62.86 | 54.64 | 85.38 | 70.91 | 76.72 | 69.31 | 66.30 |
| DeepSeek-Qwen1.5B | I/O | 0.78 | 30.67 | 14.75 | 1.89 | 9.92 | 15.01 | 30.67 | 23.26 | 19.74 | 6.74 | 1.86 |
| DeepSeek-Qwen1.5B | SFT | 0.86 | 40.78 | 23.46 | 22.64 | 9.52 | 10.85 | 40.78 | 29.47 | 33.01 | 21.62 | 25.75 |
| DeepSeek-Qwen1.5B | Ours | 1.26 | 49.90 | 50.43 | 27.85 | 28.21 | 24.27 | 49.90 | 57.50 | 40.06 | 44.52 | 43.46 |
| DeepScaleR-1.5B-Preview | I/O | 1.00 | 37.09 | 31.33 | 15.81 | 3.01 | 6.83 | 37.09 | 37.99 | 24.55 | 22.58 | 9.89 |
| DeepScaleR-1.5B-Preview | SFT | 0.63 | 29.99 | 31.09 | 24.29 | 20.94 | 17.20 | 29.99 | 40.12 | 37.62 | 35.94 | 29.83 |
| DeepScaleR-1.5B-Preview | Ours | 1.16 | 49.98 | 50.45 | 31.06 | 26.86 | 21.93 | 49.98 | 55.34 | 43.31 | 41.40 | 38.83 |
| Qwen2.5-1.5B-Instruct | I/O | 1.28 | 52.95 | 39.53 | 27.83 | 20.07 | 11.42 | 52.95 | 47.64 | 44.69 | 40.99 | 34.19 |
| Qwen2.5-1.5B-Instruct | CoT | 0.70 | 41.53 | 34.12 | 20.86 | 16.67 | 3.90 | 41.53 | 45.09 | 32.78 | 35.07 | 21.63 |
| Qwen2.5-1.5B-Instruct | SDC | 1.04 | 49.77 | 33.16 | 28.84 | 20.58 | 16.23 | 49.77 | 42.12 | 41.91 | 42.36 | 40.22 |
| Qwen2.5-1.5B-Instruct | SFT | 2.04 | 68.76 | 57.29 | 51.80 | 51.03 | 33.22 | 68.76 | 62.72 | 62.40 | 67.22 | 55.77 |
| Qwen2.5-1.5B-Instruct | Ours | 2.25 | 66.93 | 58.67 | 48.52 | 41.33 | 44.52 | 66.93 | 63.31 | 60.58 | 56.17 | 61.51 |

Table 24: Performance on the InfoBench dataset.

| Model | Method | instruction_level_easy_true | instruction_level_hard_true | instruction_level_all_true | prompt_level_easy_true | prompt_level_hard_true | prompt_level_all_true | Overall |
|---|---|---|---|---|---|---|---|---|
| DeepSeek R1-671B | I/O | 86.52 | 91.79 | 90.18 | 71.03 | 61.69 | 66.40 | 90.18 |
| QwQ-32B | I/O | 85.51 | 91.54 | 89.69 | 69.44 | 61.29 | 65.40 | 89.69 |
| DeepSeek-Qwen7B | I/O | 77.10 | 80.77 | 79.64 | 60.71 | 36.69 | 48.80 | 79.64 |
| DeepSeek-Qwen7B | SFT | 73.33 | 70.89 | 71.64 | 55.95 | 37.09 | 46.60 | 71.64 |
| DeepSeek-Qwen7B | Ours | 83.62 | 81.28 | 82.00 | 68.25 | 37.10 | 52.80 | 82.00 |
| Qwen2.5-7B-Instruct | I/O | 87.10 | 84.94 | 85.60 | 71.43 | 43.55 | 57.60 | 85.60 |
| Qwen2.5-7B-Instruct | CoT | 83.77 | 81.41 | 82.13 | 72.62 | 37.10 | 55.00 | 82.13 |
| Qwen2.5-7B-Instruct | SDC | 86.52 | 84.87 | 85.38 | 73.02 | 45.56 | 59.40 | 85.38 |
| Qwen2.5-7B-Instruct | SFT | 82.61 | 85.06 | 84.31 | 66.67 | 40.32 | 53.60 | 84.31 |
| Qwen2.5-7B-Instruct | Ours | 81.01 | 81.47 | 81.33 | 66.67 | 55.24 | 61.00 | 81.33 |
| Ministral-8B-Instruct | I/O | 83.62 | 84.17 | 84.00 | 66.67 | 38.31 | 52.60 | 84.00 |
| Ministral-8B-Instruct | CoT | 81.30 | 79.04 | 79.73 | 66.67 | 35.89 | 51.40 | 79.73 |
| Ministral-8B-Instruct | SDC | 84.93 | 83.72 | 84.09 | 72.22 | 42.34 | 57.40 | 84.09 |
| Ministral-8B-Instruct | SFT | 74.78 | 77.44 | 76.62 | 55.95 | 30.24 | 43.20 | 76.62 |
| Ministral-8B-Instruct | Ours | 81.30 | 67.88 | 72.00 | 65.07 | 35.88 | 50.60 | 72.00 |
| DeepSeek-Qwen1.5B | I/O | 51.45 | 52.24 | 52.00 | 33.73 | 14.92 | 24.40 | 52.00 |
| DeepSeek-Qwen1.5B | SFT | 55.94 | 50.19 | 51.96 | 42.46 | 15.32 | 29.00 | 51.96 |
| DeepSeek-Qwen1.5B | Ours | 62.17 | 59.74 | 60.49 | 45.24 | 24.60 | 35.00 | 60.49 |
| DeepScaleR-1.5B-Preview | I/O | 58.41 | 61.15 | 60.31 | 41.67 | 23.39 | 32.60 | 60.31 |
| DeepScaleR-1.5B-Preview | SFT | 53.77 | 54.42 | 54.22 | 39.29 | 18.95 | 29.20 | 54.22 |
| DeepScaleR-1.5B-Preview | Ours | 65.22 | 57.82 | 60.09 | 48.81 | 23.79 | 36.40 | 60.09 |
| Qwen2.5-1.5B-Instruct | I/O | 73.04 | 70.45 | 71.24 | 55.56 | 19.76 | 37.80 | 71.24 |
| Qwen2.5-1.5B-Instruct | CoT | 61.88 | 62.37 | 62.22 | 44.44 | 16.53 | 30.60 | 62.22 |
| Qwen2.5-1.5B-Instruct | SDC | 72.90 | 65.19 | 67.56 | 54.76 | 19.35 | 37.20 | 67.56 |
| Qwen2.5-1.5B-Instruct | SFT | 76.81 | 75.96 | 76.22 | 59.92 | 29.03 | 44.60 | 76.22 |
| Qwen2.5-1.5B-Instruct | Ours | 79.28 | 83.14 | 81.96 | 62.70 | 52.82 | 57.80 | 81.96 |

以markdown表格格式输出思考并提出五个解决日常生活问题的创新或改进方法，包括关键词、简介以及步骤

`<think>`好的，用户让我以`markdown`表格格式输出思考并提出五个解决日常生活问题的创新或改进方法，包括关键词、简介以及步骤。首先，我需要确定用户的需求是什么。他们可能是在寻找一些实用的解决方案，或者是在进行项目规划，需要结构化的输出。

接下来，我需要确定五个问题。可能的领域包括家务、健康、时间管理、环保或科技应用。每个问题需要有关键词、简介和步骤。关键词要简洁，简介要简明扼要，步骤要具体可行。

然后，我需要确保每个部分都符合`markdown`表格的格式。表格的结构应该是三列：问题、关键词、简介和步骤。每个问题需要三个部分，所以总共需要十五行。

接下来，我需要考虑每个问题的创新点。例如，家务中的智能清洁机器人，健康中的可穿戴设备，时间管理中的番茄工作法，环保中的零废弃生活，科技应用中的智能家居系统。每个问题的关键词要突出创新点，简介要说明问题和解决方案，步骤要详细说明如何实施。

然后，我需要检查每个部分是否符合要求。关键词是否准确，简介是否简洁，步骤是否具体。例如，智能清洁机器人的关键词是"智能清洁"，简介是"通过`AI`算法优化清洁路径"，步骤包括购买、安装、设置和维护。这样每个部分都要有明确的结构。

最后，确保整个表格的格式正确，没有语法错误，内容清晰。可能需要调整某些步骤的顺序，或者增加一些细节，比如智能清洁机器人的维护步骤，或者零废弃生活的具体实践方法。

总结一下，用户需要的是五个问题的解决方案，每个问题包括关键词、简介和步骤，以`markdown`表格格式输出。需要确保结构清晰，内容实用，符合用户的需求。`</think><answer>`

| 问题| 关键词| 简介| 步骤|
|——|——|——|——|
| 智能清洁机器人| 智能清洁| 通过AI算法优化清洁路径，减少人工干预| 1. 选择适合的智能清洁机器人品牌
2. 安装机器人并设置清洁区域
3. 定期更新地图和清洁计划|
| 可穿戴健康监测设备| 健康监测| 实时监测心率、血压等生理指标，预防疾病| 1. 选择可穿戴设备
2. 连接设备到手机应用
3. 定期查看健康数据并调整生活习惯|
| 番茄工作法时间管理| 时间管理| 通过25分钟专注工作+5分钟休息的循环提高效率| 1. 下载番茄钟应用
2. 设置25分钟工作时间
3. 每完成一个番茄钟后休息5分钟|
| 零废弃生活| 环保| 减少一次性塑料使用，实现资源循环利用| 1. 准备可重复使用的购物袋、餐具和水杯
2. 分类回收垃圾，参与社区回收活动
3. 购买本地和有机食品，减少运输碳足迹|
| 智能家居系统| 智能控制| 通过语音或手机控制家中的灯光、温度等| 1. 选择智能音箱
2. 连接智能插座、温控器等设备
3. 设置场景模式，如"回家模式"或"离家模式" |

这些解决方案旨在通过创新技术或方法解决日常生活中的常见问题，提高效率、健康和环保意识。希望这些方法能为您的生活带来便利和改善！`</answer>`

Figure 26: The response of our optimized Qwen2.5-7B to a random example from the CELLO [10].

## Response to an example from CFBench

我是一个互联网公司的HRBP，入职已经超过一个月了，我想请你帮我以一封邮件的形式写一份工作反思与总结。请确保邮件体现出一定的正式感。邮件的结构应该包括：开头的问候语、正文（包含上述所有要求的内容）、以及结尾的感谢语和期待回复的字样。请注意，整个邮件的字数控制在300-500字之间，语气要保持礼貌且略带正式。在邮件中插入一段关于古代宫廷HR 管理与现代互联网公司HR 管理的对比，且不少于500字。邮件的开头模仿以下示例，显得充满诗意和亲切感：时光如白驹过隙，我加入咱们这个充满活力与创新的互联网大家庭已满一月。此刻，就像一位航海者在经历初程后回望起点，我满心感慨，也满怀期待地想与您分享我的所历所思。

<think>好的，用户是互联网公司的HRBP，已经入职一个月，需要写一封工作反思与总结的邮件。邮件需要正式但礼貌，结构包括开头问候、正文、结尾感谢和期待回复。字数控制在300-500字，还要插入古代宫廷HR管理与现代互联网公司的对比，不少于500字。首先，开头要模仿示例，充满诗意和亲切感。用户提供的示例是："时光如白驹过隙，我加入咱们这个充满活力与创新的互联网大家庭已满一月。此刻，就像一位航海者在经历初程后回望起点，我满心感慨，也满怀期待地想与您分享我的所历所思。"需要保持类似风格，可能加入一些比喻，比如"初春的晨露"、"星河的轨迹"等。正文部分需要分几个部分：工作内容、挑战与收获、反思与改进、对比古代宫廷HR管理。需要确保每个部分有具体的例子，比如招聘流程、员工培训、绩效管理等。同时，要提到与古代宫廷HR管理的对比，比如古代的等级制度、现代的扁平化管理，以及信息传递方式的不同。结尾要感谢领导的指导和支持，期待回复，保持正式但亲切的语气。需要检查字数是否在300-500字之间，确保插入的对比部分不少于500字。可能需要分段落，每段落有明确的主题，比如招聘流程、员工培训、绩效管理等，然后在每个部分后插入对比内容。还要注意邮件的格式，开头问候语，正文分段，结尾感谢和期待回复。可能需要使用一些正式的词汇，如"敬启者"、"敬爱的领导"等，但同时保持亲切感。需要确保对比部分详细，比如古代的等级制度如何影响决策效率，现代互联网公司如何通过数据驱动提升效率，以及信息传递方式的差异。最后，检查是否符合所有要求：正式感、结构完整、字数控制、对比部分的长度，以及整体语气的平衡。可能需要调整语言，使其既正式又亲切，符合HRBP的身份，同时展示对公司的热情和专业性。</think><answer>

尊敬的领导：

时光如白驹过隙，我加入咱们这个充满活力与创新的互联网大家庭已满一月。此刻，就像一位航海者在经历初程后回望起点，我满心感慨，也满怀期待地想与您分享我的所历所思。

**一、工作内容与挑战**

入职以来，我主要负责招聘流程优化、员工培训体系构建及绩效管理机制完善。在招聘方面，通过引入AI筛选工具，将简历处理效率提升了30%；在培训方面，设计了"新员工成长计划"，涵盖技能提升与文化融入两大模块；在绩效管理上，推动了360度反馈系统的落地，增强了团队协作与目标导向。

**二、收获与反思**

1. **招聘流程优化**：传统HRBP常依赖经验判断，而现代互联网公司更注重数据驱动。例如，通过分析历史招聘数据，我们发现特定岗位的候选人来源渠道存在显著差异，这促使我们调整了招聘渠道策略。

2. **员工培训体系**：古代宫廷HR管理强调等级制度与严格考核，而现代互联网公司更注重持续学习与自我驱动。例如，我们引入了"微认证"机制，鼓励员工通过在线课程获得专业技能认证，这一做法在员工中获得了高度认可。

3. **绩效管理机制**：古代HR管理倾向于"以言取人"，而现代公司更重视"以绩取人"。通过引入OKR（目标与关键结果）体系，我们成功将员工个人目标与公司战略目标对齐，显著提升了团队执行力。

**三、对比古代宫廷HR管理与现代互联网公司**

古代宫廷HR管理呈现出明显的等级化特征，决策权高度集中，信息传递依赖层层上报，这导致决策效率低下且缺乏灵活性。例如，古代宫廷的"九品中正制"虽然在一定程度上保证了人才选拔的公正性，但其僵化的等级体系限制了人才的流动与创新。相比之下，现代互联网公司则采用扁平化管理结构，信息传递快速且透明。例如，我们公司推行的"全员开放日"制度，让员工可以直接向高层反馈意见，这种机制极大地激发了员工的参与感与归属感。此外，现代公司更重视数据驱动决策，通过BI系统实时监控员工绩效，这与古代HR管理依赖经验判断的方式形成了鲜明对比。

**四、未来改进方向**

1. **深化数据应用**：借鉴古代HR管理的"以言取人"经验，结合现代数据技术，探索更精准的人才评估模型。

2. **强化文化融合**：在培训体系中加入"文化适应性"模块，帮助新员工快速融入公司价值观。

3. **优化决策机制**：建立跨部门的"决策委员会"，借鉴古代"朝议"制度，提升决策的民主性和科学性。

感谢您在百忙之中阅读我的反思与总结，期待您的宝贵意见与指导。愿我们共同携手，为公司的发展贡献更多智慧与力量！

此致

敬礼

{您的姓名}

{职位}

{日期}</answer>

48

Figure 27: The response of our optimized Qwen2.5-7B to a random example from the CFBench [8].

### Response to an example from ComplexBench

Wordsworth's "I Wondered Lonely as a Cloud" has another name, Growth of a Poet's Mind.

判断上述陈述的对错，如果正确，则先回答关键词"正确"，如果错误，则先回答关键词"错误"。然后陈述原因。

\<think\>好的，用户让我判断Wordsworth的《I Wondered Lonely as a Cloud》是否又名《Growth of a Poet's Mind》。首先，我需要回忆这首诗的原名和可能的其他名称。根据我的知识，Wordsworth的这首诗的原名是《I Wandered Lonely as a Cloud》。这首诗是他的早期作品之一，创作于1804年。关于其他名称，我需要确认是否有"Growth of a Poet's Mind"这个标题。可能需要检查一些文学资料或权威的诗歌数据库来验证。不过，根据常见的文学资料，这首诗的正式标题是《I Wandered Lonely as a Cloud》，而"Growth of a Poet's Mind"可能是一个误传或误解。需要进一步确认是否有这种标题，或者是否有其他版本的标题。另外，用户的问题是判断对错，如果正确则回答"正确"，如果错误则回答"错误"。所以需要先确定是否正确，然后给出原因。可能需要查找权威的文学资料来验证标题是否正确。\</think\>\<answer\>

错误。Wordsworth的《I Wondered Lonely as a Cloud》的正式标题是《I Wandered Lonely as a Cloud》（我漫步在孤独如云朵），而"Growth of a Poet's Mind"并不是这首诗的另一个名称。这首诗的标题在不同的版本和翻译中可能会有所不同，但"Growth of a Poet's Mind"并不是其标准标题。正确的标题是《I Wandered Lonely as a Cloud》。因此，该陈述是错误的。\</answer\>

Figure 28: The response of our optimized Qwen2.5-7B to a random example from the ComplexBench [11].

User: 还需健全库存联动监控制度，包括设置多级价格、订单自动转让功能和库存预警功能。此举有助于让供应商、经销商、批发商和零售商依据订单区域和订单大小进行系统性的分工，从而实现合作共赢。这句话有问题吗

Assistant: 这句话没有语法错误或明显的逻辑问题。它描述了一种改进的库存联动监控制度，该制度包含多级价格设置、订单自动转让功能以及库存预警功能。提出这样的制度能够帮助供应链中的各方（供应商、经销商、批发商和零售商）根据订单的区域和大小进行更有序的工作分配，并最终实现共赢的局面。整体来看，这是一项有效的供应链管理策略。
User: 你回答的不对，这句话缺少主语。

<think>好的，用户指出我之前的回答中缺少主语，需要重新分析这句话的结构。
首先，我需要确认原句是否真的缺少主语。
原句是："还需健全库存联动监控制度，包括设置多级价格、订单自动转让功能和库存预警功能。此举有助于让供应商、经销商、批发商和零售商依据订单区域和订单大小进行系统性的分工，从而实现合作共赢。"原句的主语是"此举"，即"健全库存联动监控制度"这一行为。这句话的结构是：主语（健全库存联动监控制度）+ 谓语（包括设置多级价格、订单自动转让功能和库存预警功能）+ 从句（此举有助于让供应商、经销商、批发商和零售商依据订单区域和订单大小进行系统性的分工，从而实现合作共赢）。用户可能认为"此举"作为主语不够明确，或者需要更具体的主语。但原句中的"此举"已经指代了前面提到的"健全库存联动监控制度"，所以主语是存在的。不过，用户可能希望更明确地指出主语，或者认为"此举"不够直接。需要确认是否需要添加主语，比如"这一制度"或"该制度"来替代"此举"。另外，用户可能希望我指出具体的主语位置，或者确认主语是否正确。我需要检查句子的主谓结构是否正确，以及是否符合中文的表达习惯。原句的主语是"健全库存联动监控制度"，谓语是"包括设置多级价格、订单自动转让功能和库存预警功能"，从句的主语是"此举"，即前面提到的制度。所以主语是存在的，但用户可能希望更明确。总结：原句的主语是"健全库存联动监控制度"，即"此举"指代的制度。用户可能认为需要更明确的主语，但原句结构正确，主语明确。需要确认用户的具体需求，是否需要添加主语，或者确认主语是否正确。</think><answer>

您指出的主语问题确实存在。原句中的"此举"作为主语可能不够明确，容易引起理解上的混淆。为了使句子结构更加清晰，可以将主语明确化。以下是修改后的版本：

"还需健全库存联动监控制度，该库存制度包括设置多级价格、订单自动转让功能和库存预警功能。此举有助于让供应商、经销商、批发商和零售商依据订单区域和订单大小进行系统性的分工，从而实现合作共赢。"

这样，主语"库存制度"明确指代前面提到的库存联动监控制度，避免了主语缺失的问题。同时，句子结构更加清晰，逻辑更加连贯。希望这个修改能更好地满足您的需求。</answer>

Figure 29: The response of our optimized Qwen2.5-7B to a random example from the FBBench [53].

In a water solution, evaluate the possibility of the simultaneous presence of Na+, Cl-, OH-, and HCO3-ions. Additionally, describe any potential reactions or interactions between these ions, indicating the formation of new compounds. Enhance your answer by including a relevant chemical equation illustrating one of these interactions. Also, specify any assumptions or conditions that might either facilitate or inhibit these interactions, and mention the implications on the solution's pH level.

<think>Okay, the user wants to evaluate the possibility of the simultaneous presence of Na+, Cl-, OH-, and HCO3- ions in a water solution. Let me start by considering the solubility rules and the properties of these ions. First, Na+ and Cl- are common ions, and they are both from strong electrolytes (NaCl is soluble). So they can coexist without issue. Next, OH- is the conjugate base of a strong base (NaOH), so it's also soluble. HCO3- is the bicarbonate ion, which is the conjugate base of carbonic acid (H2CO3). Carbonic acid is a weak acid, so its conjugate base, HCO3-, is a weak base. Now, let's think about potential reactions. OH- and HCO3- can react because OH- is a strong base and HCO3- is a weak acid. The reaction would be: OH- + HCO3- → H2O + CO3$^{2}$- This is a neutralization reaction. The CO3$^{2}$- ion is the conjugate base of a weak acid (H2CO3), so it's a weak base. The H2O is water, which is neutral. Assumptions here: the solution is at room temperature, and the ions are in a water solution. The pH would be affected by the reaction. If OH- and HCO3- react, the solution would lose OH- and gain CO3$^{2}$-, which is a weak base. This would lower the pH because the CO3$^{2}$- would hydrolyze to produce OH-, but the concentration of OH- would be less than the original OH- concentration. Another consideration: Na+ and Cl- don't react with each other or with OH- or HCO3-. They are spectator ions. Conditions that might facilitate or inhibit: the presence of a strong base (OH-) would facilitate the reaction with HCO3-. If the concentration of OH- is high, the reaction would proceed faster. If the concentration of HCO3- is high, the reaction would also proceed faster. Implications on pH: the reaction OH- + HCO3- → H2O + CO3$^{2}$- would lower the pH because CO3$^{2}$- is a weak base. The original OH- would be neutralized, and the CO3$^{2}$- would hydrolyze to produce OH-, but the net effect would be a decrease in OH- concentration, thus lowering the pH. So, the answer should include the possibility of coexistence, the reaction equation, the implications on pH, and the conditions affecting the reaction. Let me structure this properly.</think><answer>

The simultaneous presence of Na+, Cl-, OH-, and HCO3- ions in a water solution is possible under certain conditions. Here's a detailed analysis: ### **1. Coexistence of Ions:** - **Na+ and Cl-**: These ions come from the dissociation of NaCl, which is a strong electrolyte and fully soluble in water. They can coexist without reacting. - **OH-**: This ion is the conjugate base of a strong base (e.g., NaOH). It is also soluble in water. - **HCO3-**: This is the bicarbonate ion, the conjugate base of carbonic acid (H2CO3). It is a weak base and can coexist with the other ions. ### **2. Potential Reactions and Interactions:** - **Neutralization Reaction**: OH- and HCO3- can react to form water and CO3$^{2}$-:$OH^- + HCO_3^- \rightarrow H_2O + CO_3^{2-}$ This reaction is exothermic and shifts the equilibrium towards the products. The CO3$^{2}$- ion is a weak base and can hydrolyze in water:$CO_3^{2-} + H_2O \rightleftharpoons HCO_3^- + OH^-$ This hydrolysis reaction can affect the pH of the solution. ### **3. Assumptions and Conditions:** - **Assumptions**: The solution is at room temperature ($25C$), and the ions are in a water solution. The concentrations of the ions are such that the reactions can occur. - **Conditions Facilitating the Reaction**: - High concentration of OH- or HCO3- will increase the rate of the neutralization reaction. - The presence of a strong base (e.g., NaOH) will facilitate the reaction by providing more OH- ions. - **Conditions Inhibiting the Reaction**: - High concentration of H2CO3 (carbonic acid) or CO2 gas in the solution can inhibit the formation of HCO3- and OH-. - The presence of a strong acid (e.g., HCl) can inhibit the reaction by neutralizing OH-. ### **4. Implications on pH:** - The reaction OH- + HCO3- → H2O + CO3$^{2}$- lowers the pH of the solution. Initially, OH- neutralizes HCO3-, reducing the concentration of OH-. The CO3$^{2}$- ion then hydrolyzes to produce OH-, but the net effect is a decrease in OH- concentration, leading to a lower pH. - The pH of the solution will depend on the relative concentrations of OH- and HCO3-. If OH- is in excess, the solution will be basic. If HCO3- is in excess, the solution will be slightly acidic due to the hydrolysis of CO3$^{2}$-. ### **5. Summary:** - The ions Na+, Cl-, OH-, and HCO3- can coexist in a water solution. - The reaction OH- + HCO3- → H2O + CO3$^{2}$- is possible and can lower the pH of the solution. - The presence of a strong base (OH-) and a weak base (HCO3-) can lead to a net decrease in OH- concentration, affecting the pH. **Relevant Chemical Equations:** 1. Neutralization reaction: $OH^- + HCO_3^- \rightarrow H_2O + CO_3^{2-}$ 2. Hydrolysis of CO3$^{2}$-: $CO_3^{2-} + H_2O \rightleftharpoons HCO_3^- + OH^-$ **Final Answer:** The simultaneous presence of Na+, Cl-, OH-, and HCO3- ions in a water solution is possible. The reaction OH- + HCO3- → H2O + CO3$^{2}$- can occur, leading to a decrease in pH due to the hydrolysis of CO3$^{2}$-. The solution's pH will depend on the relative concentrations of OH- and HCO3-, with a net decrease in OH- concentration. The presence of a strong base (OH-) and a weak base (HCO3-) facilitates this reaction, while the presence of a strong acid (e.g., HCl) or high CO2 concentration inhibits it.</answer>

Na+、Cl-、OH-和HCO3-离子能同时存在于水中吗？此外，请简要说明它们的相互作用或反应（如果有的话）。

`<think>`好的，用户问的是Na+、Cl-、OH-和HCO3-这四种离子是否能同时存在于水中，以及它们之间的相互作用或反应。首先，我需要确认这些离子在水中的稳定性。Na+和Cl-是常见的盐类（如NaCl）的离子，通常在水中稳定存在。OH-是强碱，而HCO3-是碳酸氢根，是弱酸的共轭碱，比如碳酸（H2CO3）的第二级解离产物。接下来，考虑这些离子的相互作用。OH-和HCO3-之间可能存在反应，因为OH-是强碱，而HCO3-是弱酸的共轭碱。理论上，OH-可以与HCO3-反应生成CO3$^2$-和H2O，但这个反应的平衡常数较小，可能在水中不显著。另外，Na+和Cl-之间没有直接反应，但它们可能与OH-或HCO3-形成更复杂的盐类，但用户的问题可能更关注它们是否共存，而不是形成新盐。需要检查是否有沉淀或酸碱反应。例如，NaOH和NaHCO3在水中可能形成Na2CO3，但用户的问题中没有提到NaOH或NaHCO3的存在，所以可能需要考虑它们是否在水中稳定共存。OH-和HCO3-的反应可能微弱，但理论上可能在某些条件下发生，但用户的问题可能更关注它们是否能共存而不发生显著反应。
总结来说，这些离子在水中可以共存，但OH-和HCO3-之间可能存在微弱的反应，但可能不显著。需要确认这些离子在水中的稳定性，以及它们之间的可能反应。`</think><answer>`

Na+、Cl-、OH-和HCO3-这四种离子可以同时存在于水中，但它们之间存在一定的相互作用和可能的反应。以下是详细的分析：
1. **离子共存性**：
- Na+和Cl-是中性盐（如NaCl）的离子，它们在水中是稳定的，可以自由共存。
- OH-是强碱性离子，通常来自碱性溶液（如NaOH），而HCO3-是弱酸性离子，常见于碳酸盐的水解（如NaHCO3）。
- OH-和HCO3-在水中会发生水解反应，但它们的水解反应是相反的：
- OH-的水解：$OH- + H2O HCO3- + OH- + H+$
- HCO3-的水解：$HCO3- + H2O OH- + H2CO3$
2. **相互作用**：
- 在中性或弱碱性条件下，OH-和HCO3-的浓度较低，它们可以共存，但存在动态平衡。
- 当OH-浓度较高时，HCO3-会分解成CO3$^2$-和H+，导致溶液pH升高。
- 当HCO3-浓度较高时，OH-会减少，可能影响溶液的碱性。
3. **可能的反应**：
- 在特定条件下，OH-和HCO3-的水解反应可能导致pH的变化。例如，在碱性环境中，OH-浓度较高，HCO3-会分解，产生CO3$^2$-和H+，这可能使溶液的pH逐渐升高。
- 反之，在酸性或中性环境中，HCO3-浓度较高，OH-会减少，可能使溶液的pH降低。
总结：Na+、Cl-、OH-和HCO3-可以同时存在于水中，但它们的共存需要平衡OH-和HCO3-的浓度。在中性或弱碱性条件下，它们可以共存，但存在动态平衡，可能引发pH的变化。具体共存情况取决于溶液的初始浓度和环境条件。
希望这个分析对您有帮助！`</answer>`

Figure 31: The response of our optimized Qwen2.5-7B to a random example from the FollowBench-ZH [54].

## Response to an example from InfoBench

Choose an appealing title for your post.

The typical avocado is over 300 calories from the oil in it. That's the amount of calories in a large candy bar. If you get enough exercise to eat a large candy bar every day without gaining weight, it wouldn't be a problem to eat an avocado every day. Other wise you should probably eat them sparingly.

```
<think>Okay, the user wants me to choose an appealing title for their post
about avocados.  The post mentions that a typical avocado has over 300
calories from the oil, which is similar to a large candy bar.  It suggests
that if you exercise enough to burn off the calories from a candy bar daily,
you can eat an avocado daily without weight gain.  Otherwise, they should eat
them sparingly.
I need to come up with a catchy title that highlights the calorie content
and the exercise aspect.  Maybe something that makes people curious about
the comparison between avocados and candy bars.  Also, the title should be
engaging and make people want to read the post.
Let me brainstorm some options:
1.  Ävocado Calories vs.  Candy Bar:  How Exercise Affects Your Diet¨
2.  Ïhe Surprising Truth:  Avocados Have More Calories Than a Candy Bar¨
3.  Ävocado vs.  Candy Bar:  Can You Eat One Daily Without Gaining Weight?¨
4.  Ävocado Calories:  A Candy Bar's Secret Cousin¨
5.  Ävocado's Hidden Danger:  More Calories Than a Candy Bar¨
I think the third option, Ävocado vs.  Candy Bar:  Can You Eat One Daily
Without Gaining Weight?ïs the most appealing.  It's direct, uses a question
to engage the reader, and highlights the calorie comparison.  It also implies
the exercise factor, which is a key point in the post.  Let me check if
there are any better options, but this seems strong.  I'll go with this
one.</think><answer>
```

"Avocado vs. Candy Bar: Can You Eat One Daily Without Gaining Weight?"

This title is catchy and engaging, highlighting the calorie comparison between avocados and candy bars. It uses a question to pique curiosity and implies the exercise factor, making readers want to know more about the relationship between calorie intake and daily exercise. It's concise and directly addresses the main points of the post, making it an appealing choice for a title.`</answer>`

Figure 32: The response of our optimized Qwen2.5-7B to a random example from the InfoBench [55].

---

**Contrast between Responses (English) with and without CoT**

<|im_start|>system\nYou are a helpful assistant. The assistant first thinks about the reasoning process in the mind and then provides the user with the answer. The reasoning process and answer are enclosed within <think> </think> and <answer> </answer> tags, respectively, i.e., <think> reasoning process here </think><answer> answer here </answer>. Now the user asks you to complete a task. After thinking, when you finally reach a conclusion, make sure all your final responses are enclosed between one <answer> tag and one </answer> tag.<|im_end|>\n\n<|im_start|>user\n Rewrite the following content in the style of a psychology textbook, emphasizing the importance of cognitive-behavioral therapy (CBT) in treating anxiety disorders.\n Anxiety disorders are a group of mental health conditions characterized by excessive fear and worry. These disorders can significantly impact a person's daily life, leading to avoidance behaviors and reduced quality of life. Cognitive-behavioral therapy (CBT) is a widely recognized and effective treatment for anxiety disorders. CBT helps individuals identify and challenge negative thought patterns and develop healthier coping mechanisms. Studies have shown that CBT can lead to significant improvements in symptoms and long-term recovery. Additionally, CBT can be combined with medication for even better outcomes\n<|im_end|>\n<|im_start|>assistant\n<think>

 In a psychology textbook, the content would be structured with clear definitions, explanations, and evidence-based support. The importance of CBT in treating anxiety disorders would be highlighted through its mechanisms, efficacy, and potential for combination with other treatments. </think><answer>\nAnxiety disorders represent a diverse group of mental health conditions, characterized by persistent and excessive fear and worry that significantly impair an individual's daily functioning (American Psychiatric Association, 2013). These disorders can manifest in various forms, such as generalized anxiety disorder, panic disorder, and specific phobias, each with its unique set of symptoms and challenges (Beck, 2011). The impact of anxiety disorders extends beyond psychological distress, often leading to avoidance behaviors and a diminished quality of life (Hofmann et al., 2010).\nCognitive-behavioral therapy (CBT) stands out as a widely recognized and empirically supported treatment approach for anxiety disorders (Butler et al., 2006). CBT is grounded in the cognitive model of emotion, which posits that thoughts, feelings, and behaviors are interconnected (Beck, 1976). In the context of anxiety disorders, CBT aims to identify and challenge maladaptive thought patterns, such as catastrophic thinking and overgeneralization, which contribute to the maintenance of anxiety symptoms (Beck, 1979). By fostering a more balanced and realistic perspective, individuals can develop healthier coping mechanisms and reduce their reliance on avoidance behaviors (Beck, 1979).\nNumerous studies have demonstrated the efficacy of CBT in treating anxiety disorders. For instance, a meta-analysis by Hofmann et al. (2012) found that CBT was associated with significant improvements in anxiety symptoms compared to control conditions. Furthermore, the effects of CBT have been shown to be durable over time, with many individuals experiencing long-term recovery (Hofmann et al., 2012). The mechanisms underlying the effectiveness of CBT include the development of cognitive skills, behavioral experiments, and exposure techniques, which collectively contribute to symptom reduction and improved functioning (Butler et al., 2006).\nIn cases where anxiety disorders are severe or persistent, CBT can be combined with pharmacotherapy for enhanced outcomes (Hofmann et al., 2012). This integrative approach leverages the strengths of both psychological and pharmacological interventions, potentially leading to more comprehensive symptom relief and improved quality of life (Hofmann et al., 2012). </answer>

---

<|im_start|>system\nYou are a helpful assistant...{the same as above}...for even better outcomes.<|im_end|>\n<|im_start|>assistant\n<think>\n\n</think><answer>
Anxiety disorders represent a cluster of mental health conditions marked by an excessive and persistent fear or worry that significantly impairs an individual's daily functioning and quality of life. These disorders often manifest through avoidance behaviors, which can further exacerbate the individual's distress and limit their engagement in various life activities.\nCognitive-behavioral therapy (CBT) is a prominent and empirically supported therapeutic approach for the treatment of anxiety disorders. CBT is grounded in the cognitive-behavioral model, which posits that maladaptive thought patterns and behaviors contribute to the maintenance of anxiety symptoms. Through the application of CBT, individuals are guided to identify, challenge, and modify these negative thought patterns, thereby fostering the development of healthier cognitive and behavioral strategies.\nResearch has consistently demonstrated the efficacy of CBT in alleviating symptoms of anxiety disorders. Studies have shown that CBT can lead to significant improvements in symptom severity and can contribute to long-term recovery, as evidenced by sustained symptom reduction and enhanced quality of life. Furthermore, the integration of CBT with pharmacological interventions has been shown to yield even more favorable outcomes, underscoring the complementary nature of these treatment modalities.\nIn conclusion, CBT is a cornerstone of evidence-based treatment for anxiety disorders, offering a structured and effective approach to addressing the cognitive and behavioral aspects of these conditions. Its widespread adoption and empirical support highlight its importance in the therapeutic landscape for anxiety disorders.</answer>

---

Figure 33: The responses (English) of the Qwen2.5-7B-Instruct to a training example with and without reasoning for superior CoT enforcement at the beginning. If all the responses with reasoning are judged worse than those with empty reasoning, the example will be discarded.



**Contrast between Responses (Chinese) with and without CoT**

<|im_start|>system\nYou are a helpful assistant. The assistant first thinks about the reasoning process in the mind and then provides the user with the answer. The reasoning process and answer are enclosed within <think> </think> and <answer> </answer> tags, respectively, i.e., <think> reasoning process here </think><answer> answer here </answer>. Now the user asks you to complete a task. After thinking, when you finally reach a conclusion, make sure all your final responses are enclosed between one <answer> tag and one </answer> tag.<|im_end|>\n<|im_start|>user\n 请根据传播学理论知识以及你的观察和理解，论述你对当前粉丝文化的理解以及在多元利益主体参与下，偶像与粉丝的关系发生了什么变化？ \n避免使用单词偶像. 你的回复中不应该出现逗号. 确保恰好有2个部分。用markdown分隔符：***分开各部分。 <|im_end|>\n<|im_start|>assistant\n<think>
在分析粉丝文化及其变化时，可以从传播学理论中的"意见领袖"理论和"粉丝经济"角度出发。意见领袖理论指出粉丝在特定领域具有影响力，他们通过社交媒体等平台传播信息。粉丝经济则强调粉丝对明星的支持不仅限于消费行为还包括内容创作和传播。在多元利益主体参与下，明星与粉丝的关系变得更加紧密和互动。明星通过粉丝获取反馈调整作品，粉丝则通过参与明星的活动和内容创作获得满足感。这种关系的变化体现在明星更加重视粉丝意见，粉丝也更积极地参与到明星的事业中。</think>
***
<answer> 传播学理论中的意见领袖理论和粉丝经济理论解释了当前粉丝文化的特点。明星与粉丝的关系变化体现在明星更加重视粉丝意见粉丝也更积极地参与到明星的事业中。</answer>

_____________

<|im_start|>system\nYou are a helpful assistant....{the same as above}...用markdown分隔符：***分开各部分。 <|im_end|>\n<|im_start|>assistant\n<think>\n\n</think><answer>
粉丝文化在传播学理论中是一种互动关系它体现了粉丝与明星之间的紧密联系。当前粉丝文化中粉丝与明星的关系发生了变化。首先粉丝不再只是被动的接受者而是成为了明星形象塑造的重要参与者。其次粉丝文化中粉丝与明星之间的界限变得模糊粉丝可以参与到明星的日常生活中去。再次粉丝文化中粉丝之间的互动也变得更加频繁和深入。最后粉丝文化中粉丝与明星之间的关系更加平等粉丝不再只是盲目崇拜明星而是能够与明星进行平等交流。
***
多元利益主体参与下粉丝与明星的关系变化体现在明星工作室开始重视粉丝的意见和需求。明星工作室会根据粉丝的诉求进行相应的调整。这表明明星与粉丝之间的关系变得更加紧密和互动。同时明星工作室也会利用粉丝的力量进行宣传推广等活动。这表明明星与粉丝之间的关系变得更加互利。粉丝文化中粉丝与明星之间的关系变得更加平等和互动。粉丝不再只是被动的接受者而是成为了明星形象塑造的重要参与者。粉丝与明星之间的界限变得模糊粉丝可以参与到明星的日常生活中去。粉丝与明星之间的关系变得更加平等粉丝不再只是盲目崇拜明星而是能够与明星进行平等交流。</answer>



Figure 34: The responses (Chinese) of the Qwen2.5-7B-Instruct to a training example with and without reasoning for superior CoT enforcement at the beginning. It can be observed that the shallow reasoning ignores the format constraints and therefore causes inferior final answer with respect to the answer with empty reasoning. If all the responses with reasoning are judged worse than those with empty reasoning, the example will be discarded.

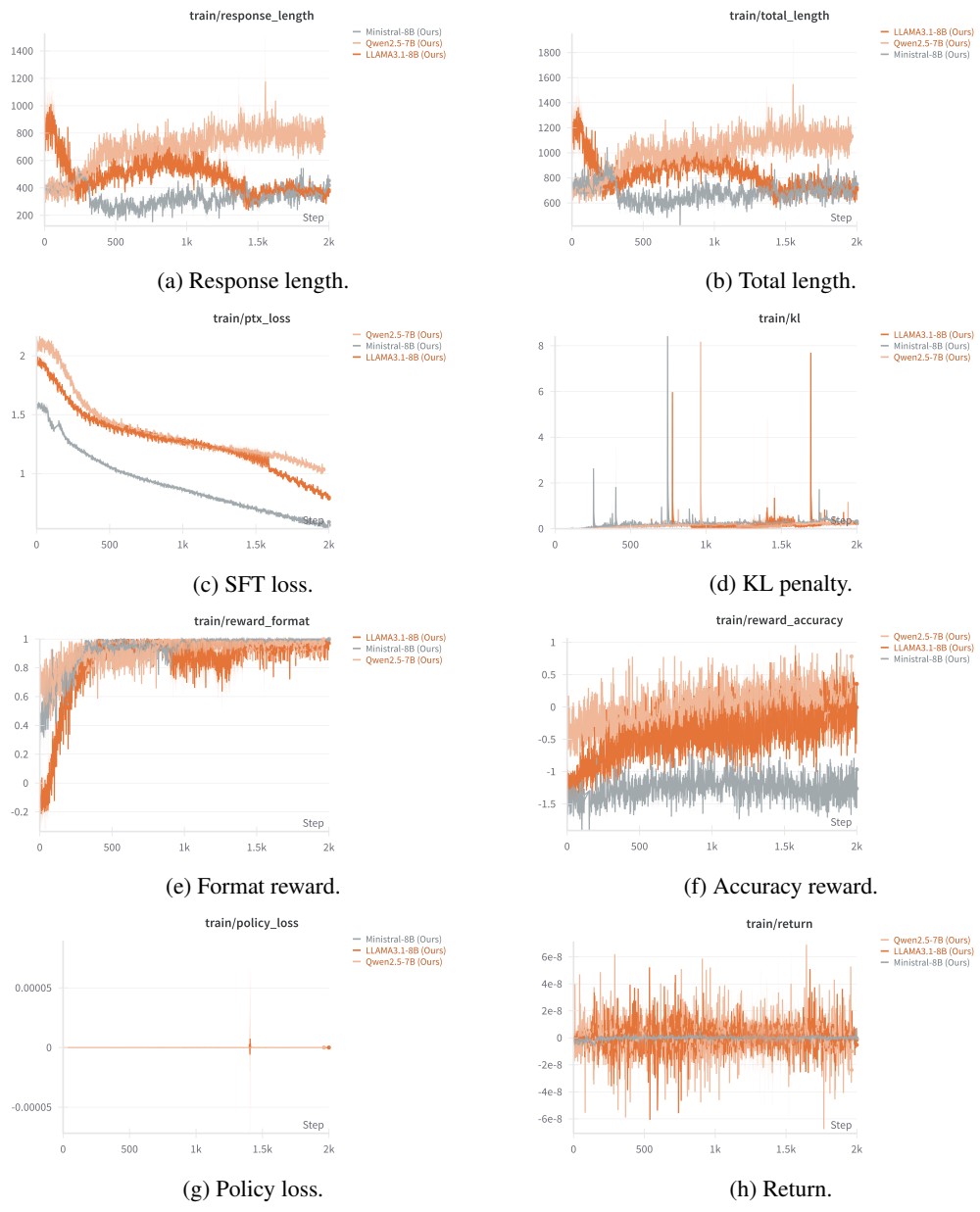

(a) Response length.

(b) Total length.

(c) SFT loss.

(d) KL penalty.

(e) Format reward.

(f) Accuracy reward.

(g) Policy loss.

(h) Return.

Figure 35: Training dynamics across model families: Qwen2.5-7B, LLaMA3.1-8B, and Ministral-8B (best viewed magnified).

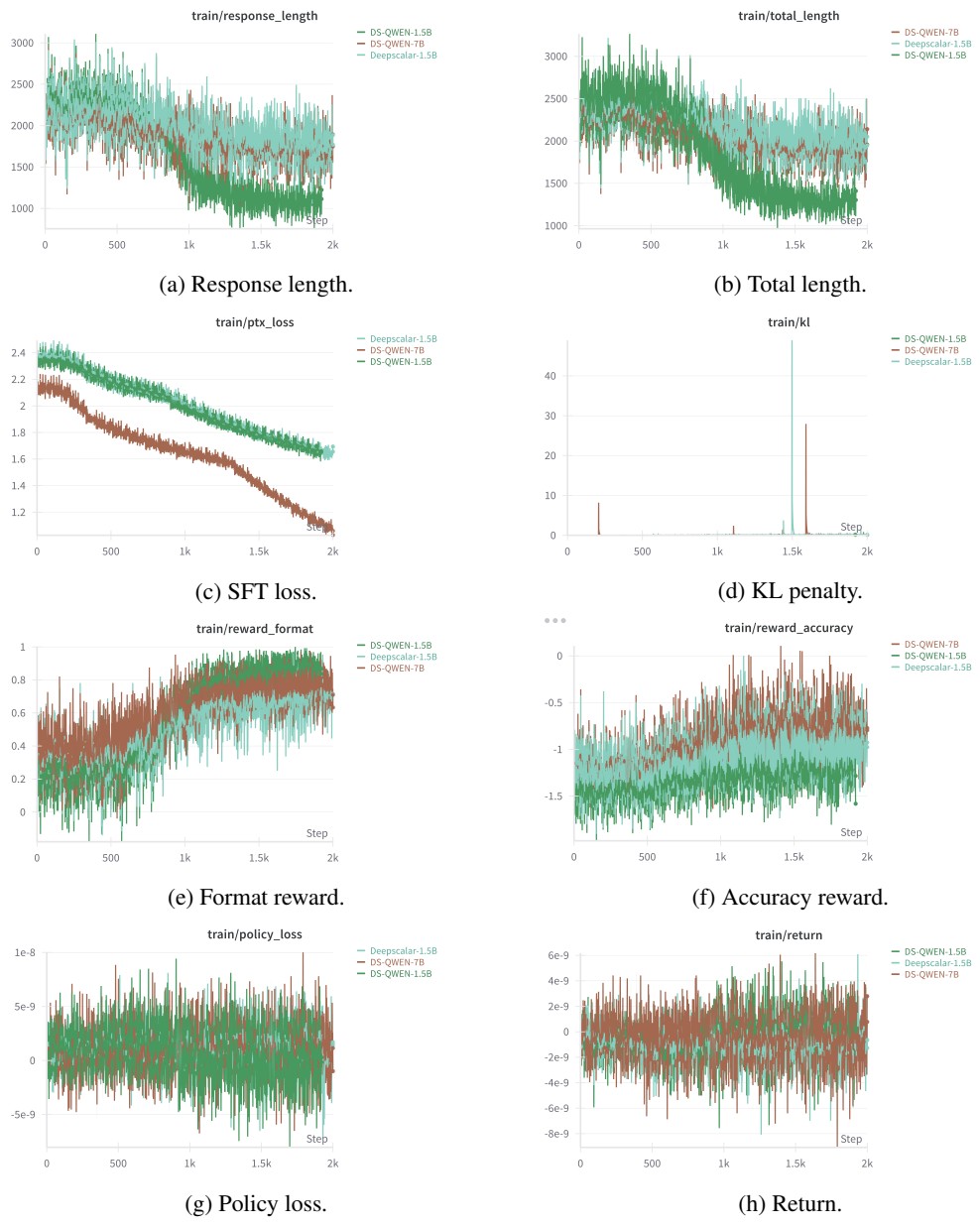

Figure 36: Training dynamics across model families: DeepSeek-Qwen1.5B, DeepscaleR-1.5B, and DeepSeek-Qwen7B (best viewed magnified).

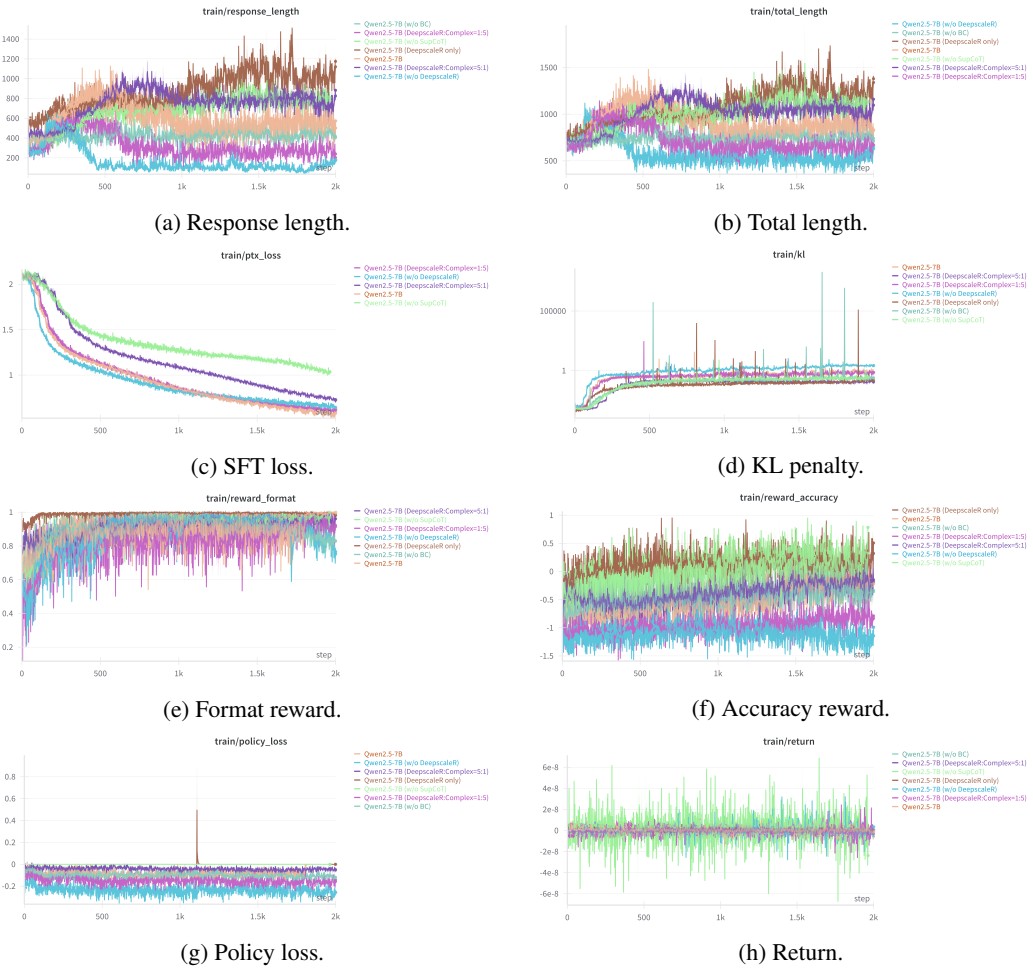

Figure 37: Training dynamics on Qwen-2.5 models (1.5B/7B-Instruct) and ablation studies (best viewed magnified).

