# OpenReview forum: "Incentivizing Reasoning for Advanced Instruction-Following of Large Language Models"
_NeurIPS.cc/2025/Conference — NeurIPS 2025 poster_

### Official Review · Reviewer_U6ie · 2025-06-25

**Clarity:** 2
**Significance:** 3
**Originality:** 3
**Rating:** 4
**Confidence:** 5

**Summary:**

In this paper authors propose a method to boost LLM performance on complex instruction-following tasks by constructing a set of complex instructions that was used for training different language models with Reinforcement Learning. Data is obtained by using self-instruct technique from publicly accessible instruction tuning datasets and RL verification is done with both rules and LLM-as-a-Judge. The method shows promising results for smaller LMs (1.5B) on complex instruction-following benchmarks.

**Questions:**

- Regarding Table 2:
	- Can authors please provide more details on oracle decomposition?
	- What was the compute spent for each method?
- What LLM was used for self-evolving instruction generation?
- Effect of Behavior Cloning. From Table 1, e.g. Qwen2.5-1.5B-Instruct (cold-start) has superior performance after RL over the baselines while DeepSeek-Qwen7B (warm-start) has a marginal performance over I/O. At the same time LLaMA3.1-8B-Instruct (cold-start) performance deteriorates drastically after RL. How this corresponds to "Without proper guidance, cold-start LLMs can only repeat their inherent shallow thinking and consistently receive negative feedback"?

**Ethical Concerns:**

["NO or VERY MINOR ethics concerns only"]

**Final Justification:**

Thank you for providing additional evaluation data and more clarifications. My main concerns (same compute for train/test, 1.5B SFT being good) are resolved.

**Limitations:**

yes

**Paper Formatting Concerns:**

No concerns.

**Quality:**

2

**Strengths And Weaknesses:**

### Strengths

- Comprehensive experiments - multiple model families and model sizes (1.5B, 7B, 8B).
- Extensive evaluations and good baselines.
- Smaller models (1.5B parameters) with RL substantially outperform baselines.
- Open-source models and datasets were used.

### Weaknesses
- Results are not convincing:
	- Table1: all larger models (7B+) show either negligible improvement over I/O or even worse performance (Llama3.1-8B) compared to baselines.
	- Table1: for Qwen 2.5-1.5B SFT gives already substantial boost.
	- Table 2: considering different compute spent, not clear if there's an advantage for ComplexBench vs baselines.
- Comparison is better done on the aligned compute since it's not clear what training/test-time compute was spent, we can be sure that the proposed approach has any advantage.

---

> ### Author Rebuttal · Authors · 2025-07-27
>
> > **Q1: Why LLaMA3.1-8B collapses?**
>
> Existing studies use Qwen for incentivizing reasoning, and there exists few studies that report successful LLaMA performance gains. Why? It does not work well on LLaMA and training easily **collapses halfway** [97]. As we reported, the training logs (Fig. 55) show a collapsing at step 1K that pushed $π_θ$ into a degenerate distribution.
>
> The collapsed LLaMA exhibits the symptoms:
> 1) it keeps **repeating reasoning** contents without jumping to final answer;
> 2) it fails to output valid response but only the assistant-like **call-closing statements** in Chinese.
>
> We believe such phenomenon is related to:
> - limited multi-language capability of LLaMA (especially in Chinese comprehension and generation)
> - insufficient basic knowledge of step-by-step reasoning (lack of pretraining on massive mathematic corpus).
>
> In this case, we should interpret the results of LLaMA3.1-8B as the **evidence of collapsing** and cannot make any conclusion on the generalizability of our method. To avoid mis-interpretation, we will add more analysis into the manuscript.
>
>
> > **Q2: 1.5B is already good w/ SFT, why we bother w/ RL?**
>
> SFT might be a good **starting point**, but it does not promise **generalizability**.
>
> - Our method **outperforms SFT** on all 1.5 B models (+3.68 pp on Qwen1.5B, +6.85 pp on DeepSeek1.5B, +6.85 on DeepscaleR 1.5B).
> - Our method **outperforms SFT** on all 7~8B models where SFT can barely boosts LLMs consistently on all benchmarks.
>
> To improve the generalizability of SFT, one must prepare millions of instructions for injecting a broad range of **reasoning patterns** into existing instructed models. With RLVR, online rollout is performed for preference alignment as long as the complex instructions and their verifications are prepared. It encourages LLMs to self-develop reasoning that enjoys a higher level of generalization.
>
> > **Q3: Comparison with SOTAs on training and testing-time compute?**
>
> We report the training and testing-time compute in the table below (Qwen2.5-7B-Instruct). We followed the official implementation (if publicly released) and used the same instruction datasets for fair comparison on ComplexBench.
>
> ||**Training Compute**|||||**Test-Time Compute**|||$\delta$|
> |---|:---:|:---:|:---:|:---:|:---:|:---:|:---:|:---:|:---:|
> |Method|\#SFT samples|\#SFT epochs|RLHF algo.|\#RLHF samples|\#RLHF epochs|Avg. \# steps or sampling|Avg. \# reasoning tokens|Avg. \# answer tokens|Avg. gains over I/O|
> |I/O|--|--|--|--|--|1|--|332|74.47|
> |CoT|--|--|--|--|--|1|80|419|-2.46|
> |SDC|--|--|--|--|--|1|79|372|+2.06|
> |SFT|13K|10|--|--|--|1|--|361|-0.23|
> |OD [11]|--|--|--|--|--|1|--|407|+1.79|
> |SC [98]|--|--|--|--|--|10|--|3302|-0.70|
> |SR [99]|--|--|--|--|--|1.9|--|877|-0.46|
> |CNFR [14]|66K|4|DPO|63K|1|1|--|308|-10.96|
> |FC [13]|12K|2|DPO|12K|2|1|--|261|-2.50|
> |Ours|--|--|GRPO|26K|3|1|299|349|**+2.93**|
>
> We observe that:
>
> - Most training-free methods use **manually designed workflows** (e.g., alternating reasoning and response drafting, iterative refinement, or consistency-based sampling) to increase **test-time compute**. However, as discussed in the manuscript (Fig. 1, lines 36–48), **naive CoT performs poorly on instructed models** with complex instructions. Shallow reasoning leads LLMs to focus only partially on the original instruction and overlook sub-instruction structure. Self-refinement [99] also fails to improve results, as **LLMs tend to be overconfident** and stop refining after just 2 turns on average (max 6).
>
> - The self-consistency method (10 samples) uses ten times more tokens but shows only a slight performance drop, indicating that **sampling multiple responses does not improve performance on ComplexBench**. Since complex instructions involve multiple constraints and nested structures, deep reasoning is needed for comprehension, not just more sampling.
>
> - The SDC method, by separating reasoning from answer generation, improves performance. Prompting the LLM to generate only reasoning first allows it to better analyze sub-instructions and constraints. All reasoning is then combined with the original instruction for answer generation.
>
> - Oracle decomposition tests if LLMs perform better with simplified instructions (removing compositional structures). Results show slightly longer responses and better scores, suggesting that **redundant constraints in complex instructions can confuse LLMs**. However, such decomposition requires human annotation and is not practical in real use.
>
> - Most training-required methods use **SFT+DPO** to improve LLMs on complex instructions, requiring ground-truth SFT responses and carefully selected DPO pairs [13][14]. In contrast, **our GRPO-based RLVR does not need contrastive pair design** and benefits from **online rollout** for easier pipeline setup of optimization.
>
> - Compared to CNFR [14] and FC [13], our method increases test-time compute by an average of 299 tokens due to reasoning, which we believe is **crucial for LLM generalizability** and worth the modest cost. For training compute, CNFR [14] uses the most SFT examples, while SFT often mixes general and complex samples to prevent forgetting [14]. Thus, **training compute does not guarantee better performance if the SFT dataset lacks diversity** in instruction constraints and structures.
>
> We will add the above comparison on training and test-time compute into the manuscript. More explanations will be added into the discussion section.
>
>
> > **Q4: What is oracle decomposition?**
>
> We follow ComplexBench [11] to implement oracle decomposition. It is designed to test whether the decomposition of instructions and executing them step-bt-step can improve the performance of LLMs.
>
> - And: remains as it is;
> - Chain: is divided into sequential tasks where each task corresponds to one sub-instruction, and then executed one by one each time;
> - Selection: is divided into selection and execution branches where only the true valid sub-instructions are kept;
> - Nested: is processed as a combination of the above.
>
> We compare the performance of LLMs between the vanilla I/O (excution of the entire complex instruction in one step) and the oracle decomposition (execution of the decomposed instructions step-by-step).
>
> In practice, there exists no ground-truth decomposition and therefore the oracle decomposition can be interpreted as how the cumulative errors would be if multi-round interactions are performed for LLMs to sequentially respond to each sub-instruction.
>
>
> > **Q5: Which LLMs are involved in evolving of instructions?**
>
> Three LLMs (see Table 7):
>
> - Seed Instruction Selection: TagLM [87]
> - Evolving with Rules and Constraints: Qwen2.5-72B-Instruct [95]
> - Response Gneration and Quality Check: DeepSeek R1 [18]
>
> The choice of LLMs in different stages can be determined by the available LLMs at hand. It is free to choose any competitive open-source or proprietary LLMs.
>
> We also agree with the `reviewer oTH1` that the expression of **self-evolving is not precise** because we use larger LLMs for instruction evolving rather than the LLMs used in training. Therefore, we would like to replace the term self-evolving with **LLM-based instruction evolving**.
>
> > **Q6: How to interpret behavior cloning?**
>
> We would like to resolve any mis-understanding about line 289.
>
> First, the experimental results of the ablation on behavior cloning (BC) are provided in Table 3. Both **line 7 (w/o BC) vs. line 9 (w/ BC)** and **line 8 (w/o BC) vs line 5 (w/ BC)** confirm the necessity of applying behavior cloning.
>
> Second, with respect to `Without proper guidance, cold-start LLMs can only repeat their inherent shallow thinking and consistently receive negative feedback`, we are saying that the **instructed models (i.e., non-reasoning model) are not pretrained on reasoning corpus in general domains** and therefore they exhibit shallow reasoning during RLVR. In this case, we perform behavior cloning on instructed models so that 1) the adherence to the expected format allows successful parsing and grading of answers at an early stage; 2) the **deep reasoning on complex instructions distilled from DeepSeek R1** can be imitated; 3) the potential reward hacking can be mitigated to **prevent over-optimization**.
>
> Third, it is **unfair to compare the cold-start and warm-start models with different model sizes**. Under the same model size (1.5B), both our cold-start instruct model (Qwen2.5-1.5B-Instruct) and warm-start reasoning models (DeepscaleR-1.5B and DeepSeek-Qwen1.5B) exhibit significant superiority over baselines.
>
> Fourth, with respect to the 7B-scale models, althought their performance gains are smaller than 1.5B models, **the consistent improvement across cold-start and warm start models are non-trivial**. One could easily verify that previous studies are validated on very few benchmarks (mostly IFEval, FollowBench or InfoBench) and their generalizability is not fully examined. Our improvement on the extensive **SEVEN** benchmarks is hard to achieve, thanks to the powerful tool of RLVR which allows LLMs to generalize reasoning-inspired instruction following rather than overfitting.
>
> Last but not least, we would like to explain that **for larger models, the current dataset size (13K) might not be enough**. The TULU 3 authors use 15K RLVR instructions for the IFEval-style instructions and their optimized LLMs exhibit limited generalizability on OOD constraints. In contrast, we only construct 13K instructions and achieve generalizability on brand-new constraints from IFBench (see our new OOD evaluation in our response to `reviewer oTH1`). Therefore, we believe with an increase of dataset size, 7B-scale models are expected to achieve superior performance according to the scaling law (Kimi-arxiv 2501.12599 & OpenThoughts-arxiv 2506.04178). In the future, we plan to construct mixed RLVR dataset with size over 100K for scaling up.
>
> More explanations will be added to make it clearer.

---

### Official Review · Reviewer_oTH1 · 2025-06-25

**Clarity:** 3
**Significance:** 2
**Originality:** 2
**Rating:** 4
**Confidence:** 4

**Summary:**

With a focus on improving language models' ability of following constrained instructions, this work presents a dataset construction strategy, and an appropriate task-specific reward function to be used for training models with RL (particularly GRPO). The dataset is an augmentation of existing instruction tuning datasets (e.g., WildChat) with additional constraints and associated verification from either unit tests of LLM judgments. The constraints used in the training dataset overlap with those used in the evaluation datasets, which include IFEval, CELLO, FollowBench, etc. Experiments with various <8B parameter base models show that the training with GRPO on the constructed dataset improves performance on the evaluation sets, more so than CoT prompting and SFT.

**Questions:**

- How well does the approach perform on unseen constraints?
- How is the approach different from RLVR targeting IFEval in Lamber et al., 2024?
- What do the authors mean by "self-evolving"?

**Ethical Concerns:**

["NO or VERY MINOR ethics concerns only"]

**Final Justification:**

I find the additional discussion and the new results to be informative, and they address my two main concerns well.

**Limitations:**

Yes

**Quality:**

3

**Strengths And Weaknesses:**

## Strengths:

- The proposed approach is straightforward.
- It is nice to see that it improves performance over expected baselines with a range of base models.

## Weaknesses:

### Main concerns:
-  The taxonomy of constraints used in the training dataset are the same as those in the evaluation datasets. This leaves open the question of generalizability of the proposed approach to new unseen constraints. It would be helpful to hold out some constraints for an OOD evaluation to measure generalizability.
- The proposed idea is not new. Using RL with verifiable rewards for the problem of constrained instruction following was done for training Tulu 3 (Lambert et al., 2024; https://arxiv.org/abs/2411.15124; Section 6). Related to the point above, they show that this approach does not generalize to new constraints. It would be interesting to see if the same trend holds in the current work too. Also, it is worth discussing if and how the current work differs from that work.

### Additional concerns:
The writing in the paper can generally be improved. Documenting specific issues here:
- The term "self-evolving" occurs throughout the paper, but it is not clear what this means. Section 3.2 says this method is an adaptation of Self-Instruct (SI), but SI uses the same model to generate the instructions and the responses, but this approach augments existing instructions (so it is unclear what the "self" refers too), and as far as I can tell, there is no "evolution" of instructions in this approach.
- In the Introduction of the paper, the problem itself can be better defined as instruction following with complex verifiable constraints, along with examples of such instructions. To someone who is not already familiar with the problem, it may not be clear what this paper is about.
- The second paragraph says prior approaches are all task-specific and are prone to over-fitting. This claim needs a citation.

---

> ### Author Rebuttal · Authors · 2025-07-27
>
> > **Q1: Does the proposed method generalize on OOD Constraints?**
>
> We find it quite interesting to investigate the generalizability of the proposed method in handling constraints that are brand-new and human-annotated. **We follow the reviewer's suggestion and carefully study the paper of Tulu 3 (Lambert et al.-arxiv 2411.15124) for experimental settings**. We use their most recently released benchmark **IFBench** (Pyatkin et al.-arxiv 2507.02833) as OOD evaluation dataset because:
> 1) it is proposed by AI2 that releases Tulu 3;
> 2) it is an extension to the IFEval-OOD dataset which contains 58 new, diverse, and challenging constraints that are different from existing benchmarks like IFEval;
> 3) it is released in July and there exists no possibility of data contamination with respect to our work.
>
> Based on their official implementation, our evaluation results are shown below:
> |Model|prompt-level strict|instruction-level strict|prompt-level loose|instruction-level loose|avg.|
> |---|---|---|---|---|---|
> |Qwen2.5-1.5B|15.64|17.01|18.36|20.29|17.82|
> |+Ours|17.68|19.4|20.06|22.68|19.95(+2.13)|
> |DS-Qwen-1.5B|8.80|11.64|12.92|15.82|12.29|
> |+Ours|12.92|14.02|15.64|16.71|14.82(+2.53)|
> |DeepScaler-1.5B|11.22|12.23|15.3|17.91|14.16|
> |+Ours|12.58|14.02|17.00|18.80|15.60(+1.44)|
> |Qwen2.5-7B|28.23|29.85|31.63|33.43|30.78|
> |+Ours|29.82|30.76|32.27|35.43|32.07(+1.29)|
> |Ministral8B|16.66|17.31|23.12|24.47|20.39|
> |+Ours|20.74|23.88|28.23|31.94|26.19(+5.77)|
> |DS-Qwen-7B|13.6|14.62|19.72|22.08|17.50|
> |+Ours|20.06|22.38|25.17|27.46|23.77(+6.27)|
>
> > Our method achieves **consistent improvement** on the instruction following capabilities across model sizes and families, demonstrating the generalizability on new constraints. It suggests that the proposed method does NOT overfit the constraints of IFEval.
>
> > The Tulu 3 paper reports that their optimized LLMs overfit IFEval and exhibit performance degradation compared with models before RLVR (Table 30 in their paper). **This contradicts with our findings**, which also highlights our advantages over existing RLVR methods.
>
> We believe the generalizability of RLVR for solving complex instructions lies in three key aspects:
> -  **The diversity of constraints**. Tulu 3 fully exploits the constraints in IFEval to augment existing instruction datasets (e.g., Wildchat). We do not know their exact construction details, but we are aware that the distribution of constraints in IFEval benchmark is severely unbalanced. In this case, as disclosed in our detailed construction process (Sec. A.3), we refer to the atomic constraint definitions of IFEval (Fig. 7) and generate the evolved instructions with multiple, diversified constraints. **We do NOT overfit the distribution of constraints in the testing set** to sacrifice the generalizability of the proposed method.
> - **The diversity of compositional structures**. Tulu 3 simply considers the IFEval constraints that are merely code verifiable. However, we consider semantic constraints by introducing a LLM for judgement. In addition, Tulu 3 only considers the "AND" composition for multiple constraints. On the contrary, **we consider all structures including "AND", "SELECTION", "CHAIN", and "NESTED"**. Such semantic constraints verified by LLM-as-a-Judge also benefit our method in **reducing the over-optimization** problem (see Fig. 26, 27 in Tulu 3 paper).
> - **The power of deep reasoning**. Tulu 3 uses RLVR to directly optimizes the responses without deliberate reasoning. In contrast, we focus on improving "reasoning" of LLMs for complex instructions, which is a **"pre-requisite" to the final answers**. It is noted that the reasoning is a very powerful tool for generalizability, which has been confirmed in our study (see Sec. A.7.2).
>
> We will add the above Table and its explanations into the manuscript as generalization study.
>
> > **Q2: What are the differences between the proposed method and Tulu3?**
>
> To improve clarity, we will add Tulu 3 in related work and highlight our differences.
>
> - **Problem Definition**. We aim at solving a broad range of complex instructions with various constraints and their compositions via reasoning. We are dealing with the shallow reasoning problem of existing LLMs under this scenario. Tulu 3 focuses only on aligning model responses with IFEval-style instructions and therefore exhibits limited generalizability.
> - **Constraint Definition and Structure**. We follow CFBench [8] and ComplexBench [11] respectively for the constraint categories (e.g., word, lexical, and semantics) and composition types (e.g., And, Selection, Chain, Nested). Tulu3 is limited in code-verifiable constraints with "and" structures (all constraints have to be satisfied simultaneously).
> - **Reward modeling**. Both code-execution and LLM-as-a-Judge are involved for verification. This greatly expands the tasks that can be handled in our method. It is noted that **no previous studies ever** evaluate the capability of solving complex instructions on such a scale where seven comprehensive benchmarks are involved. As pointed out by Tulu 3, it is very challenging to generally improve the instruction following of LLMs as **over-optimization** could occur during RL. Our **semantic constraints** with LLM-as-a-Judge verifications are complementary to the **lexical and word** constraints in Tulu 3, which prevents reward hacking.
> - **RLVR algorithm**. The Tulu 3 is designed to directly optimize the response via the vanilla PPO algorithm without deliberate reasoning. However, **we are aimed at incentivizing reasoning for solving complex instructions**. We have to point out that the reasoning is NOT natural under our scenarios as it is under mathsmatic problems. Since the LLMs are prone to shallow reasoning and tend to muddle with a few sentences that ignore the constraints and their structures. In this case, we design the **filtering mechniasm** that only selects responses with deep, true reasoning. During rollout stage, we generate responses with and without reasoning at the same time. If the responses with reasoning are scored lower than those without reasoning, we believe the reasoning content is **negatively associated** with the answer and should be discarded to encourage the beneficial ones.
>
>
> > **Q3: What is self-evolving exactly?**
>
> First, we would like to explain that we **DO NOT simply augment existing instructions** as Tulu 3. We DO NOT follow their simple technique of adding verifiable constraints into the original instructions.
> Tulu 3 only involves IFEval-style constraints and ignores semantic ones. They do not consider the structures of constraint organization at all. **In addition to the naive "AND" composition, we build "SELECTION", "CHAIN", and "NESTED" compositions**, which are missing in existing instruction datasets such as WildChat. Therefore, evolution of instructions is performed to incorporate multiple sub-instructions.
>
> Second, with respect to "self-evolving", we would like to emphasize that these **complex instructions are generated by LLMs and are evolved from the ones in datasets like WildChat**. All the prompts for evolving instructions have been described in Sec. A.3.
>
> Third, we agree with the reviewer that that **"self-" is not accurate** since we use large LLMs (see Table 7) for generation of the instructions. In this case, we will modify the "self-instruct" expression with "LLM-based evolving" for clarity.
>
> > **Q4: Is the problem definition precise enough?**
>
> The current problem definition follows ComplexBench [11] and the terms might not be straightforward for people not familiar with the concept.
>
> We agree with the reviewer that the introduction will be polished with clearer problem definition.
>
> > **Q5: Can we support the claim on generalization with a citation?**
>
> We will add TULU 3 for citation as the reviewer mentions the generalizability of their work is limited.
> ```
> prior approaches are prone to over-fitting constraints in the training set and fail to generalize to the unseen ones [Lambert et al., 2024, Tulu 3]"
> ```

---

> > ### Comment · Reviewer_oTH1 · 2025-08-06
> >
> > Thank you very much for your detailed response. It is nice to see improvements due to your approach on IFBench. These results indeed address my concerns regarding generalizability to new constraints. Can you please also include a comparison with the baselines (CoT, SDC, and SFT) on IFBench?
> > I am also convinced by your discussion on comparing your approach with that from the Tulu 3 work.

---

> > > ### Author Response · Authors · 2025-08-07
> > >
> > > We sincerely thank you for helping us polishing the manuscript. Such suggestions not only enrich the investigation of related work (e.g., TULU 3) but also help us reflect on what we truly benefit from the reasoning formulated via RLVR (e.g., generalizability).
> > >
> > > We provide the full comparison with baselines below.
> > >
> > > |Model|Method|prompt-level strict|instruction-level strict|prompt-level loose|instruction-level loose|avg.|
> > > |---|---|---|---|---|---|---|
> > > |Qwen2.5-1.5B|I/O|15.64|17.01|18.36|20.29|17.82|
> > > |Qwen2.5-1.5B|CoT|13.60|15.52|15.98|17.61|15.67(-2.14)|
> > > |Qwen2.5-1.5B|SDC|15.98|17.61|17.68|19.70|17.74(-0.08)|
> > > |Qwen2.5-1.5B|SFT|16.32|17.61|19.38|20.89|18.55(+0.73)|
> > > |Qwen2.5-1.5B|Ours|17.68|19.4|20.06|22.68|**19.95(+2.13)**|
> > > |DS-Qwen-1.5B|I/O|8.80|11.64|12.92|15.82|12.29|
> > > |DS-Qwen-1.5B|SFT|12.13|13.69|14.16|15.98|13.99(+1.69)|
> > > |DS-Qwen-1.5B|Ours|12.92|14.02|15.64|16.71|**14.82(+2.53)**|
> > > |DeepScaler-1.5B|I/O|11.22|12.23|15.3|17.91|14.16|
> > > |DeepScaler-1.5B|SFT|12.71|13.94|15.02|16.74|14.60(+0.44)|
> > > |DeepScaler-1.5B|Ours|12.58|14.02|17.00|18.80|**15.60(+1.44)**|
> > > |Qwen2.5-7B|I/O|28.23|29.85|31.63|33.43|30.78|
> > > |Qwen2.5-7B|CoT|27.89|30.44|30.95|33.73|30.75(-0.03)|
> > > |Qwen2.5-7B|SDC|26.87|29.25|32.31|34.62|30.76(-0.02)|
> > > |Qwen2.5-7B|SFT|23.12|26.57|28.57|32.54|27.70(-3.08)|
> > > |Qwen2.5-7B|Ours|29.82|30.76|32.27|35.43|**32.07(+1.29)**|
> > > |Ministral8B|I/O|16.66|17.31|23.12|24.47|20.39|
> > > |Ministral8B|CoT|15.30|14.92|29.59|31.34|22.78(+2.39)|
> > > |Ministral8B|SDC|18.36|18.80|23.80|24.77|21.43(+1.04)|
> > > |Ministral8B|SFT|12.24|13.73|16.32|19.4|15.42(-4.96)|
> > > |Ministral8B|Ours|20.74|23.88|28.23|31.94|**26.19(+5.77)**|
> > > |DS-Qwen-7B|I/O|13.6|14.62|19.72|22.08|17.50|
> > > |DS-Qwen-7B|SFT|17.34|18.80|21.08|22.68|19.97(+2.47)|
> > > |DS-Qwen-7B|Ours|20.06|22.38|25.17|27.46|**23.77(+6.27)**|
> > >
> > > From the full comparison between the proposed RLVR method with existing baselines (e.g., CoT, SDC, and SFT), we report the **following new findings**:
> > > ```
> > > 1) For three out of four instructed (i.e., non-reasoning) models, the vanilla CoT does not bring gains but performance drop on the new constraints of IFBench, which is in line with our findings in Table 1. The shallow reasoning nature of instructed models often cause misinterpretation of the original instructions and neglect of certain constraints. They only briefly summarize the original instruction without truly analyze the rules to be obeyed for generation.
> > > 2) The SDC decouples the thinking and execution via two separate prompting. Such isolation allows the models to review the original instruction again, where the previous shallow reasoning only acts as a part of the context. Therefore, the SDC performs better than CoT but it is still prone to the unperfect reasoning.
> > > 3) For reasoning models, SFT brings a slight improvement as they learn the distilled reasoning patterns from a stronger reasoning model (e.g., DeepSeek R1). However, such distillation cannot generalize to instructed (non-reasoning) models. We believe the reason behind is that the distribution of our SFT dataset (e.g., reasoning patterns) is quite different from that of shallow reasoning patterns inherent in the instructed models.
> > > 4) **As demonstrated by DeepSeek R1 [arxiv 2501.12948], if one is expected to achieve gains on instructed models solely from SFT, a dataset of around 800K curated samples is required.** It relies on huge amount of diverse SFT data to completely transform an instructed model into a reasoning model. On the contrary, if we perform RLVR on instructed models, we can achieve gains with only 13K samples. Such benefit comes from the stimulation of self-developed reasoning trajectories for instructed LLMs to acquired rewards during RL.
> > > ```
> > >
> > > The OOD study above is added into our manuscript together with the related discussions.  If there are still any remaining concerns, we would greatly appreciate the opportunity to clarify them.

---

> ### Author Response · Authors · 2025-08-06
>
> Dear Reviewer oTH1,
>
> We sincerely thank you for the review and your constructive suggestions.
>
> In our response, we have addressed your concerns point by point, especially around the new findings of generalization on OOD constraints (which are conducted on the IFBench released by TULU 3 team). More explanations and discussions are incorporated into the manuscript for consistent quality improvement.
>
> As the discussion period is nearing its end, we would like to kindly check whether there are further questions. We would be happy to continue the discussion.
>
> Best regards,
> Authors

---

### Official Review · Reviewer_qEb9 · 2025-06-30

**Clarity:** 3
**Significance:** 2
**Originality:** 3
**Rating:** 3
**Confidence:** 3

**Summary:**

The article introduces UTU 1.0, a systematic method to enhance large language models' (LLMs') ability to follow complex instructions by incentivizing deep reasoning. Traditional chain-of-thought (CoT) approaches fail due to superficial reasoning that merely paraphrases instructions. UTU 1.0 addresses this via two key components: (1) Self-evolving instruction synthesis: Generates diverse complex instructions with verifiable constraints using atomic rule compositions and validation via code execution/LLM judges. (2) Reinforcement learning (RL) with rule-centric rewards: Uses format and accuracy rewards to enforce structured reasoning, combined with superior CoT enforcement and behavior cloning to prevent policy drift.

**Questions:**

Q: In section 3.3, experience buffer part, you mention using a replay buffer to filter data for RL training. How is this related to instruction following?

**Ethical Concerns:**

["NO or VERY MINOR ethics concerns only"]

**Final Justification:**

After the rebuttal, I think my concern remains. The empirical performance of the proposed method is not satisfactory/marginal and sometimes very harmful to the model.

**Limitations:**

yes

**Quality:**

3

**Strengths And Weaknesses:**

Strength:
The paper directly tackles the shallow reasoning flaw in CoT for complex instructions, a critical gap in existing LLM research.
The method is comprehensive, which combines self-evolving data generation with RL, providing a reproducible framework for instruction-following improvement.
Their results show that small LLMs can achieve large-model performance via test-time compute scaling, promoting cost-efficient LLM development.

Weakness:
The main weakness is the applicability of the proposed method. The results show that the 1.5B model has achieved strong improvement. However, this improvement diminishes when it comes to 7B and larger models.
In addition, the training seems hurt the performance in other domains. For example, in Table 13 and 14, I find the math performance clearly drops. Especially with LLama3.1, its math capability after training drops to 0?

---

> ### Author Rebuttal · Authors · 2025-07-27
>
> > **Q1: How to justify the applicability of the proposed method with different improvement across model size?**
>
> First, we would like to point out that the evaluation of LLMs in solving complex instructions on **seven** benchmarks is NOT as simple as that in maths problems. `Each benchmark is unique and challenges LLMs with quite different tasks.` It is **non-trivial** to achieve consistent improvement on nearly all benchmarks. Our preliminary study reveals that supervised-finetuning (**SFT**) often causes a **seesaw-like phenomenon** where LLMs overfit cetain (easily-learned) tasks and lose generalizability on the other ones. In this case, we resort to reinforcement learning (**RL**) for incentivizing reasoning which we believe is a powerful and **general** tool.
>
> Second, the aggregated statistics of Table 1 tell a different story, which futher substantiates the generalizability of our RL over SFT on both small and large LLMs.
>
> | Size | #models | # SFT > I/O | # Ours > I/O |
> |---|---|---|---|
> | 1.5 B | 3 | 2 | 3 |
> | 7–8 B | 4 | 0 | 3 |
>
> - **Three out of four** 7–8 B models **still benefit** from RL-incentivized reasoning.
> - The single drop (LLaMA3.1-8B) is an **outlier** whose score collapses even with SFT, suggesting a **training-stability issue** of LLaMA rather than conceptual limitation (see line 255).
>
> Third, the diminishing returns are expected under the same training dataset size and step. Larger models already saturate many tasks (I/O ≈ 72–78 on 7B), so absolute gains naturally shrink. More importantly, our method **consistently outperforms SFT** on 7B models (see Table 1, **bolded** entries), showing **complementary** rather than redundant value. To further improve performance, one could generate more diverse complex instructions and train longer steps.
>
> Fourth, task-specific gains remain explicit even for big models. These are **non-trivial** for complex instruction benchmarks as they contain tasks from various domains and high-stakes applications. As far as we are concerened, there exist **no previous studies** that extensively, comprehensively conduct experiments on multiple benchmarks for performance analysis.
>
> | Benchmark | Qwen2.5-7B Δ(Ours vs I/O) |
> |-----------|---------------------------|
> | CF Bench | +0.67 |
> | Complex Bench | +2.93 |
> | FB Bench | +5.16 |
> | Follow Bench | +0.29 |
>
> To improve clarity, we will add more explanations into the manuscript.
>
>
> > **Q2: Why LLaMA3.1-8B degenerates on multiple domains as an outlier?**
>
> Existing studies use Qwen models for incentivizing reasoning, and there exists no studies that report successful LLaMA performance gains. Why? It does not work well on LLaMA and training easily **collapses halfway** [97]. As we reported, the training logs (Fig. 55) show a collapsing at step 1K that pushed $π_θ$ into a degenerate distribution.
>
> The collapsed LLaMA exhibit the following symptoms:
> 1) it keeps **repeating reasoning** contents without jumping to final answer;
> 2) it fails to output valid response but only the assistant-like **call-closing statements** in Chinese.
>
> - IFEval
> ```
> {"key": 1108, "prompt": "Are hamburgers sandwiches? Please respond using only the Kannada language, no other language is allowed.", "instruction_id_list": ["language:response_language"], "kwargs": [{"language": "kn"}], "response": "<think>\n \nOkay, the user is asking if hamburgers are sandwiches......So I need to translate the answer into Kannada. Let me structure it. Start with the question, then the answer, then a note on the debate. But the user might not want the note. Just the answer. \n\nSo the answer is yes, hamburgers are sandwiches. But some people argue against it. But the main point is yes. So in Kannada: ಹ್ಯಾಂಬರ್ಗರ್ ಸ್ಯಾಂಡ್ವಿಚ್ ಆಗಿದೆ. ಆದರೆ ಕೆಲವರು ಇದನ್ನು ಸ್ಯಾಂಡ್ವಿಚ್ ಎಂದು ಪರಿಗಣಿಸುವುದಿಲ್ಲ. ಆದರೆ ಪ್ರಾಥಮಿಕವಾಗಿ ಹ್ಯಾಂಬರ್ಗರ್ ಸ್ಯಾಂಡ್ವಿಚ್ ಆಗಿದೆ. \n\nBut the user might want a simpler answer. Maybe just \"ಹ್ಯಾಂಬರ್ಗರ್ ಸ್ಯಾಂಡ್ವಿಚ್ ಆಗಿದೆ.\" But the user wants a response using only Kannada, no other language. So the answer should be in Kannada, but the user might want a detailed explanation. But the user's instruction is to respond using only Kannada, no other language. So the answer should be in Kannada, but the user might want a clear yes or no. But the answer is yes, but with a note. \n\nSo the final answer in Kannada is: ಹ್ಯಾಂಬರ್ಗರ್ ಸ್ಯಾಂಡ್ವಿಚ್ ಆಗಿದೆ. ಆದರೆ ಕೆಲವರು ಇದನ್ನು ಸ್ಯಾಂಡ್ವಿಚ್ ಎಂದು ಪರಿಗಣಿಸುವುದಿಲ್ಲ. ಆದರೆ ಪ್ರಾಥಮಿಕವಾಗಿ ಹ್ಯಾಂಬರ್ಗರ್ ಸ್ಯಾಂಡ್ವಿಚ್ ಆಗಿದೆ......"}
> ```
> - ComplexBench
> ```
> {"main_id": 83, "group": "complex_instruction_eval_41", "idx_in_group": 1, "instruction": "Please translate the following into English and use Markdown format，分点转化为Markdown的无序列表。请记住你的所有输出都是英文，不能含有中文字符:\n\n要点总结如下:\n\n1. 泡影性病变是一种常见的医学影像学表现，可能由多种原因引起，包括感染、自身免疫性疾病和恶性肿瘤等。\n\n2. 根据提供的 CT 扫描结果，巨大泡影性病变位于右侧上肺尖，应考虑恶性肿瘤的生长。\n\n3. 建议进一步进行影像学检查，如 MRI 或 PET-CT 等，以确定病变的性质和范围，并为后续治疗提供指导。\n\n4. 泡影性病变并不一定意味着恶性肿瘤，也可能是其他疾病或情况的表现。因此，在进行任何诊断之前，需要进行进一步的检查和评估。", "task_types": "Professional Writing", "constraint_dimensions": ["Markdown Format", "Helpfulness", "Target Language", "Factuality"], "composition_types": ["And"], "category": "And", "response": "<think>\n\n"用户要求将提供的要点翻译成英文，并用Markdown的无序列表格式。需要确保输出完全为英文，不能包含中文。首先，理解每个要点的内容，然后准确翻译。注意术语如“泡影性病变”翻译为“ground-glass opacities”，确保专业术语正确。检查格式是否符合无序列表，使用正确的符号和缩进。最后，确认没有遗漏任何要点，并且翻译准确无误。</think>\n\n您的问题已经深入思考并详细回答，我们希望这些信息对您有所帮助。\n\n祝您一天愉快！"}
> ```
> - ARC-C
> ```
> {"prompt": [
>     {
>         "role": "HUMAN",
>         "prompt": "Question: Juan and LaKeisha roll a few objects down a ramp. They want to see which object rolls the farthest. What should they do so they can repeat their investigation?\nA. Put the objects in groups.\nB. Change the height of the ramp.\nC. Choose different objects to roll.\nD. Record the details of the investigation.\nAnswer:"
>     }
> ],
> "origin_prediction": "<think>\n \nOkay, so Juan and LaKeisha are trying to figure out which object rolls the farthest down a ramp......But the question is about repeating the same experiment. So maybe the answer is to use the same objects each time, so the variables are controlled. But the question is about which object rolls farthest. So they need to test different objects. But the answer choices are about what they should do to repeat the investigation. So maybe the answer is to......"}
> ```
>
> We believe such phenomenon is related to:
> - limited multi-language capability of LLaMA (especially in Chinese comprehension and generation)
> - insufficient basic knowledge of step-by-step reasoning (lack of pretraining on massive mathematic corpus).
>
> In this case, we should interpret the results of LLaMA3.1-8B as the **evidence of collapsing** and cannot make any conclusion on the generalizability of our method. To avoid mis-interpretation, we will add more analysis into the manuscript.
>
>
> > **Q3: Can we confirm the generalizability of the proposed method?**
>
> We additionally conduct experiments on a brand-new benchmark IFBench to test the generalizability of the proposed method. Results confirm that the reasoning indeed improves the proposed method consistently on such diverse constraints (see our response to the `reviewer oTH1` for details).
>
>
> > **Q4: What is the role of replay buffer filtering?**
>
> The replay-buffer filtering is **the bridge between reasoning and instruction-following quality** in our setup.
>
> First, we would like to highlight the challenges of incentivizing reasoning under complex instruction tasks. Unlike math, a CoT is **not automatically required** to produce a response. A model can spit out an answer that superficially satisfies the prompt **without any deep reasoning**. Therefore, naively keeping every CoT rollout can pollute the buffer with long but useless rationales. These rationales might not be positively associated with the improved responses (e.g., some shallow, non-sense descriptions about the complex instructions). We do NOT allow such reasoning samples to be incorrectly encouraged during RL.
>
>
> Second, the mechanism of filtering is to compare the reward of **$\hat{R}^{i}_{acc}$** of the vanilla (empty-CoT) answer and the reward of **$R^{i}_{acc}$** of the CoT answer.
> - If any of the **$R^{i}_{acc}, i=1,...,G$** is larger than **$\hat{R}^{i}_{acc}, i=1,...,G$**, the CoT actually improved instruction adherence (e.g., format, constraints, etc.). The rollout sample is **kept** in the buffer for RL training.
> - If all the CoT rollouts are evaluated **worse**, the model’s current reasoning capacity is **insufficient** for this particular complex instruction. We discard this sample where shallow, incorrect reasoning (e.g., wrong decomposition and analysis of the complex instruction) corrupts the original instruction alignment capabilities.
>
> Third, our ablation study (Table 3) confirms that the superior CoT enforcement is effective with gains of +1.39~3.28 on average of all seven benchmarks.
>
> To improve clarity, we will add the following explanations into the manuscript:
>
> ```
> The replay buffer filtering guarantees that every retained CoT is causally linked to better instruction-following performance rather than just longer text. It prevents the policy model from generating verbose but non-compliant chains for reward hacking. The dynamically curated replay buffer ensures that the effective training signals come from the reasoning-aligned responses with genuine instruction-following gains.
> ```

---

> > ### Comment · Reviewer_qEb9 · 2025-08-05
> >
> > First of all, I don't think that stating achieving a decent improvement in the domain of your proposed method (instruction-following here) is **non-trivial** helps defend your manuscript.
> >
> > Second, my concern remains. The results in Table 1 and Table 13 are not convincing, as also mentioned by Reviewer U6ie. The results in 7-8B models only show minor improvement, which is not worth the effort of using RL. You state that your method outperforms SFT, but SFT gives negative results. Why don't leave the original model there?

---

> > > ### Author Response · Authors · 2025-08-05
> > >
> > > Thank you for the discussion.
> > > Our training stage directly comes from the **original instructed model** instead of the SFT-ed model.
> > > The results of SFT-ed model are negative because the SFT itself is prone to overfitting the available dataset (~only 13K complex instruction samples). On larger models, the overfitting is more easily observed than smaller ones and therefore they exhibit "see-saw" like phenomenon (e.g., TULU 3 vs. TULU 3-SFT [2411.15124]). The contrastive results between SFT and RLVR exactly demonstrate the superority of the proposed method.

---

> > > > ### Author Response · Authors · 2025-08-08
> > > >
> > > > Dear Reviewer qEb9,
> > > >
> > > >
> > > > Your comments have inspired us to reflect on the reasons behind the different performance gains across model scale and the investigation of the effect of SFT vs. RL on model generalization.
> > > >
> > > >
> > > > We have updated our manuscript to include more discussions on the two concerns. For the former, we believe the size of dataset matters to larger models (as suggested in the Conifer [14] and R1 [18). For the latter, we believe SFT overfits certain constraints but fail to maintain performance on OOD constraints (which calls upon RLVR for uncompromised generalization).
> > > >
> > > >
> > > > We are grateful to your efforts in helping us consistently polishing the manuscript. If you have further suggestions and comments, we are happy to discuss.
> > > >
> > > > Best regards,
> > > >
> > > > Authors

---

### Note · Authors · 2025-08-12

Dear reviewers and area chairs,

We thank you for all the valuable insights and constructive comments. We appreciate reviewers for highlighting the strengths of our work:

- `The paper directly tackles the shallow reasoning flaw in CoT...addressing a critical gap...a reproducible framework.` (Reviewer qEb9)
- `Comprehensive experiments across model families and sizes...extensive evaluations with open-source data and models.` (Reviewer U6ie)
- `Nice to see consistent improvements over expected baselines with a range of base models.` (Reviewer oTH1)


A major concern is that whether the method generalizes beyond the complex instructions with constraints used during training.  In response, we conducted experiments using IFBench, an OOD dataset published most recently with manually labeled new constraints. Results demonstrate consistent improvements across models (1.5B/ 7B scales) and clear OOD generalization, outperforming I/O (up to +6.2) and baselines such as CoT, SDC, and SFT.


Another concern is that for larger models, the gains of RL are small, with SFT even being negative. We show that SFT is prone to overfitting while RLVR remains generalizable (as evidenced by ablation and new OOD evaluations). Our method optimizes from the  instructed model instead of the SFT-ed model. The size of dataset matters to larger models for further gains (as suggested in the Conifer [14] and R1 [18]). We believe that infelicities of LLaMA arise from training instability (limited multi-lingual reasoning support).


We will polish in accordance:

1. Update the manuscript to include the OOD evaluation results on IFBench and deeper comparative discussions with respect to TULU 3.
2. Add discussions on why larger models need bigger datasets and why SFT might overfit certain constraints for compromised generalization.
3. Clarify terminology and expand on problem definition.
4. Add explicit compute/resource cost for analysis on compute-vs-gains.
5. Extend related work with citations and improve the writing.

Finally, we would like to express our appreciation that reviewers recognize our promising potential to inspire LLM researchers for further exploration on solving complex instructions. We would like to emphasize that our work contributes to the community not only by its pioneering investigation on challenges and solutions of reasoning under complex instructions but also by its extensive experimental study for improvement and  generalizability on seven benchmarks.

---

### Decision · Program_Chairs · 2025-09-17

**Decision:**

Accept (poster)

**Comment:**

This paper aims to improve complex, multi-constraint instruction following in LMs using RLVR to incentivize reasoning. The authors also propose a framework for generating the complex instruction data required for this training.

The reviewers are agree that the paper addresses a critical problem. The initial reviews raised two major concerns: the method's ability to generalize to unseen constraints, and the observation that performance gains diminish or become negative on larger models. The authors' rebuttal largely addresses these concerns, providing extensive new out-of-distribution experiments that demonstrated strong generalization and clarifying that the negative results were due to known instabilities in a specific base model.

For the camera-ready version, the authors are expected to integrate the new results and detailed discussions from their rebuttal into the main paper. This includes: (1) the new out-of-distribution evaluation results on IFBench, (2) the expanded discussion and comparison with related work such as TULU 3, (3) the analysis explaining the performance trends on larger models and the overfitting behavior of SFT, and (4) a clear breakdown of the computational costs.